# What controls the stable isotope composition of precipitation in the Mekong Delta? A model-based statistical approach

Nguyen Le Duy[1,2], Ingo Heidbüchel[1], Hanno Meyer[3], Bruno Merz[1,4], Heiko Apel[1]

[1]GFZ German Research Centre for Geosciences, Section 5.4 – Hydrology, Potsdam, Germany
[2]SIWRR Southern Institute of Water Resources Research, Ho Chi Minh City, Vietnam
[3]Alfred Wegener Institute for Polar and Marine Research, Research Unit Potsdam, Germany
[4]University Potsdam, Institute of Earth and Environmental Science, Germany

*Correspondence to*: Nguyen Le Duy (duy@gfz-potsdam.de)

**Abstract**

This study analyzes the influence of local and regional climatic factors on the stable isotopic composition of rainfall in the Vietnamese Mekong Delta (VMD) as part of the Asian monsoon region. It is based on 1.5 years of weekly rainfall samples. In a first step, the isotopic composition of the samples is analyzed by local meteoric water lines (LMWL) and single-factor linear correlations. Additionally, the contribution of several regional and local factors is quantified by multiple linear regression (MLR) of all possible factor combinations and by relative importance analysis. This approach is novel for the interpretation of isotopic records and enables an objective quantification of the explained variance in isotopic records for individual factors. In this study, the local factors are extracted from local climate records, while the regional factors are derived from atmospheric backward trajectories of water particles. The regional factors, i.e. precipitation, temperature, relative humidity and moving distance of the backward trajectories, are combined with equivalent local climatic parameters to explain the response variables $\delta^{18}$O, $\delta^{2}$H, and d-excess of precipitation at the station of measurement.

The results indicate that (i) MLR can much better explain the isotopic variation of precipitation ($R^2 = 0.8$) compared to single-factor linear regression ($R^2 = 0.3$); (ii) the isotopic variation in precipitation is controlled dominantly by regional moisture regimes (~70%) compared to local climatic conditions (~30%); (iii) the most important climatic parameter during the rainy season is the precipitation amount along the trajectories of air mass movement; (iv) the influence of local precipitation amount and temperature is not significant during the early rainy season, unlike the regional precipitation amount effect; (v) secondary fractionation processes (e.g. sub-cloud evaporation) take place mainly in the dry season, either locally for $\delta^{18}$O and $\delta^{2}$H, or along the air mass trajectories for d-excess. The analysis shows that regional and local factors vary in importance over the seasons and that the source regions and transport pathways, and in particular the climatic conditions along the pathways have a large influence on the isotopic composition of rainfall. While the general results have been reported qualitatively in previous studies (proving the validity of the approach), the proposed method provides quantitative estimates of the controlling factors, both for the whole data set and for distinct seasons. Therefore it is argued that the approach constitutes an advancement in the

statistical analysis of isotopic records in rainfall that can supplement and precede more complex studies utilizing atmospheric models. Due to its relative simplicity, the method can be easily transferred to other regions, or extended with other factors.

The results illustrate that the interpretation of the isotopic composition of precipitation as a recorder of local climatic conditions, as for example performed for paleorecords of water isotopes, may not be adequate in the Southern part of the Indochinese Peninsula, and likely neither in other regions affected by monsoon processes. However, the presented approach could open a pathway towards better and seasonally differentiated reconstruction of paleoclimates based on isotopic records.

## 1 Introduction

The analysis of stable water isotopes ($\delta^{18}$O and $\delta^{2}$H) and their use as tracers have become an effective tool in hydrology. They are widely used to characterize water resources in a given region and to understand dynamics of hydro-geo-ecological processes such as precipitation, groundwater recharge or groundwater-surface water interactions – from the plot to the catchment scale.

Precipitation is typically composed of regional contributions where atmospheric moisture has been transported over large distances and local contributions, where the moisture has been provided by evapotranspiration within the close vicinity. Understanding the sources of precipitation and their relative contribution is critical for basin-wide water balance studies (Ingraham, 1998). Stable isotopes offer the possibility to identify the sources of precipitation and to quantify the contribution of regional and local sources (Gat, 1996). Furthermore, they can be used to investigate hydrological processes such as mechanisms responsible for streamflow generation (e.g. Kendall and Caldwell, 1998), in groundwater studies (e.g. Gonfiantini et al., 1998) and in rainfall-runoff studies (e.g. Genereux and Hooper, 1998).

Isotopic variation in precipitation has been correlated with climatic parameters such as precipitation amount, air temperature, and air mass history (Dansgaard, 1964;Rozanski et al., 1992;Gat, 1996), termed amount effect, temperature effect (Dansgaard, 1964), and circulation effect (Tan, 2009;Tan, 2014), respectively. The circulation effect describes the changes in isotopic composition in precipitation that appear because arriving moisture is coming from different areas of the ocean.

Delineating the present-day relationship between climatic factors and stable isotope variation in precipitation can also help to understand past climatic conditions at regional and global scales. However, the factors controlling isotopic variation of precipitation are numerous and complex; hence a better understanding of the climatic influences on isotopic values would improve the use of precipitation isotopes as a proxy to reconstruct paleoclimates (Yang et al., 2016).

In the Asian monsoon region, the isotopic signature of precipitation has been found to correlate with large-scale climatic parameters such as sea surface temperature and relative humidity of the air masses (Dansgaard, 1964;Merlivat and Jouzel, 1979;Clark and Fritz, 1997;Lachniet, 2009), ENSO (Ichiyanagi and Yamanaka, 2005;Tan, 2014;Yang et al., 2016) and the vertical wind shear index (Vuille et al., 2005). Other relevant processes were identified as distillation during vapor transport

(Araguás-Araguás et al., 1998;Yoshimura et al., 2003;Vuille et al., 2005;Dayem et al., 2010;Pausata et al., 2011;Lee et al., 2012;Liu et al., 2014) , re-evaporation and rain-vapor interactions (Risi et al., 2008b;Chakraborty et al., 2016).

Relations between climate and water isotopes have been analyzed by univariate statistical regression methods (e.g. Araguás-Araguás et al., 1998;Bowen, 2008), isotope-enabled global climate models (GCMs) (Yoshimura et al., 2008;Risi et al.,
2010b;Yoshimura et al., 2014;Okazaki and Yoshimura, 2017), isotope-incorporated Lagrangian models (Pfahl and Wernli, 2008;Sodemann et al., 2008), or the combination of GCMs (or Lagrangian models) with statistical analysis (Vuille et al., 2003;Vuille et al., 2005;LeGrande and Schmidt, 2009;Tindall et al., 2009;Ishizaki et al., 2012;Conroy et al., 2013). While statistical models are not able to represent the actual process causing a phenomenon (e.g. the physical controls of isotope variations in precipitation), in contrast to physical models (e.g. GCMs or Lagrangian models), they can, however, detect the
results of a process, and thus help to identify the responsible processes. Both approaches have their advantages and disadvantages and hence coexist supplementing each other. We argue that clearly taking into consideration the limitations and advantages of both statistical and physical models (discussed in next paragraphs) can enhance their power to interpret the relations between climate and water isotopes.

As illustrated in previous studies (e.g. Noone and Simmonds, 2002) and discussed in Sturm et al. (2010), the inherent
limitations of empirical (or statistical) climate reconstructions from precipitation isotopes can lead to incorrect paleoclimate reconstructions. A major limitation is the assumption that the isotopic signal is controlled by a single climatic factor and that the stationary relationship (e.g. between temperature and $\delta^{18}O$) remains valid over the entire proxy record. This mono-factorial relationship does not consider the interplay of different climatic factors and is possibly biased. Another limitation is the assumption of a constant precipitation source or similar isotopic signatures of different moisture sources throughout the study
period when using only local parameters (e.g. local precipitation) to interpret precipitation isotopes. In real cases, these assumptions are rarely fulfilled and often unrealistic because of the changes in seasonality and atmospheric circulation patterns. This is particularly true in those parts of the Asian monsoon region located in the transition zone between the Indian and Western North-Pacific monsoon where precipitation originates from both the Indian and Pacific Oceans (Delgado et al., 2012a), with the isotopic signatures of air masses originating from the Indian Ocean differing considerably from those of the
Pacific Ocean (Araguás-Araguás et al., 1998). Seasonally varying sources of precipitation have also been observed in China (Tan, 2014, and references therein), India (e.g. Breitenbach et al., 2010;Chakraborty et al., 2016), Korea (Lee et al., 2003), Thailand (Ishizaki et al., 2012), and elsewhere (Araguás-Araguás et al., 1998).

Since the pioneering work of Joussaume et al. (1984), GCMs have been frequently used for isotopic studies with at least a half-dozen GCMs (Risi et al., 2010b;Sturm et al., 2010). For a more detailed discussion about advances in the development of
GCMs, the reader is referred to Galewsky et al. (2016) and references therein. Although GCMs could provide the physical links between climate and water isotopes (Yoshimura et al., 2008), the model parameterizations are still far from perfect due to downscaling issues and intrinsic atmospheric variability (Sturm et al., 2010). Modeling isotopic composition in precipitation by GCMs has some limitations stemming from the model uncertainties, e.g. the frequently reported biases in precipitation or

temperature simulation (Mathieu et al., 2002;Lee et al., 2007;Yoshimura et al., 2008), and/or numerical inaccuracies in transport processes (Noone and Sturm, 2010). For example, the moist bias persisting in many GCMs in the tropical and subtropical middle and upper troposphere is due to excessively diffusive vertical advection (Risi et al., 2012). These limitations have obvious consequences (e.g. low correlation between simulated and observed $\delta^{18}O$) for the simulation of isotopic variations

in precipitation.

For paleoclimate reconstruction, the proxy data assimilation method has been proven to obtain adequate results (Yoshimura et al., 2014;Okazaki and Yoshimura, 2017). This approach, however, requires in-depth knowledge of the atmospheric modeling and/or data assimilation algorithm (Sturm et al., 2010). In any case, it takes a lot of effort to establish such a system if it is not already present. Particularly, even though the underlying physics is relatively simple, it would be a daunting task to develop a

GCM source code which requires tens of thousands of code lines to simulate the hydrological cycle (Sturm et al., 2010). Generally, the complexity of GCMs impedes their interpretation.

While GCMs are typically Eulerian in the sense that mass is exchanged between fixed discrete volumes, Lagrangian models (e.g. the HYSPLIT model used in this study, mentioned in section 3.5) are used to calculate the composition of infinitesimal air parcels in the atmosphere according to the mean wind field data (Galewsky et al., 2016). The transport pathway along

which air parcels travel is called a trajectory. In contrast to Eulerian models, Lagrangian models are not subject to numerical diffusion, hence they are computationally cheaper to simulate moisture sources. Because of their relative simplicity, these models are suitable to study the influences of different processes along transport trajectories (Helsen et al., 2006).They also more explicitly retain information about the history of the air parcels, which is useful to investigate controls on the isotopic composition of vapor arriving at a site of interest (Galewsky et al., 2016). In spite of their high suitability for exploring stable

isotopes in paleoclimate reconstructions, using GCMs for simulating single meteorological events is more difficult due to their coarse spatial resolution (Pfahl and Wernli, 2008). Moreover, GCMs cannot capture the seasonal cycle of water isotopes on local scales (Angert et al., 2008). For these reasons, Lagrangian models are more suitable than GCMs to investigate controls on precipitation isotopes at a given location.

Although relationships between atmospheric circulation patterns and precipitation isotopes are frequently acknowledged and

applied to reconstruct past climates, the actual causes of these relationships remain unclear (Ishizaki et al., 2012). Similarly, even if GCMs or Lagrangian models could provide much more detailed information about the fractionation processes along the transport pathways of water in the atmosphere, they cannot be used in a straightforward way to extract the impact of dominant factors and weight their relative importance for the variability of the observed isotopic signal. Statistical techniques are required to quantify the correlation between observed isotopic signal variability and regional climate change patterns

(Sturm et al., 2010). Statistical analysis techniques such as principal component analysis (PCA) (Vuille et al., 2003;Curio and Scherer, 2016), sensitivity experiments (Ishizaki et al., 2012), or machine learning techniques like random forests (Sánchez-Murillo et al., 2016) have been used to investigate dominant factors controlling isotopic composition in precipitation.

Recently, many studies have presented evidence that large-scale monsoon circulation is the primary driver of variations in precipitation isotopes instead of local controls (e.g. local precipitation amount or temperature) in some parts of the Asian monsoon region. This evidence has been found at different temporal scales including daily isotopic variability (Yoshimura et al., 2003;Yoshimura et al., 2008), seasonal isotopic variability (Araguás-Araguás et al., 1998;Kurita et al., 2009;Dayem et al.,
2010;Peng et al., 2010;Baker et al., 2015), and/or interannual isotopic variability (Vuille et al., 2005;LeGrande and Schmidt, 2009;Ishizaki et al., 2012;Tan, 2014;Kurita et al., 2015). However, the influence of the different factors has been described qualitatively only, with the exception of the study of Ishizaki et al. (2012), in which the quantitative analysis of the controls is limited to two factors (local precipitation amount and distillation of the moisture along its transport trajectories). That means, to our best knowledge, there is no study considering quantitatively the interplay of several local and regional factors.

It has been frequently stated and agreed to that local and regional factors should be considered simultaneously to explain the isotopic variation in rainfall (e.g. Johnson and Ingram, 2004). Hence, it can be hypothesized that using multiple factors in a single linear model is able to explain a larger share of the observed variance in isotopic composition. We aim at developing and testing a model-based statistical approach for the quantification of the contribution of isotopic separation processes for explaining the isotopic variation of precipitation. Such a model-based statistical method could also be applied in paleoclimate
studies, separating and quantifying the impacts of local and regional factors on the isotopic composition of local precipitation (Sturm et al., 2010), thus overcoming the shortcomings of single factor analysis.

This study uses the Vietnamese Mekong Delta (VMD) as a test case, for which isotopic data in precipitation has been collected for the first time. The rainfall samples ($\delta^{18}O$ and $\delta^{2}H$) were collected comparatively frequently (bi-weekly to weekly) over a period of 18 months. This dataset enables a better analysis of the temporal dynamics of the isotopic composition as compared
to the typical monthly Global Network of Isotopes in Precipitation (GNIP) data (IAEA/WMO, 2016). The collected data was used to characterize the isotopic composition for the Mekong Delta by means of local meteoric water lines, which were compared to other locations in South-East Asia. The local meteoric water lines (LMWLs) developed in this study can be used as a baseline for other studies using isotopic data to investigate hydrological processes in the Mekong Delta. Furthermore, the data was used to test the proposed approach for the identification and quantification of the controls on the isotopic variation of
precipitation.

The main objective of this study is to develop a model-based statistical approach that quantitatively estimates the relative contribution and the interplay of regional and local factors in controlling the isotopic variation of precipitation for a given study site. The proposed approach is based on backward trajectory analysis exploiting the benefits of a Lagrangian model (the HYSPLIT model mentioned in section 3.5), in combination with multiple linear regression (MLR) of all factor combinations
specifically considering the widespread issue of multicollinearity of the regression factors, and relative importance analysis. The effort in this study is not meant to develop a universal model to predict precipitation isotopic composition, but rather to test a comparatively simple and transferable method utilizing easily obtainable atmospheric spatial and climatic information (trajectories) to quantitatively investigate the drivers and their interplay in controlling the isotopic variation of precipitation.

## 2 Study area

The study area, the Plain of Reeds (Fig. 1), is located in the northern part of the Vietnamese Mekong Delta (VMD), between latitudes 10°42'7"N to 10°48'9"N and longitudes 105°22'45"E to 105°33'54"E. With an area of 697,000 ha, it accounts for 17.7% of the total area of the VMD. About 95% of the Plain of Reeds is used for rice paddy and vegetable cultivating, and shrimp and fish farming (Hung et al., 2014). The average elevation ranges from 1-4 m above sea level.

Located in a tropical monsoon region, the climate of the VMD has a distinct seasonality with two seasons: the rainy season (May to November) resulting from the flow of moisture from the Indian Ocean and Western North-Pacific Ocean accounting for approximately 80-90% of the annual rainfall (Tri, 2012), and the dry season (December to April) controlled by high-pressure systems over the Asian continent (Wang et al., 2001). Precipitation from the Indian monsoon is forced by the convective heat sources over the Bay of Bengal (Wang et al., 2001) and arrives earlier than precipitation from the Western North-Pacific monsoon (Delgado et al., 2012), forced by a convective heat source over the South China Sea – Philippine Sea. The average annual rainfall is 1400-2200 mm, characterized by an uneven distribution, both spatially and temporally (Renaud and Kuenzer, 2012;GSO, 2014).

During the study period, i.e. the period of isotope sampling in rainfall lasting from June 2014 to December 2015, the rainy and dry seasons are defined by the monthly precipitation amounts and the monthly number of days with precipitation for Cao Lanh (Fig. 2). The dry season is defined as the months with rainfall amount smaller than the overall average (blue line), and a monthly number of days with precipitation smaller than the overall average (red line). All other months are included in the rainy season. The definition used here is particularly developed for the local climatic conditions, the problem to be solved, and the data available. Other definitions could cause some data points to be assigned to the other season. However, those data points will most likely be from the transition period from one season to the other, i.e. other definitions would affect samples that have the least explanatory value for the actual dry and wet seasons.

The data indicates that the rainy season in 2014 lasted from May to November, and in 2015 from June to November. The dry season is thus defined from December 2014 to May 2015 and starts again in December 2015. The study period was very dry with an annual rainfall of 985 mm compared to the long-term average of 1550 mm at the station Cao Lanh. This anomaly needs to be considered in the interpretation of the results.

The annual average temperature is 27°C with the small interannual variability of about 1°C. Variation of temperature throughout the year is small with monthly averages of 25°C to 29°C (Fig. 3). The average annual relative humidity ranges from 82% to 85%, with a seasonal variation of 80% to 88% during the rainy season and 77% to 83% during the dry season (Fig. 3). The mean annual evaporation is 984 mm with a significant difference between the rainy season and the dry season. The monthly evaporation rate ranges from 67 to 80 mm and from 76 to 109 mm in the dry and rainy season, respectively. Daily sunshine duration is about 8.7 to 9.6 hours in the dry season and 5.5 to 5.9 hours in the rainy season (Renaud and Kuenzer, 2012;GSO, 2014).

## 3 Methodology

An overview of the proposed methodology is given in Fig. 4. For the derivation of local factors relevant for the isotopic composition of precipitation climate data from nearby meteorological stations were collected (section 3.1). At the test location, precipitation samples were analyzed for their isotopic composition (section 3.2 and 3.3). For the construction of local meteoric water lines (LMWL), three regression methods were applied, in order to test the robustness of the LMWL against different regression methods (section 3.4). The regional factors were derived from atmospheric back trajectory modeling (section 3.5). All possible combinations of local and regional predictors were included in multiple linear regressions, and their ability to explain the observed variance of the isotopic composition of precipitation was determined with performance statistics (MLR, section 3.6). Finally, the influence of the different factors on the explained variance of isotopic composition was determined by relative importance analysis (section 3.7).

### 3.1 Climatic and isotopic data collection

Daily precipitation, air temperature, and relative humidity were obtained from the National Centre for Hydro-Meteorological Forecasting (NCHMF) of Vietnam at two stations (Chau Doc, Cao Lanh, Fig. 1) for the period 2012-2015. Long-term (more than 30 years) climatic data at these stations was provided by SIWRP (2014). Precipitation isotopic data from six selected GNIP stations (IAEA/WMO, 2016) located in the Indochinese Peninsula (Fig. 1) was collected for comparison with the isotopic data sampled in this study in the Plain of Reeds.

### 3.2 Precipitation sampling at An Long

Precipitation at An Long in the Plain of Reeds (Fig. 1) was sampled on a weekly basis between June 2014 and May 2015 and twice a week between June 2015 and December 2015. The rain collector was a dip-in sampler type as described in the guidelines of the IAEA technical procedure for precipitation sampling (IAEA, 2014). It consists of a 5 L accumulation glass bottle fitted with a vertical 14 cm diameter plastic funnel that reaches almost to the bottom to prevent evaporative losses, and a pressure equilibration plastic tube (2 mm in diameter and 15 m in length) to minimize evaporation. All collected samples were stored in 30 mL plastic sample bottles with tight screw caps to avoid evaporation effects. Between collection and laboratory analysis, the samples were stored in the dark.

### 3.3 Isotopic laboratory analysis

All stable isotope samples were analyzed at the laboratory of the Alfred-Wegener-Institute (AWI) in Potsdam, Germany. The measurements were performed with a Finnigan MAT Delta-S mass spectrometer using equilibration techniques to determine the ratio of stable oxygen ($^{18}O/^{16}O$) and hydrogen ($^{2}H/^{1}H$) isotopes. Analytical results were reported as $\delta^{2}H$ and $\delta^{18}O$ (‰, relative to Vienna Standard Mean Ocean Water - VSMOW) with internal $1\sigma$ errors of better than 0.8‰ and 0.1‰ for $\delta^{2}H$ and

$\delta^{18}O$, respectively. The measuring procedure is described in detail in Meyer et al. (2000). The deuterium excess (d-excess) was calculated following Eq. 1 (Dansgaard, 1964):

d-excess = $\delta^2H$ - 8*$\delta^{18}O$                                                        (1)

### 3.4 Development of local meteoric water lines

For the development of local meteoric water lines (LMWL) three methods of linear correlation between $\delta^{18}O$ and $\delta^2H$ values were applied, in order to test the robustness of the LMWL against different regression methods:

  1) ordinary least squares regression (OLSR),

  2) reduced major axis (RMA) regression,

  3) precipitation amount weighted least squares regression (PWLSR).

OLSR and RMA give equal weight to all data points regardless of their precipitation amount, while PWLSR minimizes the effect of smaller precipitation amounts (Hughes and Crawford, 2012), which are more likely to have a lower d-excess due to re-evaporation of raindrops below the cloud base (Jacob and Sonntag, 1991), or biases in the sampling method (Froehlich, 2001). OLSR tends to be more useful when investigating the interaction between hydro-climatic processes and stable isotope signatures in precipitation, whereas PWLSR is adequate in studying surface and groundwater hydrology (Hughes and Crawford, 2012). For a more detailed discussion, the reader is referred to IAEA (1992); Hughes and Crawford (2012); Crawford et al. (2014).

The quality of fit of the three LMWLs resulting from OLSR, RMA, and PWLSR was evaluated based on the coefficient of determination $R^2$, also referred to as explained variance, the standard error SE and the statistical significance value (p-value). The regression model indicates a good fit to the data when $R^2$ is close to 1.0, the standard error is small in relation to the magnitude of the data, and the p-value is smaller than 0.0001 (Helsel and Hirsch, 2002).

### 3.5 Back trajectory modeling

The potential locations of atmospheric moisture sources and the direction of the air mass causing precipitation before reaching An Long station were investigated via back-trajectory analysis. This investigation was performed using the PC Windows-based HYSPLIT (Hybrid Single Particle Lagrangian Integrated Trajectory) model developed by NOAA (National Oceanic and Atmospheric Administration) at the Air Resources Laboratory (ARL) (www.arl.noaa.gov/HYSPLIT_info.php). The model builds on the Lagrangian approach, using a moving frame of reference for the advection and diffusion calculation as the air parcels move from their initial location (Draxler and Rolph, 2003;Stein et al., 2015). The model parameters and inputs are starting time and height of the trajectories, trajectory duration, vertical motion options, type of climatic dataset, and the number of trajectories. The back-trajectory outputs are the hourly locations of the trajectory segment endpoints, the altitude of trajectories, and climatic parameters (e.g. precipitation, temperature, relative humidity) along each trajectory.

The 1°x1° climatic dataset generated by the global data assimilation system (GDAS) was used as input to the HYSPLIT model. This dataset was downloaded from the ARL web server using the HYSPLIT graphical user interface. 10-day backward trajectory analysis was performed every 6 hours between 01-JUN-2014 and 31-DEC-2015 at the sampling site (10.72°N, 105.24°E) for three levels at 1000, 1500, and 2000 m above ground (corresponding to barometric surfaces of approximately

900, 850, and 800 hPa). These barometric surfaces were chosen because the 850-hPa vorticity is highly indicative of the strength of the boundary layer moisture convergence and of rainfall in regions away from the equator (Wang et al., 2001), hence rainfall is expected to mostly originate from these altitudes. Consequently, the combination of 800 hPa and 850 hPa barometric surfaces accounts for the fact that rainfall is expected to mostly originate between 1500 and 2000 m above ground level. Correspondingly, the combination of the barometric surfaces of 800, 850 and 900 hPa means that rainfall is expected to

mostly originate between 1000 and 2000 m above ground level. In total, 6948 backward trajectories were computed. The HYSPLIT outputs, i.e. precipitation, temperature, relative humidity, and moving distance of moisture sources, were used to investigate the influence of the different moisture sources on the variation of the isotopic composition of precipitation at An Long. In order to derive figures representative for each trajectory, accumulated precipitation, mean values of temperature and humidity of the hourly HYSPLIT output were calculated along the trajectory and used as predictors in the MLR.

Single backward trajectory computations by the HYSPLIT model can have large uncertainties. The horizontal uncertainty of the trajectory calculations by HYSPLIT has been estimated to be 10–20 % of the travel distance (Draxler and Hess, 1998). While errors in trajectory calculation computed from analyzed wind fields seem to be typical on the order of 20% of the distance travelled (Stohl, 1998), the statistical analysis of a large number of trajectories arriving at a study site would increase the accuracy of the trajectory analysis (Cabello et al., 2008). Harris et al. (2005) studied trajectory model sensitivity to the

input meteorological data (focusing on ERA-40 and NCEP/NCAR reanalysis data) and to the vertical transport method. They pointed out five causes of trajectory uncertainty, expressed as percentage of deviation of the average travel distance: 1) minor differences in the computational methodology: 3–4%; 2) time interpolation: 9–25%; 3) vertical transport method: 18–34%; 4) meteorological input data: 30–40%; and 5) combined two-way differences in the vertical transport method and meteorological input data: 39–47%. However, it would be difficult to prove that in all situations a single meteorological data set or a single

method of trajectory modeling was superior to another one (Gebhart et al., 2005;Harris et al., 2005). More details about the uncertainties in trajectory modeling were provided by (Stohl, 1998), later by (Fleming et al., 2012) and references therein.

In this study, several quality control measures were applied, as recommended in Stohl (1998), to increase confidence in the HYSPLIT-generated back trajectories and to improve the validity of the air mass history. Firstly, trajectories were computed for three pressure levels (900, 850, and 800 hPa). Similar origins of atmospheric moisture for these pressure levels suggest that

resolution errors and atmospheric shearing instabilities are negligible which increases the confidence in the results. Secondly, we use the shortest possible integration time step (i.e. 1 h) and a small value for the parameter TRATIO (0.25), which is the fraction of a grid cell that a trajectory is permitted to transit in one advection time step. Smaller values of TRATIO help to minimize the trajectory computation error using the HYSPLIT model. Thirdly, the statistical analysis of a large number of

trajectories (e.g. trajectory cluster analysis) arriving at the study site was applied to confirm the accuracy of the trajectory analysis. The trajectory cluster analysis is conducted by the HYSPLIT model to group trajectories with similar pathways. The cluster analysis merges these trajectories that are near each other and represents those clusters by their mean trajectory. Differences between trajectories within a cluster are minimized while differences between clusters are maximized.

Computationally, trajectories are combined to decrease the number of clusters until the total spatial variance (TSV) starts to increase significantly. This occurs when disparate clusters are combined. This number of clusters is then selected as the optimal cluster number for sorting and combining similar trajectories. More information about the HYSPLIT cluster analysis can be found at https://ready.arl.noaa.gov/documents/Tutorial/html/.

### 3.6 Analysis of factors controlling isotopic variation in precipitation

Multiple linear regression (MLR) was used to assess how the isotopic variation in precipitation is related to regional and local controlling factors. As indicators of regional factors the output of the HYSPLIT model was used, consisting of the accumulated precipitation amount along the transport pathways (hereafter P_hysplit), mean temperature (T_hysplit) and mean relative humidity (H_hysplit) along the trajectory, and the distance of moisture sources travelled within the time frame of 10 days (D_hysplit). The local climatic factors are weekly precipitation amount (P_AL) at An Long station, and weekly mean air

temperature (T_AL) and weekly mean relative humidity (H_AL) taken from the nearby Cao Lanh station during the sampling period. These seven predictors were related to isotopic values ($\delta^{18}O$, $\delta^2H$, and d-excess) defined as response variables in the MLR. Pearson linear correlation coefficients were computed to show inter-correlations between response and predictor variables and then used to determine the importance of predictors in the MLR.

All possible subset regression models consisting of all possible combinations of predictors ($2^7-1 = 127$ models) were applied

separately for $\delta^{18}O$, $\delta^2H$ and d-excess. The coefficient of determination $R^2$ for the MLR was calculated for each subset regression. The goodness of each MLR model was evaluated based on the Prediction Residual Error Sum of Squares (PRESS) (Eq. 2) and adjusted $R^2$ (Eq. 3) (Helsel and Hirsch, 2002). The PRESS residuals are defined as $e_{(i)} = y_i - \hat{y}_{(i)}$ where $\hat{y}_{(i)}$ is the regression estimate of $y_i$ based on a regression equation computed leaving out the $i^{th}$ observation. The process is repeated for all n observations:

$$PRESS = \sum_{i=1}^{n} e_{(i)}^2 \qquad (2)$$

The selection of best models with PRESS is equivalent to a leave-one-out cross-validation, which tests the regression models for robustness and reduces the chances of model over-fitting, i.e. the chances of finding spurious regression models that provide good results for the given combination of factors and selected time period only.

The adjusted $R^2$ ($R_a^2$) is defined as follows

$$R_a^2 = R^2 - (1 - R^2)\frac{p}{(n-p-1)} \qquad (3)$$

Where p is the total number of predictors in the MLR model and n is the number of observations. The statistical significance of all linear regression was evaluated based on the p-value for the F-test as part of a one-way ANOVA analysis. A good MLR model is hereby characterized by:

(i) PRESS close to zero,

(ii) Adjusted $R^2$ ($R^2_a$) close to 1.0,

(iii) a p-value smaller than 0.0001.

For each response variable, six pressure layers (800 hPa, 850 hPa, 900 hPa, and mean values of their combinations) and 10 durations of backward trajectories (from 1-day to 10-day backward) were used. The different pressure levels and combinations were chosen in order to tackle the inherent uncertainty regarding the pressure levels from which the rainfall actually stems. Similarly, different durations of the trajectories were chosen in order to avoid fixing the a-priori unknown travel time of precipitation reaching An Long. Overall, this resulted in 7620 MLR models for each response variable $\delta^{18}O$, $\delta^2H$ and d-excess (6 pressure levels times 10 trajectory durations times 127 predictor sets). The best MLR model was then identified by the smallest PRESS value (Eq. 2). Furthermore, the goodness of fit of the MLR models was characterized based on the adjusted $R^2$ values.

### 3.7 Relative importance analysis

Relative importance analysis determines the proportion of the variance explained by the individual predictors in the regression. However, this is difficult when predictors are correlated, since multicollinearity can lead to a high sensitivity of regression coefficients caused by small changes in the model. This means that the importance can strongly shift from one predictor to another well correlated one if the data set is changed even only slightly. The leave-one-out cross-validation may be particularly vulnerable to this effect. Therefore two methods were applied, namely relative weight analysis (Johnson, 2000) which has been developed to quantify the power of predictors when they are correlated, and the relative partial sum of squares (Gardner and Trabalka, 1985). For a review of approaches to estimate predictor importance, readers are referred to Tonidandel and LeBreton (2011); Kraha et al. (2012).

Relative weight analysis approximates the relative importance of a set of predictors by creating a set of variables that are highly related to the original set of variables but are uncorrelated with each other. The response variable is then regressed on the uncorrelated set of predictors to approximate the relative weight of the original set of predictors, defined as the relative contribution of each predictor to $R^2$. This method is computationally efficient even for a large number of predictors and produces very similar results compared to more complex methods. Details are given in Johnson (2000); Tonidandel et al. (2009).

In the relative partial sum of squares (RPSS) method (Gardner and Trabalka, 1985), the total sum of squares of the response variable is partitioned based on multiple linear regression between all predictors. Briefly, the RPSS represents the percentage of the total sum of squares attributable to each of the predictors. To calculate RPSS for predictor $V_i$, the difference between

the regression sum of squares (RSS) for the full model and the regression sum of squares for the model with $V_i$ missing (RSS-i) is divided by the total sum of squares (TSS) (Rose et al., 1991), and expressed as a percentage using Eq. 4.

$$RPSS = 100 * (RSS - RSS_{-i})/TSS \qquad (4)$$

The relative importance derived by the methods above quantifies the proportion of the variance explained by the individual
regression factors, and thus identifies the dominant controls on the isotopic composition of rainfall.

## 4 Results and discussion

### 4.1 Variability of moisture sources

Because there is no daily precipitation data recorded at An Long, we used daily precipitation data at Cao Lanh instead. This is the closest national meteorological station, located approximately 37 km Southeast of An Long. Backtracking trajectories in
Fig. 5 are plotted for the days when rainfall was recorded at Cao Lanh. This is based on assumption that days with precipitation at Cao Lanh and An Long coincide.

Figure 5 shows back-calculated trajectories of atmospheric moisture prior to rainy days at An Long for the sampling period from June 2014 to December 2015. Left and right panels show the results of 850 hPa trajectories for 2014 and 2015, and the upper, middle, and lower panels show the results for the early (June – September) and late (October – November) rainy season
and dry season (December – May), respectively. Figure 6 shows the spatial distribution of vapour trajectories (cluster means) for precipitation days at An Long for 3 barometric surfaces (800, 850, 900 hPa) between June 2014 and December 2015, and the change in total spatial variance (TVS) for different cluster numbers. The TSV was used to identify the optimum number of clusters. The similarity of back-calculated trajectories (Fig. 5) and trajectory cluster analysis (Fig. 6) at three barometric surfaces (900, 850, and 800 hPa) illustrates that the trajectories and thus the source regions do not differ between different
atmospheric layers. This indicates a barotropic atmosphere, with the consequence that it is unlikely that the selection of the pressure layer for the HYSPLIT trajectories modifies the results of the MLR significantly.

Figure 5 and Figure 6 demonstrate that the dry-season precipitation (from December to May) in the Plain of Reeds mainly originates from the moisture sources of the Asian continental air masses and the oceanic air masses carried by the equatorial easterlies, whereas during the rainy season (from June to November) air masses travel a longer distance over the tropical Indian
Ocean (from June to September) and the South Pacific Ocean (October to November).

These findings for An Long agree with the general characterization of monsoonal circulation and precipitation over the Southeast Asia region, with moisture from the Indian Ocean dominating during the initial stage of monsoon evolution, and the Pacific Ocean dominating in the later stages. This indicates that the HYSPLIT model provides valid trajectories to be used in the MLR.
The mean $\delta^{18}$O values for the 5 clusters are plotted in Figure 6 (in brown). The mean cluster values are similar for the three pressure levels. Also, the mean values of the two clusters from the Indian Ocean, as well as the two clusters from the Pacific,

are similar. For a fingerprinting one also has to consider the variation of the values within the clusters, which partly overlap. This means that the $\delta^{18}O$ values of precipitation in the Mekong Delta cannot be used to uniquely identify the origin of the trajectory. However, they provide a coarse indication of their origin.

## 4.2 Isotopic composition of precipitation

### 4.2.1 Meteoric water lines

The linear-regression analyses of 74 pairs of $\delta^{18}O$ and $\delta^2H$ values at An Long yield LMWLs for the Plain of Reeds as follows:
1) Ordinary least squares regression (OLSR):
$\delta^2H = (7.56 \pm 0.11)* \delta^{18}O + (7.26 \pm 0.67)$
(SE = 2.26; $r^2$ = 0.99; p < 0.0001; n = 74),
2) Reduced major axis regression (RMA):
$\delta^2H = (7.61 \pm 0.11)* \delta^{18}O + (7.58 \pm 0.68)$
(SE = 2.27; $r^2$ = 0.99; p < 0.0001; n = 74),
3) Precipitation amount weighted least squares regression (PWLSR):
$\delta^2H = (7.61 \pm 0.11)* \delta^{18}O + (7.87 \pm 0.73)$
(SE = 2.29; $r^2$ = 0.99; p < 0.0001; n = 74).
The numbers in brackets indicate the estimates of slope and intercept plus/minus the standard deviation, indicating the parameter uncertainty.

The close fit of all considered regressions indicates a very good linear relationship between $\delta^{18}O$ and $\delta^2H$ in the study area that is independent of the applied regression method. On a global scale, a good linear relationship between $\delta^{18}O$ and $\delta^2H$ is usually observed at sites where secondary fractionation processes, e.g. sub-cloud evaporation, are insignificant (Crawford et al., 2014). The LMWL for An Long is slightly different from the global meteoric water line (GMWL; defined by $\delta^2H = 8*\delta^{18}O + 10$ using OLSR, (Craig, 1961) and the LMWLs derived for six selected GNIP stations (IAEA/WMO, 2016) located in the Indochinese Peninsula (Fig. 7). The small difference in slope between these LMWLs compared to that of GMWL, and the distribution of isotope values along the GMWL indicate that evaporative isotopic enrichment during rainfall is not significant. However, the less positive intercepts of LMWLs (<10‰) (Fig. 7) may reflect smaller kinetic effects during evaporation (Ingraham, 1998) over the Mekong Delta compared to the worldwide average.

### 4.2.2 Seasonal variation and spatial homogeneity

The 74 precipitation samples at An Long showed that $\delta^{18}O$ ranges between -12.6‰ and -1.0‰, with an arithmetic mean value and standard deviation of -5.8‰ ± 2.5‰, and $\delta^2H$ ranges between -89.3‰ and 0.9‰, with an arithmetic mean value and standard deviation of -36.2‰ ± 18.7‰. Generally, less negative isotopic values are observed in the dry-season precipitation samples. The most negative values occur in the second half of the rainy season (September and October), whereas the least

negative values are observed in the late dry season in April and May (Fig. 7 and Fig. 8). This shows that the isotopic composition of precipitation at An Long station exhibits marked seasonal variations, which in turn indicates different dominant moisture sources and/or processes in the different seasons. A comparison of the seasonal variation of $\delta^{18}$O with the short-term (2014-2015) and long-term (1968-2015) monthly averages of Bangkok (Fig. 8) reveals very similar seasonality, both in terms

of timing and magnitude. The differences between $\delta^{18}$O for An Long and Bangkok are likely caused by the exceptional low rainfall in the study period compared to the long-term monthly values, particularly during May and July. Considering additionally the similarity of general factors controlling stable isotopic composition of precipitation between the two stations, i.e. annual rainfall amount, air temperature, altitude and latitude (Dansgaard, 1964;Ingraham, 1998), it can be concluded that the isotopic variations of An Long and Bangkok follow the same dynamics and controls, both on an annual and seasonal scale,

and can represent or complement each other.

In order to test the representativeness of the An Long data for a wider area, the variability of the monthly mean $\delta^{18}$O data of An Long was compared to the available GNIP data of the Indochinese Peninsula (Table 1). The Levene test (Levene, 1960) for equality of variances was used to compare the data of the different stations. As shown in Fig. 9, the test results in four distinct groups of data series with similar variances: the Northern part of the Indochinese Peninsula (Hanoi and Luang Prabang,

Fig. 9b), the Southern part of the Indochinese Peninsula (Bangkok and An Long, Fig. 9c), the islands in the Gulf of Thailand (Ko Samui and Ko Sichang, Fig. 9d), and finally Kuala Lumpur showing only little seasonal variability. The Northern and Southern parts of the Indochinese Peninsula show generally a similar seasonal behavior with a distinct higher depletion during the rainy season, but in the Northern part the highest depletion is one month earlier (August) than in the Southern part, and the magnitude of the depletion is larger. The seasonal $\delta^{18}$O variability in precipitation on the islands is much lower than on the

stations located on the continent. This is likely due to the maritime setting and could indicate a continental effect. In addition, the short-term time series of Bangkok and An Long (i.e. 2014-2015) show similar variances, resulting in a highly significant Levene test statistic of 0.98. The variation of the short-term time series of Bangkok and An Long is also very similar to the long-term time series, again shown by a highly significant Levene test statistic of 0.90 (Fig. 9c). This indicates that the isotopic variation of the An Long time series is almost identical to the one from Bangkok. In summary, the analyzed GNIP data suggests

that the data and results from this study are likely to be representative of the Southern continental part of the Indochinese Peninsula.

## 4.3 Factors controlling isotopic composition of precipitation

Prior to the MLR, the correlation of the predictors was analyzed (Table 2). The absolute values of the correlation coefficients between local (P_AL, T_AL, H_AL) and regional (P_hysplit, T_hysplit, H_hysplit, D_hysplit) climatic parameters are

relatively small and mostly not significant ($|r| < 0.4$, Table 2b). However, the correlation coefficients between regional predictors are in most cases high and significant (Table 2c). Highest correlations are found between temperature and humidity for local factors, and between the regional humidity and precipitation for regional factors. Interestingly, the correlation between

P_AL and H_AL is quite low. This indicates that the local precipitation is mainly controlled by large-scale circulation. The correlation between the predictors underlines the necessity to consider multicollinearity when investigating how the predictors control the response variables ($\delta^{18}$O and $\delta^{2}$H).

### 4.3.1 Local factors and isotopic composition in precipitation

Typically, in tropical regions subject to a monsoon climate the correlation between $\delta^{18}$O and $\delta^{2}$H values of precipitation and air temperature is virtually nonexistent, whereas a strong relation between $\delta^{18}$O and amount of precipitation has been observed (Rozanski et al., 1992;Araguás-Araguás et al., 1998). Our data show that the correlation of local precipitation amount (P_AL) and local temperature (T_AL) with isotopic values ($\delta^{18}$O and $\delta^{2}$H) are both low ($|r| < 0.45$, Table 2a). This suggests that $\delta^{18}$O and $\delta^{2}$H variation is neither dominated by local precipitation amount nor by local temperature during the sampling period. This lack of a significant correlation ($|r| < 0.5$) between $\delta^{18}$O and local rainfall amount was also observed in other regions affected by the Asian monsoon climate such as Bangkok, Hong Kong, New Delhi (Ishizaki et al., 2012), and Cherrapunji, India (Breitenbach et al., 2010). This again supports the statement that $\delta^{18}$O may not be an adequate proxy for local climatic conditions (e.g. temperature or rainfall amount) in the Asian monsoon region (Aggarwal et al., 2004;Vuille et al., 2005). Secondary fractionation processes such as sub-cloud evaporation or secondary evaporation from open water bodies tend to decrease d-excess in the residual rainwater (Stewart, 1975) and enrich it in the heavy isotopes (Guan et al., 2013). The negative correlation of humidity (H_AL) with $\delta^{18}$O and $\delta^{2}$H ($r = -0.53$, Table 2a) combined with a positive correlation with d-excess ($r = 0.2$, Table 2a), indicates that some secondary fractionation processes (Risi et al., 2008b;Crawford et al., 2017) may take place during some months at An Long. To examine in which month secondary fractionation processes are likely significant, amount-weighted mean and arithmetic mean, for both $\delta^{18}$O and d-excess are compared. The rationale is that if secondary fractionation processes are important (with the assumption that the moisture sources of different events within the month are the same), the arithmetic mean should have a $\delta^{18}$O value that is more enriched in heavy isotopes, and a much smaller d-excess than the weighted mean (Guan et al., 2013). Figure 10 shows that secondary fractionation processes may take place during the dry season, in December 2014, and in April and May 2015, because in these months a) less negative $\delta^{18}$O values and lower d-excess values compared to the overall arithmetic mean are observed, while at the same time the monthly arithmetic means are higher for $\delta^{18}$O, and lower for d-excess compared to the monthly weighted means.

To further corroborate this finding, linear regression was performed for different seasons to derive seasonal LMWL's and relations between local humidity and $\delta^{18}$O and d-excess. Table 3 suggests that secondary fractionation processes are likely to take place in the dry season between December 2014 and May 2015. This is depicted by a slope of lower than 8 (slope = 6.9) for the dry season, the slightly negative correlation between $\delta^{18}$O and local relative humidity, and the markedly positive correlation between humidity and d-excess. This is a distinctly different behavior compared to the rainy season as a whole, but also for the first (early monsoon) and second (late monsoon) parts of the rainy season. In summary, these findings indicate that secondary fractionation processes influence the isotopic composition of precipitation primarily in the dry season, which is

characterized by lower humidity and higher temperature in the Plain of Reeds. While this conclusion is plausible due to the climatic conditions and low rainfall amounts, one has to consider the low number of rainfall samples during the dry season, which associates some uncertainty to the regressions and thus the interpretation.

### 4.3.2 Regional factors and isotopic composition in precipitation

In comparison to other regional and local parameters, the precipitation amount along the transport pathways of moisture sources (P_hysplit) shows the strongest correlation with $\delta^{18}O$ and $\delta^2H$ as depicted by a correlation coefficient of -0.76 (Table 2a). Thus, P_hysplit is likely the dominant factor controlling the isotopic composition of precipitation. Other predictors show weaker correlations with $|r| < 0.55$. This, however, does not exclude that these predictors do have some predictive power for the isotopic composition of precipitation in An Long when used in combination with other predictors. Although $\delta^{18}O$ and $\delta^2H$

are rather well correlated with some climatic parameters, d-excess (which is a function of both) is not well correlated. This is because of the relative difference of the variation of $\delta^{18}O$ and d-excess, which is expressed by a low correlation coefficient between two these variables ($r = -0.44$). The weak correlation between d-excess and all climatic parameters ($|r|<0.36$) indicates that the selected predictors (i.e. selected climatic parameters) are not sufficient to explain the processes responsible for the variability of the d-excess. On a global scale, drivers controlling d-excess variation are likely sea surface temperature or near-

surface relative humidity of moisture sources (Pfahl and Wernli, 2008;Uemura et al., 2008;Pfahl and Sodemann, 2014), which are not considered in this study. In tropical areas, a major contribution to the seasonal variation in d-excess can be convective processes, e.g. re-evaporation and rain–vapor interactions (Risi et al., 2008a;Risi et al., 2010a), or the influence of large-scale processes, e.g. conditions at the vapor source, convection and recycling of moisture along trajectories (Landais et al., 2010). A complete investigation of factors controlling d-excess in precipitation is thus not possible by the presented study design and

selected predictors. However, some conclusions about the factors controlling the d-excess can be obtained with the presented method, see below.

### 4.4 MLR and relative importance analysis

The results of the MLR indicate that $\delta^{18}O$ signal in precipitation at An Long is best explained by moisture sources of 5-day backward trajectories (Fig. 11). The MLR of these trajectories produces the lowest PRESS and highest $R^2$ values, indicating

that about 80% of the variability of precipitation $\delta^{18}O$ (Fig. 11) and $\delta^2H$ (not shown) at An Long can be explained by the best MLR model. The explained variance differs only slightly between the different pressure levels used. But still, the best performance in terms of the lowest PRESS value was obtained by the mean backward trajectories of the 800 hPa and 850h Pa levels.

Contrary to $\delta^{18}O$ and $\delta^2H$, the MLR fails to explain the variation of d-excess over the whole study period to a large extent,

with a maximal $R^2$ of 0.3 (Fig. 11). This indicates that the climatic parameters used in our MLR models have only little impact on the annual d-excess variation, which corroborates the findings of the linear correlation analysis in section 4.3.2.

In the next step, the importance of the MLR predictors was analyzed. Figure 12 shows the results applying Johnson's relative weight analysis for the best performing MLR models, i.e. using the mean of the 800 hPa and 850 hPa 5-day backward trajectories. In general, the predictive power of the MLR models increases with increasing number of predictors. Both importance methods, i.e. relative weight analysis and the relative partial sum of squares, yield very similar results (not shown).

The results indicate that regional factors are always more important than local factors if the $R^2$ value is above 0.5. The local factors dominate only in MLR models with low performance, or when no regional factors are used as predictors. This is also highlighted by the sum ratio line (black line in Fig. 12), defined as the fraction of $R^2$ explained by regional factors normalized to the overall $R^2$. In the best MLR model (124[th] model) with the lowest PRESS value and an $R^2$ of 0.80, which is equivalent to an explained variance of 80%, the regional factors explain 56% of the absolute $\delta^{18}O$ variance (which is equivalent to 70%

relative to $R^2 = 0.80$), while local factors explain only 24% (30% relative to $R^2 = 0.80$). This result agrees with the two-factor analysis of Ishizaki et al. (2012) who stated that distillation during transport from source regions is the dominant contributor to inter-annual variability of $\delta^{18}O$ precipitation in Bangkok, Bombay, and Hong Kong, accounting for 70%, 60% and 70% relative to the overall explained variance, while the amount of local precipitation contributed the remaining 27%, 33%, and 25% of the explained variance, respectively.

In all models where precipitation amount along transport pathways from moisture source regions (P_hysplit) is included, this factor explains the highest proportion of $R^2$, which is always at least double and up to triple of the explained variance of other factors (Fig. 12). In turn, the absence of P_hysplit as a predictor in the MLR model considerably decreases the $R^2$, indicating that P_hysplit is the most dominant factor. In the best MLR model (124[th] model) the most important predictor is P_hysplit, explaining 47% of the total $\delta^{18}O$ variance (partial $R^2 = 0.47$, Fig. 12). The second dominant factor is T_AL, accounting for

21% of the explained the total variance. The remaining factors account for less than 13% of the $\delta^{18}O$ variance. This result indicates that the regional amount effect is a dominant process in controlling isotopic variation, whereas the local amount effect is not important in the VMD. Similar findings are reported for other regions in Asia (e.g. Rozanski et al., 1992;Araguás-Araguás et al., 1998). The local temperature T_AL, however, can be regarded as a modulating factor for the isotopic composition on top of P_hysplit.

In a next step, the predictor importance analysis is performed for different seasons, in order to analyze if seasonal differences in the dominating factors for the isotopic composition exist, as the correlation analysis of local factors and isotopic composition suggests (section 4.3.1). The samples were split into dry season and rainy season subsets, for which the MLR was applied individually. The definition of the seasons follows the analysis in section 2, i.e. the dry season lasts from December to May. However, due to the low number of samples during this period, the dry season samples were taken from mid-November to

mid-June in order to increase the sample number, thus enabling a more robust MLR fitting. This selection can be justified: Because the delineation of the dry and wet season above is based on monthly data, the "sharp" distinction between the rainy and dry season is forced by the temporal resolution of the data used. In reality, the transition between rainy and dry season is

rather gradual, thus the delineation between the rainy and dry season should rather be regarded as fuzzy. Using data from the last two weeks of November and the first two weeks of June can be seen as one way to consider this.

Furthermore, the rainy season was subdivided according to the different moisture source regions shown in section 4.1: the Indian Ocean, dominating during the initial and high stage of the Indian monsoon from June to September/mid-October, and

the South China Sea – Philippine Sea and the North-West Pacific Ocean from October to May during the late rainy and dry seasons, with some contribution from continental Asia (Fig. 5). In order to test if the factors have different importance caused by different source regions during the rainy season, the MLR models and relative importance analysis were applied for these two time periods in addition to the dry season. The number of samples for the different subsets was 42, 18 and 14 for the early rainy season, late rainy season and dry season, respectively.

Figure 13 shows the results of the MLR and importance analysis for the three seasonal subsets for $\delta^{18}$O. The sorting of the models is the same as in Fig. 12. On a first glance, the results for the rainy season subsets (Fig. 13a and Fig. 13b) are quite similar to each other and to the overall data set. The best performing model in terms of the lowest PRESS value is in all cases the model 124. However, in terms of $R^2$, the performance of the early rainy season is somewhat lower compared to the overall data set, while for the late rainy season it is significantly better, with $R^2 = 0.96$. This increase in explained variance is caused

by an increased contribution of the regional factors. In the late rainy season, the regional factors alone contribute 76% to the overall $R^2$ of 0.96 of the best PRESS model, which equals 79% of the explained variance (Table 4).

This is a much larger contribution compared to the partial $R^2$ values of 56% and 51% for the whole data set and the early rainy season, respectively. The increase stems from a larger importance of the other regional factors H_hysplit and/or T_hysplit. While their contribution to the whole data set and the early rainy period is rather low and P_hysplit dominates the contribution

of the regional factors, it is raised to about 30% in the late rainy season, either individually or in combination. For the best PRESS model marked with the cyan dot in Fig. 13b, T_hysplit contributes 27% to the overall $R^2$ of 0.96. It indicates that temperature and humidity play a larger role in the isotopic fractionation along the trajectories of water stemming from the North-West Pacific/South China Sea and continental Asia compared to water stemming from the Indian Ocean during the boreal summer months. The large regional and thus climatic heterogeneity of water sources during the late rainy season offers

a plausible explanation for this result. The source regions during this period are located in oceans and continental regions in higher latitudes outside the tropics, where large climatic differences may occur during the transport along the trajectories. Therefore, fractionation processes caused not only by the rainfall amount, but also by evaporation and condensation are likely to have a larger effect on the final isotopic composition of rainfall reaching An Long during this period, as compared to the low climatic variability of the tropical Indian Ocean region, where the rainfall during the early rainy seasons originates.

A completely different picture reveals the MLR fitting and importance analysis for the dry seasons shown in Fig. 13c. While the overall performance in terms of $R^2$ is comparable to the early rainy season, the importance of the local and regional factors is very different from the other seasons. For the dry season, the local factors dominate. In the best performing MLR model with the lowest PRESS value (cyan dot in Fig. 13c), T_AL contributes 78% of the explained variance. Similar results are

obtained for almost all of the MLR models. For the models with $R^2 > 0.5$, T_AL is the most important factor, followed by P_AL and H_AL with similar importance. The regional factors generally do not contribute more than 22% of the explained variance, if $R^2 > 0.6$. This finding corroborates the assumed higher importance of secondary fractionation processes during the dry season in the VMD, as already hypothesized in section 4.3.1. However, in combination with other predictors, T_AL seems to be a better predictor of the secondary fractionation processes compared to H_AL, which was used in 4.3.1. As T_AL and H_AL are closely correlated (Table 2), the findings of section 4.3.1 and the MLR of the dry season presented in this section agree well.

The MLR modeling of $\delta^2$H shows very similar results to $\delta^{18}$O leading to the same conclusions (Fig. S1 and Fig. S2 in the supplement). The MLR modeling of seasonal d-excess also shows an improved fit for the late rainy and dry seasons (Fig. S3 and Fig. S4 in the supplement), while for the early rainy season the results are not as satisfying as for the whole dataset. In contrast to $\delta^{18}$O and $\delta^2$H, regional factors explain the bulk of the d-excess variance also for the dry season. Among the regional factors, P_hysplit has the lowest importance for d-excess, while the others factors T_hysplit, H_hysplit, and D_hysplit explain about 65% of the best $R^2$ of 0.66. This is also a distinctively different result compared to $\delta^{18}$O and $\delta^2$H, where P_hysplit always dominated the regional factor contribution. The remaining explained variance stems mainly from the local precipitation P_AL, with some contribution of T_AL. This finding is in line with the rationale outlined in section 4.3.1, that evaporation along the transport pathway decreases the d-excess (Stewart, 1975). This effect is much more variable during the late rainy and dry season due to the transport pathways from higher latitudes, as compared to the rather uniform climatic conditions along the transport pathways during the rainy season, as already argued in the previous paragraph for the late rainy season results of $\delta^{18}$O. This means in summary that the MLR and relative importance analysis of d-excess for the late rainy and dry season corroborate the hypothesis that secondary fractionation processes caused by evaporation are relevant during the dry season, respectively for rainfall stemming from the Pacific region and continental Asia. However, for $\delta^{18}$O and $\delta^{18}$H local factors describing evaporation are more important, while for d-excess regional factors and thus evaporation processes along the transport pathways dominate.

Overall, applying all possible subset regression MLR models can much better explain the isotopic variation in rainfall compared to approaches considering only one predictor, i.e. a simple correlation analysis. Moreover, the associated relative importance analysis enables the identification of the dominant factors, thus offering interpretation aids for the identification of the processes responsible for the isotopic signature of local rainfall. The presented analysis illustrates that the investigation of dominant factors controlling isotopic composition in precipitation with simple correlation analyses may lead to wrong conclusions, particularly when predictors are correlated. Additionally, MLR is able to consider the combination of different local and regional factors, thus enabling a better identification and interpretation of the manifold processes controlling the isotopic composition of rainfall.

**5 Conclusions**

This study analyzes the influence of local and regional meteorological factors on the isotopic composition of rainfall, expressed as $\delta^{18}O$, $\delta^2H$, and d-excess, in the Vietnamese Mekong Delta (VMD). For this purpose rainfall samples were taken on a weekly to a bi-weekly basis for a period of 1.5 years at An Long in the North-Eastern part of the VMD and analyzed for stable water isotopes. The regional factors potentially influencing isotopic composition were derived by back-tracing of water particles up to 10 days from the target location using the HYSPLIT model, while the local factors were derived from local climate records. The influence of the different factors on the isotopic condition was quantified by multiple linear regression (MLR) of all factor combinations and relative importance analysis.

The MLR showed that up to 80% of the total variation of $\delta^{18}O$ can be explained by linear combinations of the selected factors. Similar results are obtained for $\delta^2H$. Contrary to this, only about 30% of the total variation of the d-excess can be explained by the selected factors, if the whole data series is used. General considerations regarding the controls of d-excess in tropical areas suggest that additional factors, like sea surface temperatures of the source region, need to be taken into account for an improved modeling of d-excess variation by MLR overall seasons and source regions.

The study showed that local climatic factors, specifically rainfall amount and temperature, play a minor role in controlling the isotopic composition of the rainfall at An Long. However, there is evidence that sub-cloud evaporation has a small effect during the dry season. Regional factors, on the contrary, dominate the isotopic composition of rainfall at An Long. 70% of the explained variance, i.e. a partial $R^2$ of up to 0.56, can be attributed to regional factors, among which precipitation amount along the transport pathway can explain most of the variance. The remaining 30% of the explained variance is attributed to local factors, among which the temperature plays the most important role. These findings indicate that local secondary fractionation processes like sub-cloud evaporation modulate the isotopic composition, which is otherwise dominated by the rainout along the transport pathway of the precipitation.

Furthermore, the analysis of transport durations implies that the moisture-producing precipitation reaching An Long travels about 4-6 days from its source, as the best regression results are obtained for these durations. For longer travel durations the explained variability of the regression decreases, suggesting that the moisture is recycled, i.e. precipitated and evaporated again, when the travel time exceeds 6 days.

If the data set is divided into seasonal subsets defined by precipitation amount and water source regions, the MLR and importance analysis enables a better identification of factors and thus processes controlling the isotopic composition in the different seasons. For the late rainy and dry seasons (i.e. October to May), the importance of regional (late rainy season) and local (dry season) factors increases compared to P_hysplit, raising the explained variance, particularly for the late rainy season. The source regions and the associated transport pathways as well as local processes are more important for these periods, indicating that secondary fractionation processes by evaporation, either along the pathway (for d-excess) or locally (for $\delta^{18}O$ and $\delta^2H$), are more important than the amount effect, which is dominant during the Indian monsoon period. This is reasonable, because moisture transported to the Mekong Delta from the Pacific region and continental Asia passes through different

climatic regimes, compared to the more uniform climatic conditions along the pathway from the Indian Ocean during the Indian summer monsoon.

In summary, it can be concluded that the proposed approach, consisting of simultaneous testing of all possible factors by MLR combined with relative importance analysis, is able to detect the relevant factors controlling the isotopic composition of rainfall

as well as their individual contributions. If applied to seasonal data subsets, the predictions can be improved and the seasonal differences in controlling factors and processes can be identified. The validity of the approach is confirmed by similar, but mainly qualitative results obtained in other studies. The comparable results provide a strong indication that the method is able to identify the dominant factors responsible for the isotopic composition of rainfall without a priori knowledge or assumptions. In contrast to previous studies, the presented approach and results provide, however, a quantitative assessment of the impact

of different factors, and thus information about the dominant processes of isotopic fractionation. It can support the interpretation of processes responsible for observed patterns of isotopic composition. The rather simple approach can, of course, not provide detailed information about atmospheric dynamics, but it provides a relatively simple and easy to apply approach supplementing or preceding more complex studies of isotopic composition with circulation models. Due to the simplicity, any scientist can easily apply this method in order to investigate factors controlling isotopic composition in

precipitation at any given study area around the world without the requirement of setting up and in-depth knowledge about running a complex numerical atmospheric circulation model. Furthermore, the approach is easily reproducible and contains a rigorous quantitative analysis of the interplay of different driving factors.  Moreover, the analysis can easily be extended to other factors and processes of importance in order to capture particularly the d-excess better, e.g. the sea surface temperatures at the source regions.

The similarity of isotopic signatures and LMWLs of stations all over Southeast Asia, as well as similar general climatic conditions, allows the conclusion that the findings are representative of a larger area. Particularly the similarity of the LMWLs, the variability of the monthly isotopic composition of rainfall, and climatic conditions of the VMD and Bangkok suggests that the results are representative for the whole Mekong Delta, and possibly for large areas of the southern tip of the continental Indochinese Peninsula.

The results have direct implications for the interpretation of paleorecords of stable water isotopes in terms of past climate conditions for Southeast Asia. Because this study shows that the factors controlling the isotopic signature of precipitation are changing between and even within seasons and that regional factors have large impacts on the local isotopic composition of rainfall. This needs to be considered in the reconstruction of past climates based on isotopic records: $\delta^{18}O$ and $\delta^2H$ values are likely to be representative for the rainfall during the dry season. However, as regional factors dominate during most of the

rainy season (receiving the bulk of the total annual rainfall), reconstructions of the past climate have to be carefully interpreted. The proposed approach might open a pathway for an improved reconstruction of paleoclimates based on isotopic records. It may e.g. be used for identifying suitable variables to improve the performance of proxy data assimilation in paleoclimate reconstruction.

**Acknowledgements**

We acknowledge the German Academic Exchange Service (DAAD) for the scholarship that enabled the first author to pursue a Ph.D. degree at the University of Potsdam, Germany. We acknowledge the International Atomic Energy Agency (IAEA) for their on-going maintenance of the Global Network of Isotopes in precipitation. The authors gratefully acknowledge the NOAA

Air Resources Laboratory (ARL) for the provision of the HYSPLIT transport and dispersion model and/or READY website (http://www.ready.noaa.gov) used in this publication. We would also like to thank the staff at An Long for isotope sample collection of precipitation. Our thanks extend to the staff who analyzed part of the isotope samples at the laboratory of the Alfred-Wegener-Institute (AWI) in Potsdam, Germany. We thank anonymous reviewers for their constructive comments to improve the manuscript.

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

**Table 1. Isotopic composition of precipitation at An Long and six selected GNIP stations throughout the Indochinese Peninsula.**

| Station Name | Location | Country | AL | P (mm) | T (°C) | Period | $\delta^{18}O$ | | $\delta^2H$ | | d-excess | LMWL (OLSR) | | LMWL (RMA) | | LMWL (PWLSR) | |
|---|---|---|---|---|---|---|---|---|---|---|---|---|---|---|---|---|---|
| | | | | | | | WM | M | WM | M | WM | S | Int. | S | Int. | S | Int. |
| An Long | 105.24°E - 10.72°N | Vietnam | 2 | 985* (1550) | 27.4* (27.2) | 06.2014 - 12.2015 | -6.4 ±1.5 | -5.8 ±2.5 | -40.9 ±11.5 | -36.2 ±18.7 | 10.4 ±1.8 | 7.56 ±0.11 | 7.26 ±0.67 | 7.61 ±0.11 | 7.58 ±0.67 | 7.61 ±0.11 | 7.87 ±0.73 |
| Hanoi ** | 105.84°E - 21.02°N | Vietnam | 10 | 1659 ±257 | 24.8 ±0.5 | 2004-2007 | -8.8 ±0.7 | -5.9 ±0.5 | -56.9 ±4.2 | -33.8 ±3.6 | 13.5 ±1.5 | 7.91 ±0.10 | 12.45 ±1.25 | 7.99 ±0.18 | 12.90 ±1.22 | 7.77 ±0.21 | 10.92 ±1.91 |
| Bangkok ** | 100.50°E - 13.73°N | Thailand | 2 | 1558 ±314 | 28.5 ±0.6 | 1968-2015 | -6.5 ±1.0 | -5.2 ±1.0 | -42.6 ±7.6 | -33.2 ±7.2 | 9.4 ±1.6 | 7.35 ±0.04 | 5.36 ±0.47 | 7.53 ±0.08 | 6.29 ±0.47 | 7.68 ±0.07 | 7.25 ±0.49 |
| Ko Samui ** | 100.03°E - 09.28°N | Thailand | 7 | 1265 ±611 | 27.9 ±0.2 | 1979-1983 | -5.8 ±1.4 | -4.8 ±0.9 | -28.8 ±7.1 | -24.1 ±4.4 | 10.8 ±0.0 | 7.18 ±0.10 | 6.89 ±1.20 | 7.30 ±0.25 | 7.41 ±1.16 | 7.45 ±0.25 | 7.89 ±1.26 |
| Ko Sichang ** | 100.80°E - 13.17°N | Thailand | 26 | 877 ±320 | 27.9 ±0.6 | 1983-1995 | -6.2 ±0.6 | -6.2 ±1.1 | -39.3 ±5.1 | -39.7 ±8.8 | 10.2 ±0.6 | 7.62 ±0.06 | 7.61 ±1.15 | 7.72 ±0.18 | 8.16 ±1.12 | 7.77 ±0.23 | 8.65 ±1.44 |
| Luang Prabang ** | 102.13°E - 19.88°N | Lao PDR | 305 | 1228 ±178 | 25.7 ±0.7 | 1961-1967 | -7.8 ±1.2 | -6.7 ±0.3 | -54.2 ±7.6 | -45.9 ±0.9 | 8.4 ±1.9 | 7.90 ±0.13 | 7.97 ±2.00 | 8.01 ±0.27 | 8.70 ±1.93 | 7.80 ±0.28 | 7.52 ±2.29 |
| Kuala Lumpur ** | 101.68°E - 03.13°N | Malaysia | 26 | 1801 ±787 | - | 1993-2012 | -7.3 ±0.8 | -7.0 ±0.7 | -46.6 ±7.7 | -45.1 ±6.8 | 11.8 ±4.1 | 7.63 ±0.07 | 8.10 ±1.93 | 8.26 ±0.26 | 12.53 ±1.92 | 7.73 ±0.29 | 8.95 ±2.24 |

Note:

* Measured at An Long in 2015; numbers in parentheses show mean values of long-term measurements at Cao Lanh.

** Data is from https://nucleus.iaea.org/wiser/gnip.php (IAEA/WMO, 2016)

P: annual precipitation (mm/year); T: average temperature (°C); AL: altitude (meter above sea level); WM: weighted mean value; M: mean value; S: slope; Int.: intercept

**Table 2. Pairwise correlation coefficients between regional factors (P_hysplit, T_hysplit, H_hysplit, D_hysplit) and local factors (P_AL, T_AL, H_AL) and stable isotopic values ($\delta^{18}O$, $\delta^2H$, and d-excess). Bold and italic numbers denote significance at the 0.01 and 0.05 level (2-tailed), respectively. The meteorological data are aggregated to weekly values corresponding to the precipitation sampling at An Long.**

| (a) | P_hysplit | H_hysplit | T_hysplit | D_hysplit | P_AL | H_AL | T_AL | Isotopic values vs. Regional and Local factors |
|---|---|---|---|---|---|---|---|---|
| $\delta^{18}O$ | **-0.74** | **-0.45** | **-0.38** | *0.24* | **-0.34** | **-0.53** | **0.45** | |
| $\delta^2H$ | **-0.76** | **-0.47** | **-0.39** | 0.20 | -0.32 | **-0.53** | **0.45** | |
| d-excess | 0.18 | 0.04 | 0.07 | **-0.36** | *0.27* | 0.20 | -0.15 | |

| (b) | P_hysplit | H_hysplit | T_hysplit | D_hysplit | Regional factors vs. Local factors | | |
|---|---|---|---|---|---|---|---|
| P_AL | 0.13 | 0.23 | 0.04 | 0.03 | | | |
| H_AL | **0.38** | 0.17 | 0.21 | 0.10 | | | |
| T_AL | -0.21 | 0.05 | 0.17 | **-0.33** | | | |

| (c) | P_hysplit | H_hysplit | T_hysplit | D_hysplit | Regional factors vs. Regional factors | | |
|---|---|---|---|---|---|---|---|
| P_hysplit | 1 | | | | | | |
| H_hysplit | **0.77** | 1 | | | | | |
| T_hysplit | **0.59** | **0.67** | 1 | | | | |
| D_hysplit | -0.10 | -0.17 | **-0.49** | 1 | | | |

| (d) | P_AL | H_AL | T_AL | Local factors vs. Local factors | | | |
|---|---|---|---|---|---|---|---|
| P_AL | 1 | | | | | | |
| H_AL | 0.20 | 1 | | | | | |
| T_AL | -0.14 | **-0.78** | 1 | | | | |

**Table 3. Results of the linear regression analysis between local relative humidity (H_AL) and isotopic values at An Long. Regressions that are statistically significant at the 0.05 level are marked in bold.**

|  | *Linear regression line* | $r$ | $R^2$ | *p-value* | $n$ | Period |
|---|---|---|---|---|---|---|
| $\delta^{18}O$ - $\delta^2H$ | $\delta^2H = 7.56*\delta^{18}O+7.26$ | **0.99** | **0.99** | **0.000** | 74 | full year |
|  | $\delta^2H = 7.62*\delta^{18}O+7.74$ | **0.99** | **0.99** | **0.000** | 67 | rainy season (Jun-Nov) |
|  | $\delta^2H = 7.58*\delta^{18}O+7.21$ | **0.99** | **0.98** | **0.000** | 42 | early monsoon (Jun-Sep) |
|  | $\delta^2H = 7.68*\delta^{18}O+8.6$ | **0.99** | **0.99** | **0.000** | 25 | late monsoon (Oct-Nov) |
|  | $\delta^2H = 6.9*\delta^{18}O+3.98$ | **0.98** | **0.96** | **0.000** | 7 | dry season (Dec-May) |
| $\delta^{18}O$ - Humidity | $\delta^{18}O = -0.51*H\_AL+36.05$ | **-0.53** | **0.28** | **0.000** | 74 | full year |
|  | $\delta^{18}O = -0.46*H\_AL+32.09$ | **-0.47** | **0.22** | **0.000** | 67 | rainy season (Jun-Nov) |
|  | $\delta^{18}O = -0.33*H\_AL+21.84$ | **-0.42** | **0.17** | **0.006** | 42 | early monsoon (Jun-Sep) |
|  | $\delta^{18}O = -0.83*H\_AL+63.12$ | **-0.61** | **0.37** | **0.001** | 25 | late monsoon (Oct-Nov) |
|  | $\delta^{18}O = -0.56*H\_AL+41.34$ | **-0.88** | **0.77** | **0.010** | 7 | dry season (Dec-May) |
| d-excess - Humidity | d-excess = 0.2*H_AL-6.36 | 0.20 | 0.04 | 0.090 | 74 | full year |
|  | d-excess = 0.13*H_AL-0.46 | 0.13 | 0.02 | 0.301 | 67 | rainy season (Jun-Nov) |
|  | d-excess = 0.18*H_AL-5.35 | 0.21 | 0.04 | 0.211 | 42 | early monsoon (Jun-Sep) |
|  | d-excess = -0.08*H_AL+17.44 | -0.07 | 0.01 | 0.734 | 25 | late monsoon (Oct-Nov) |
|  | d-excess = 0.34*H_AL-19.42 | 0.31 | 0.10 | 0.455 | 7 | dry season (Dec-May) |

**Table 4. Explained variance (partial $R^2$) of regional and local factors of the best MLR model according to the PRESS value. The first value indicates the absolute partial $R^2$, the second value the relative contribution to the overall explained variance.**

|  | Whole period | Early rainy season | Late rainy season | Dry season |
|---|---|---|---|---|
| Regional factors | 0.56 \| 70% | 0.51 \| 68% | 0.76 \| 79% | 0.14 \| 22% |
| Local factors | 0.24 \| 30% | 0.24 \| 32% | 0.20 \| 21% | 0.51 \| 78% |
| Total | 0.80 \| 100% | 0.75 \| 100% | 0.96 \| 100% | 0.65 \| 100% |

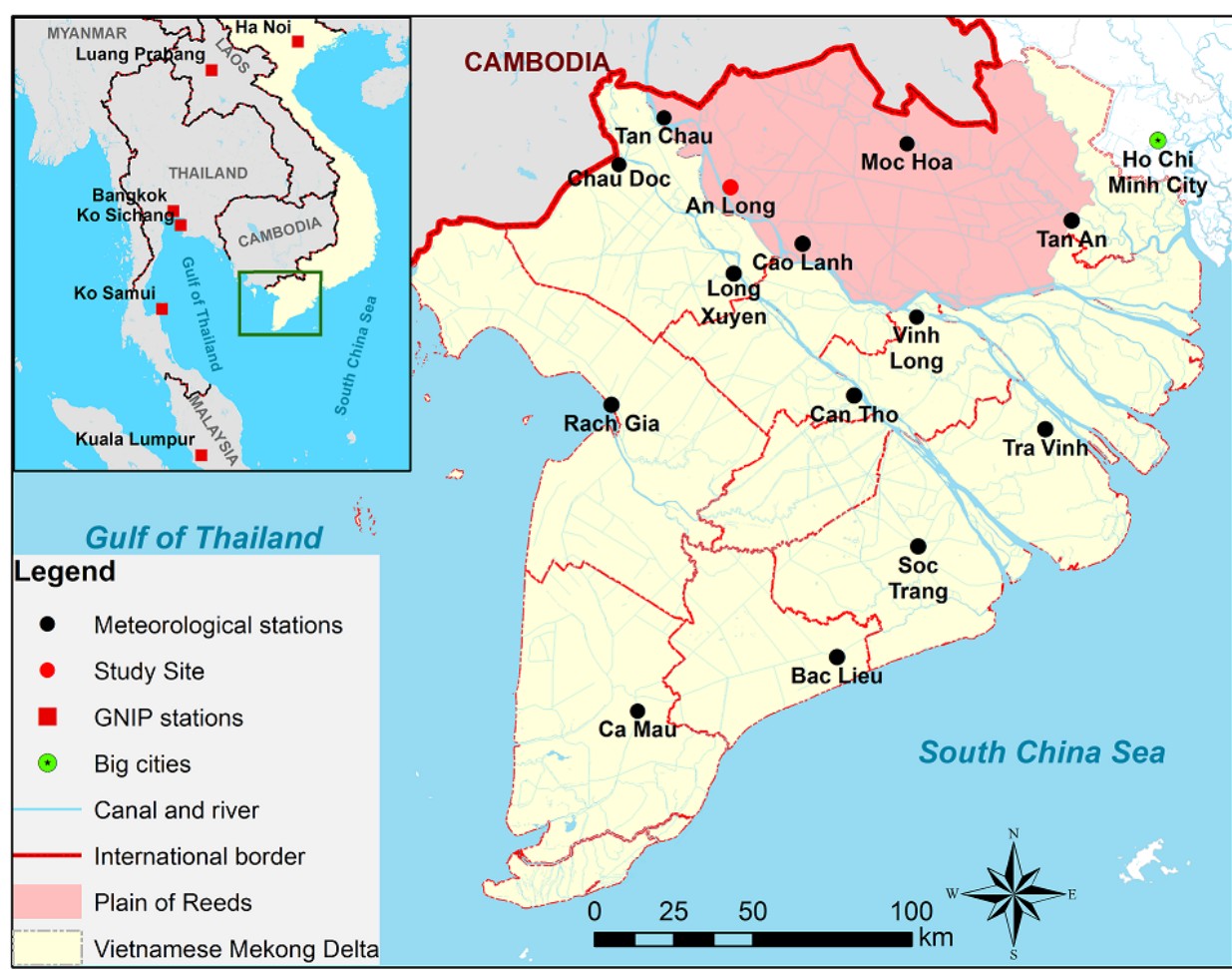

**Figure 1: Sampling and monitoring sites in the study area**

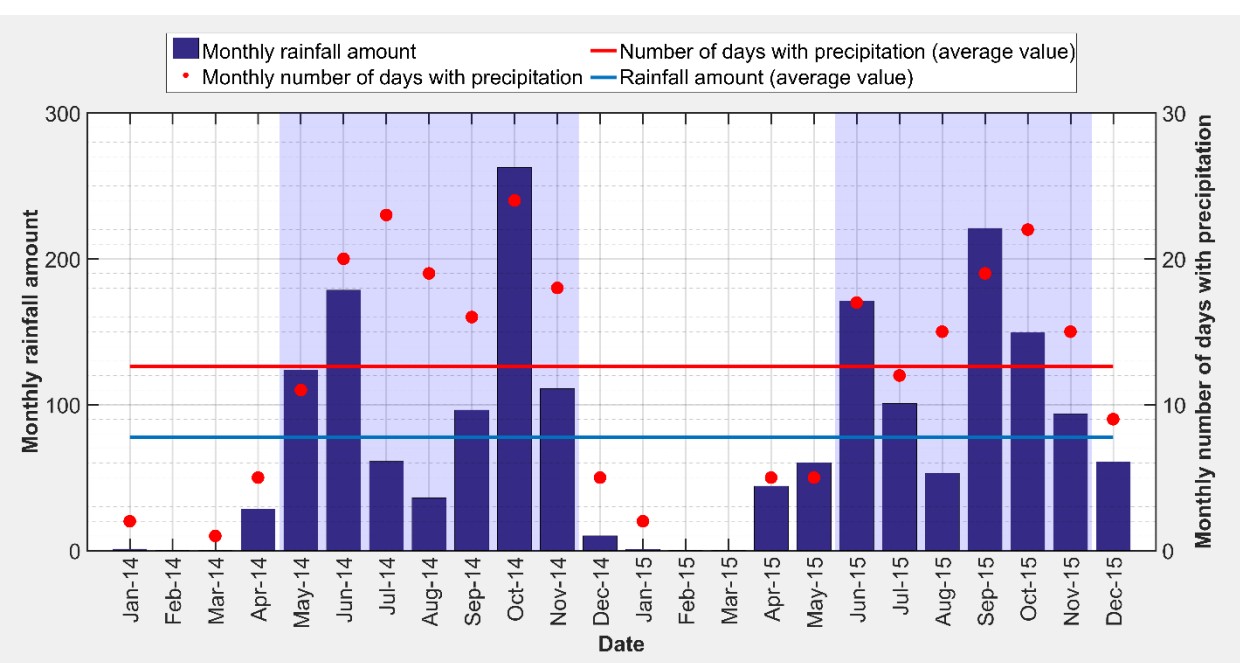

**Figure 2: Monthly precipitation (mm) and a monthly number of days with precipitation for Cao Lanh station. Light blue background indicates rainy season.**

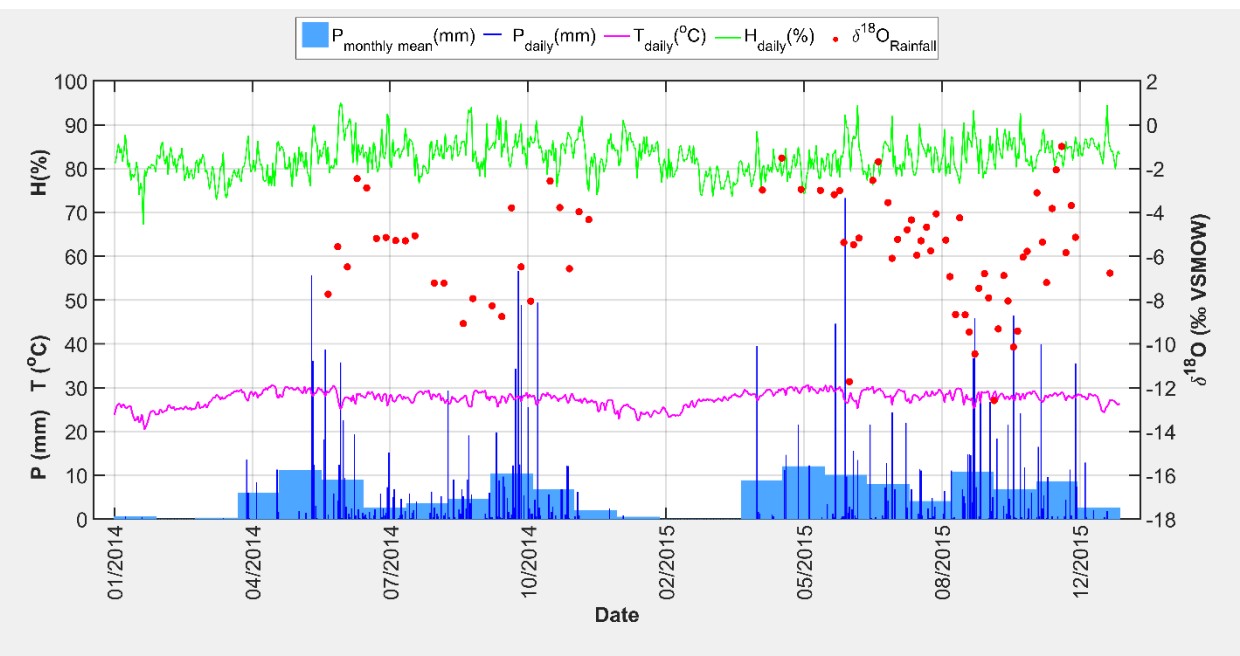

**Figure 3: Climate data from the Cao Lanh meteorological station for the study period. Daily temperature (T) is given together with monthly and daily precipitation (P) and daily relative humidity (H). Weekly and bi-weekly δ$^{18}$O (‰ VSMOW) values of rainwater are presented as red circles.**

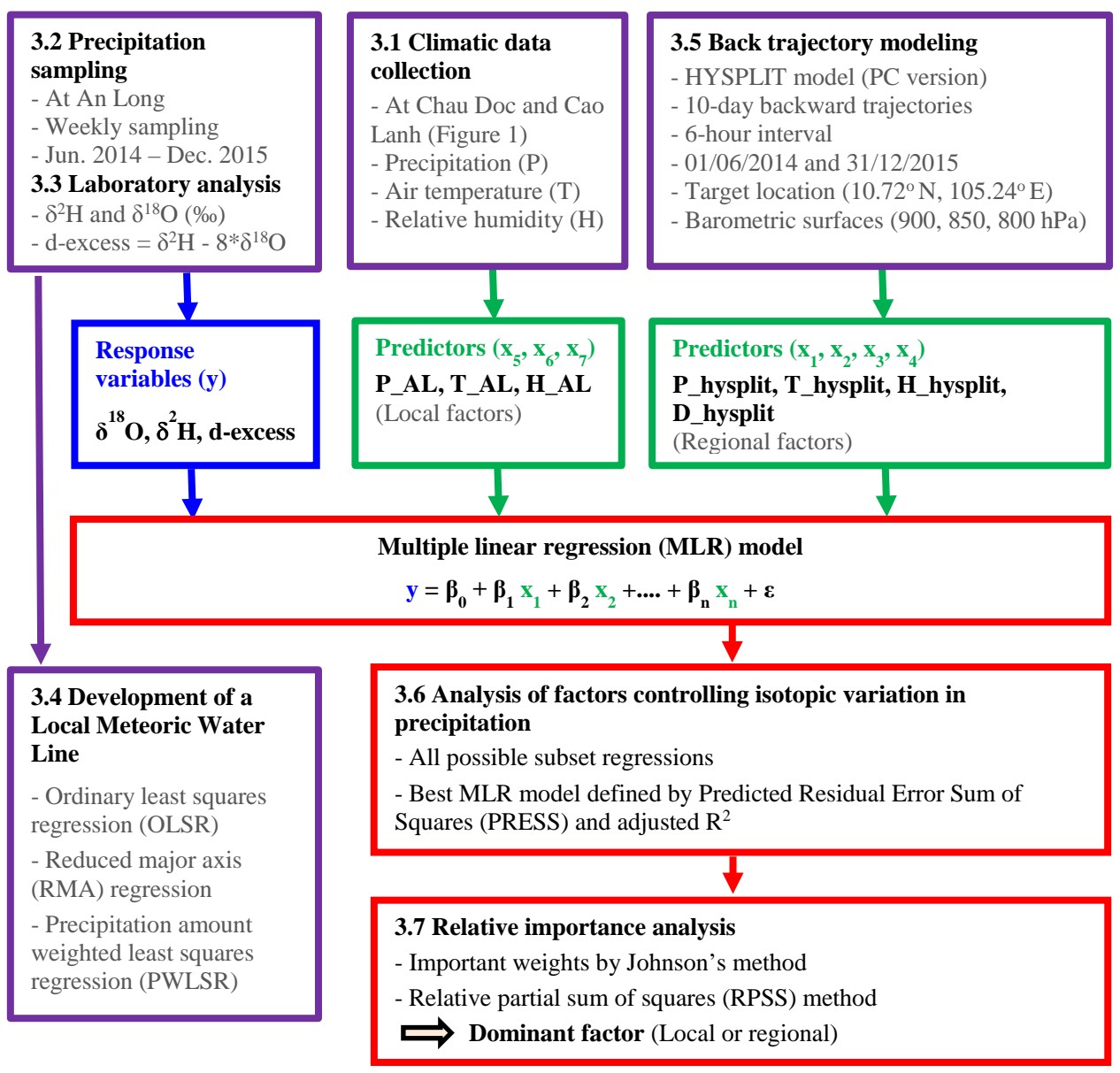

**Figure 4: Methodology used in the study. Local precipitation (P_AL), air temperature (T_AL), and relative humidity (H_AL) at An Long. Precipitation amount (P_hysplit), mean temperature (T_hysplit) and relative humidity (H_hysplit) along the transport pathways, and moving distance from moisture sources (D_hysplit).**

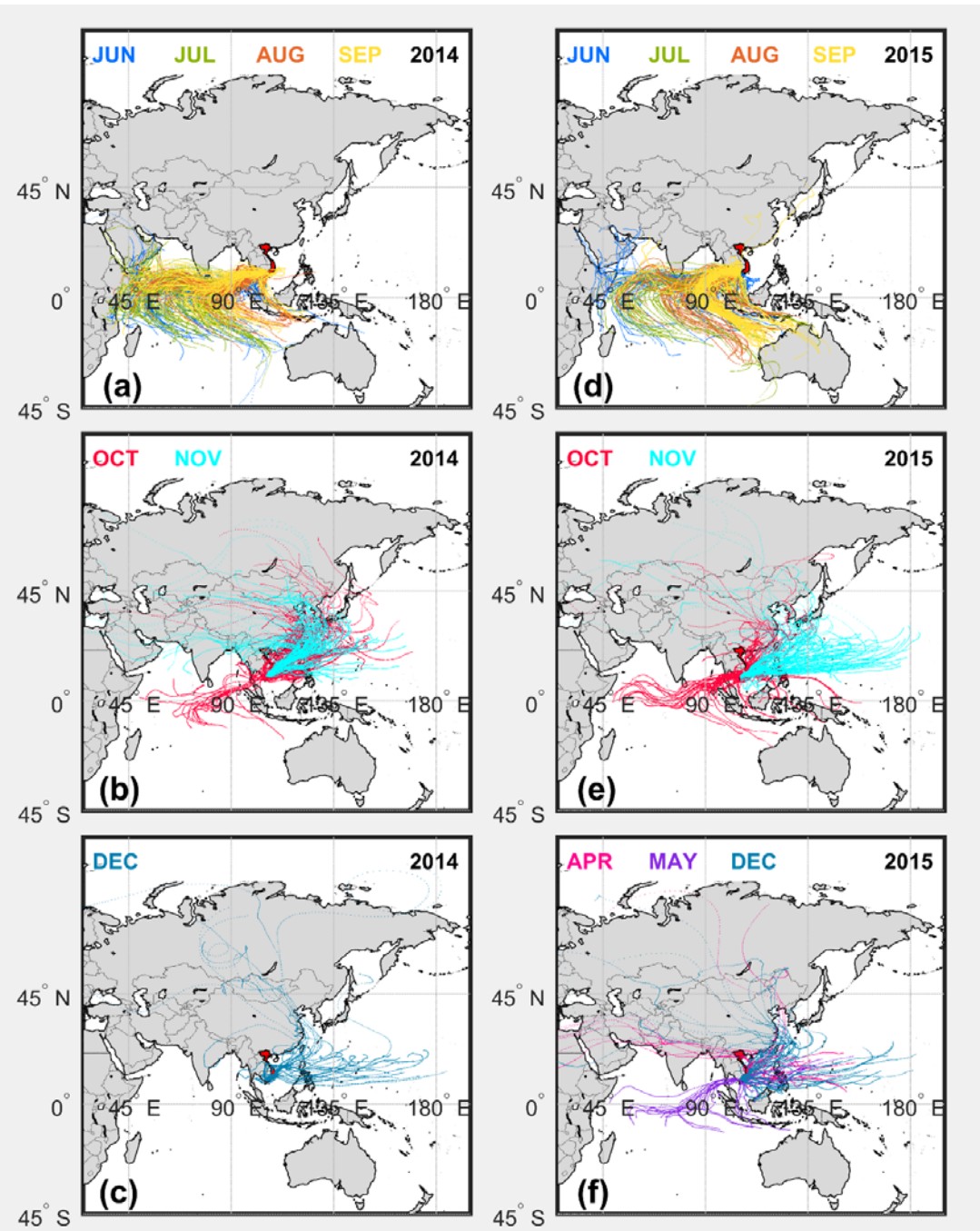

**Figure 5: Back-trajectories indicating potential moisture sources of precipitation (plotted only for days with precipitation) at An Long station for the barometric surfaces at 850 hPa between June 2014 and December 2015. Left panels show the results for 2014, right panels for 2015; top row (a, d) early rainy season (June – September), middle row (b, e) late rainy season (October – November), bottom row (c, f) dry season (December – May). In January, February and March 2015 no rainfall was recorded.**

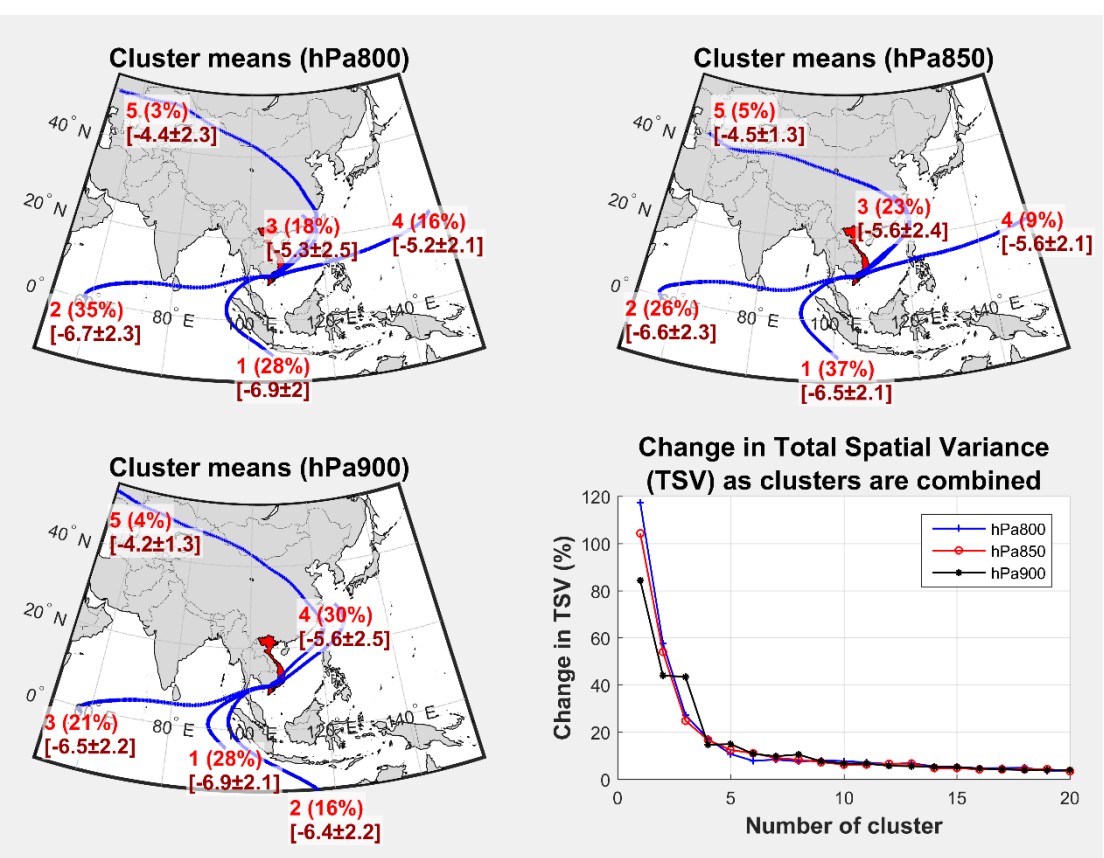

**Figure 6: Spatial distribution of vapor trajectories (cluster means) for precipitation days at An Long for 3 barometric surfaces (800, 850, 900 hPa) between June 2014 and December 2015, and change in total spatial variance (TVS) for different cluster numbers. The TSV was used to identify the optimum number of clusters (hereby 5 clusters). Red texts indicate the cluster number (1-5) and the percent of all trajectories assigned to each of the five clusters. Brown texts indicate the mean δ$^{18}$O values for each cluster plus/minus the standard deviation of each cluster.**

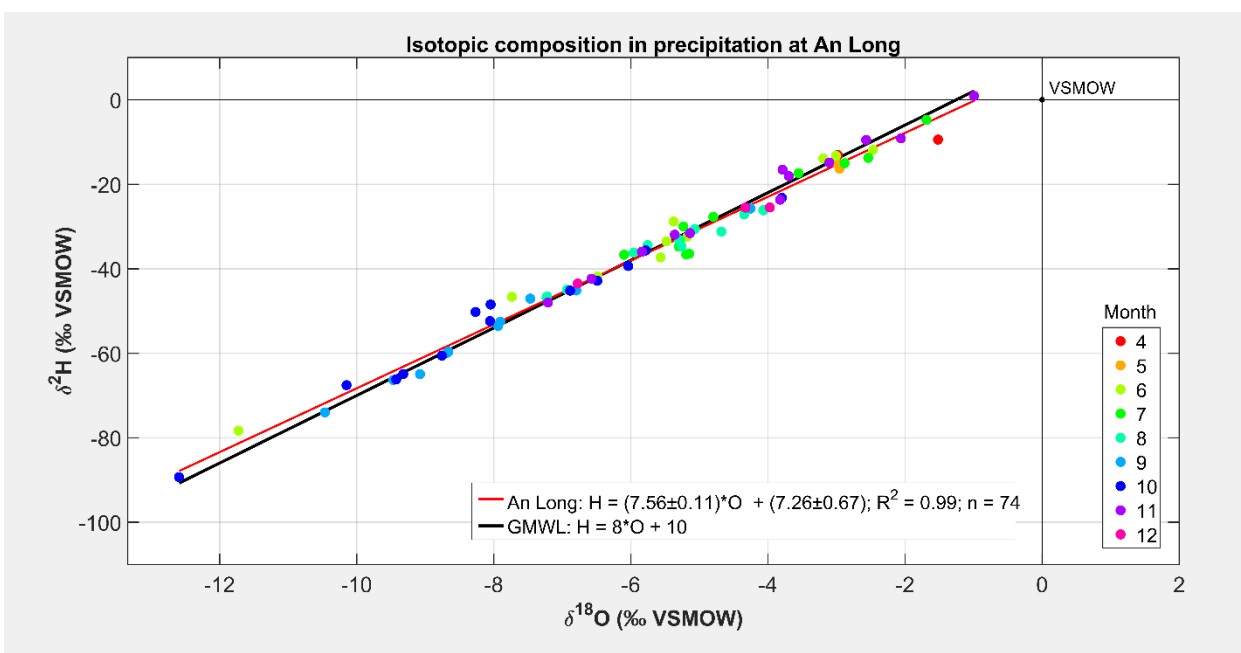

**Figure 7: The LMWL of An Long in comparison to the GMWL.**

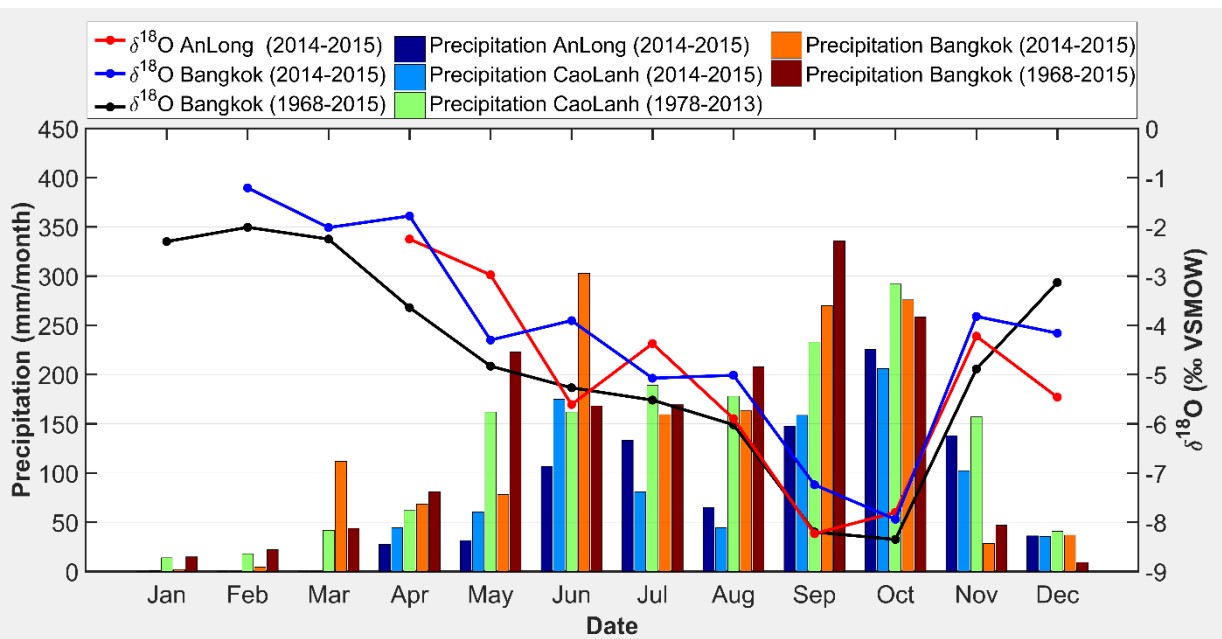

**Figure 8: Seasonal variation of the average monthly precipitation for An Long and Cao Lanh and δ¹⁸O values of precipitation for An Long (for the period of observation (red)) and Bangkok (both for the period of observation (blue) and the long-term mean (black)).**

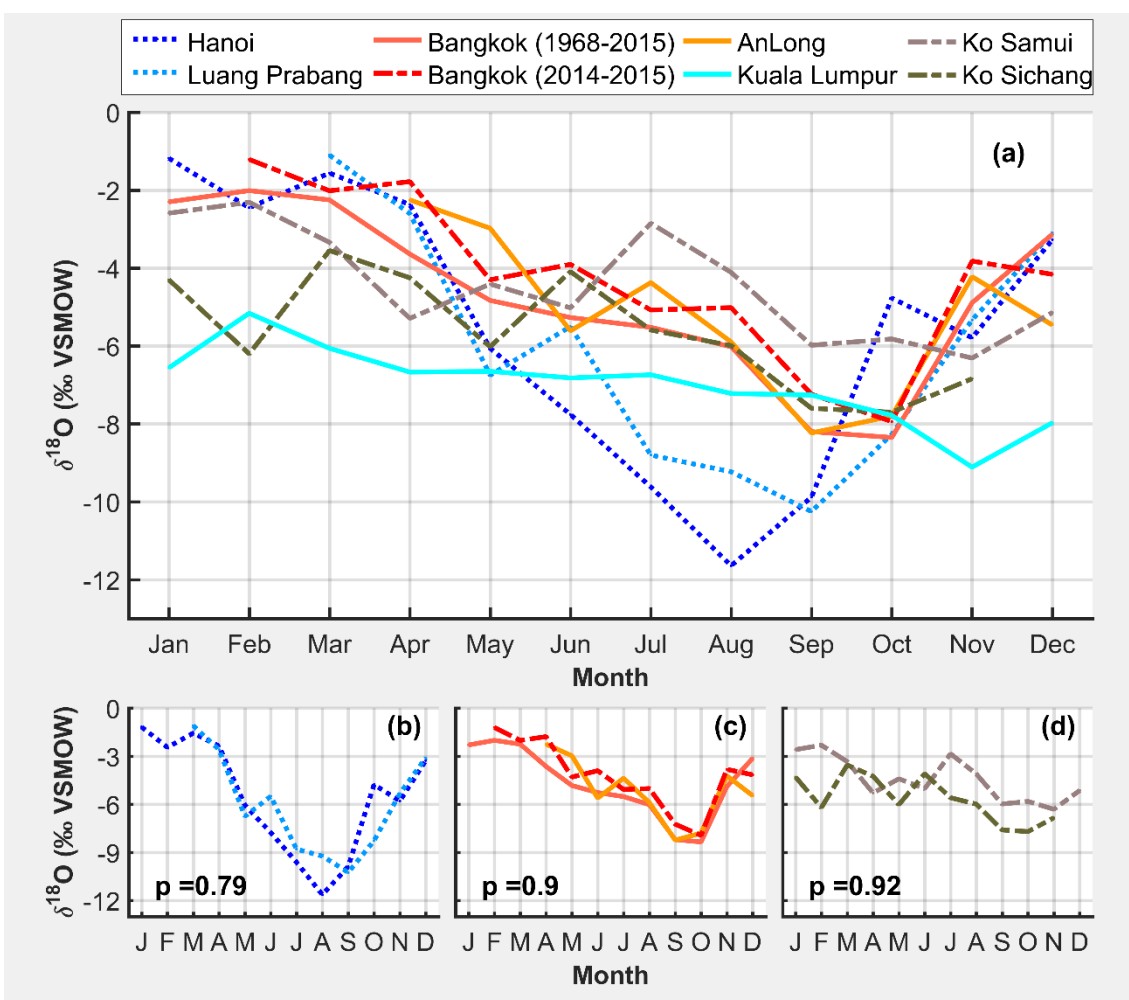

**Figure 9: Seasonal monthly mean δ¹⁸O values for An Long and GNIP data from the Indochinese Peninsula. The data is grouped according to similar variability tested with the Levene test. The p-values given in (b) to (d) are the test statistics. High values indicate similar variance. The time series of Bangkok is plotted for short-term (2014-2015) and long-term (1968-2015) periods.**

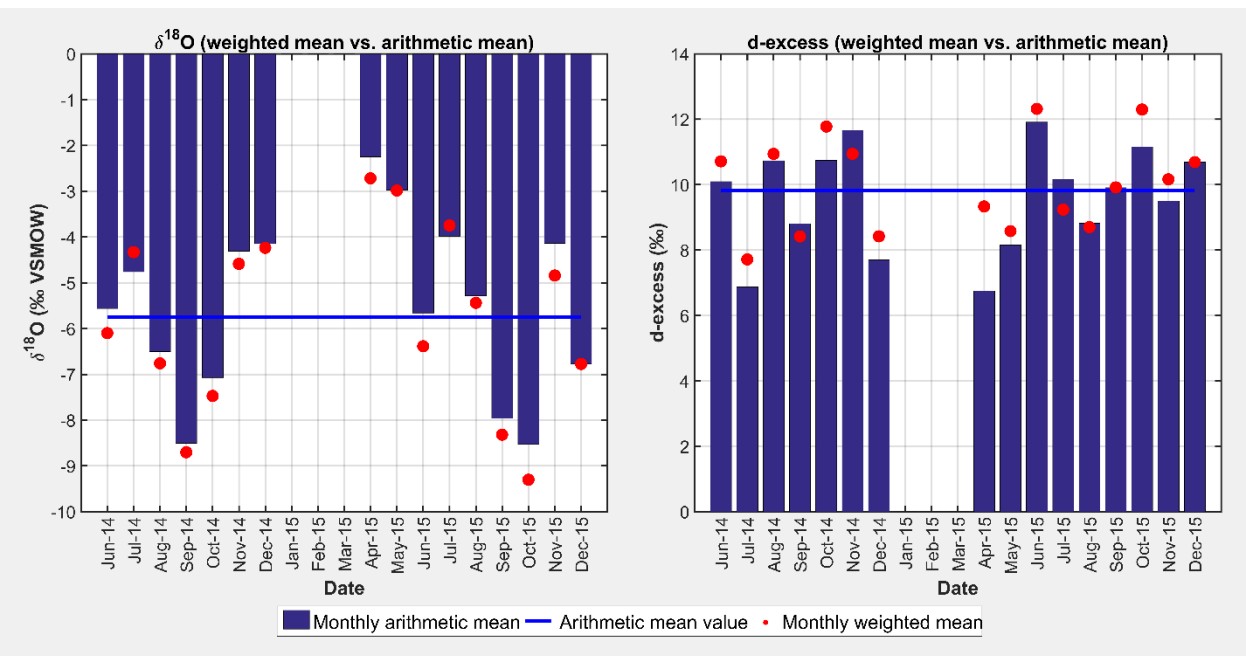

**Figure 10: Arithmetic mean and amount-weighted mean monthly δ¹⁸O (left) and d-excess (right) at An Long for the sampling period June 2014 to December 2015.**

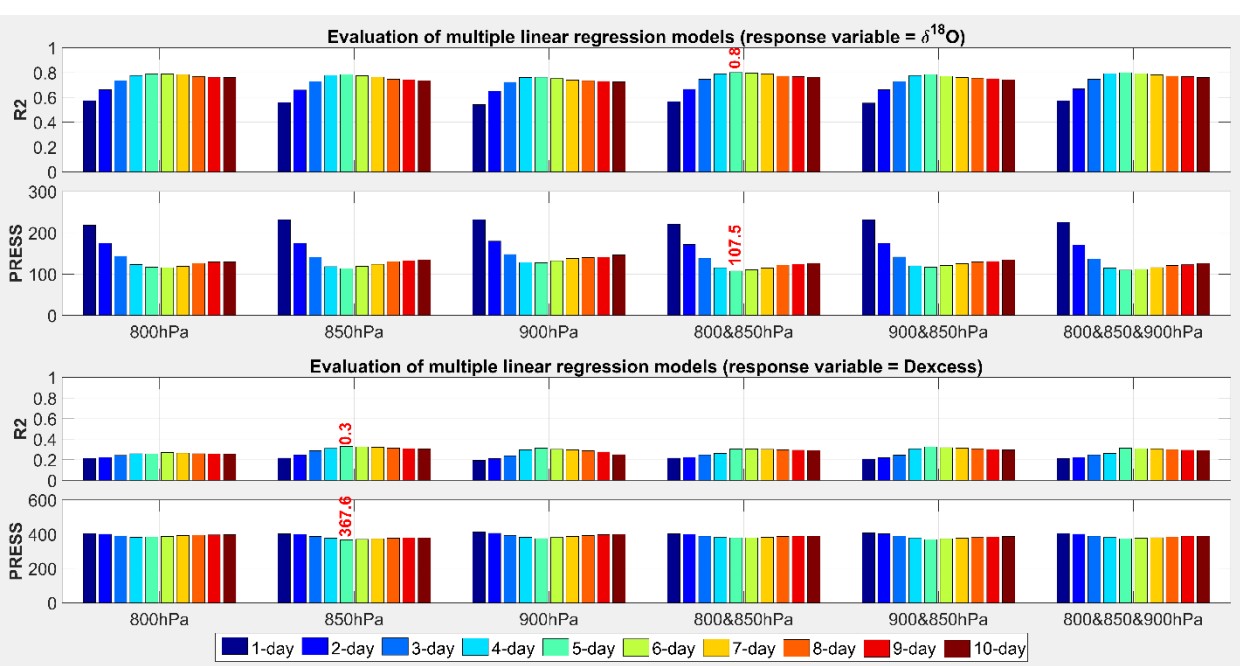

**Figure 11: Evaluation of multiple linear regression (MLR) models applied for δ¹⁸O and d-excess as response variables for different pressure levels used for three HYSPLIT backward trajectories and their combinations (mean values of the different levels). The best MLR model is highlighted with red text.**

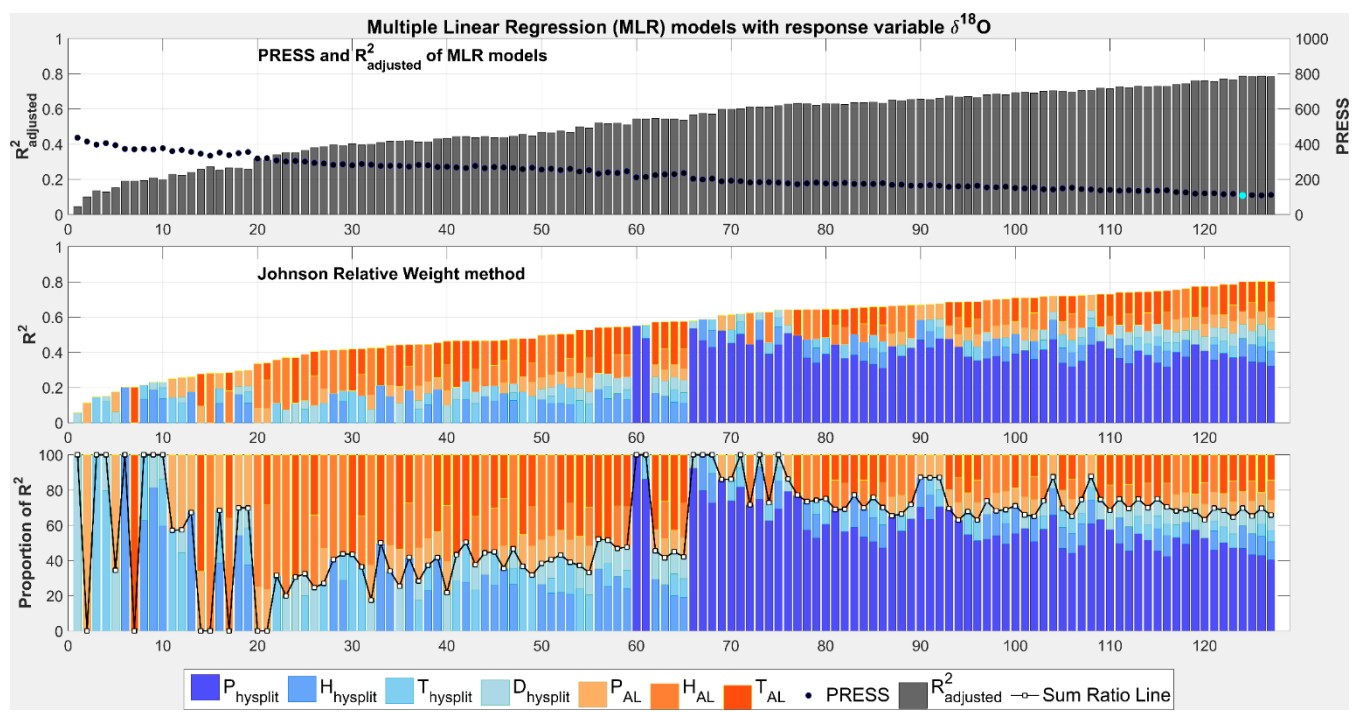

**Figure 12. MLR with response variable δ¹⁸O and relative importance analysis applied for all possible subsets. The 127 MLR models are sorted according to their R² values in ascendant order. Colors represent the relative contribution (in %) of the predictors. The sum ratio line separates the contribution of local (in red and orange) and regional (in blue) factors. PRESS and adjusted R² values indicate the quality of the MLR model. The best MLR model depicted by the lowest PRESS (model 124, highlighted by the cyan dot) explains 80% of the δ¹⁸O variation (R² = 0.8).**

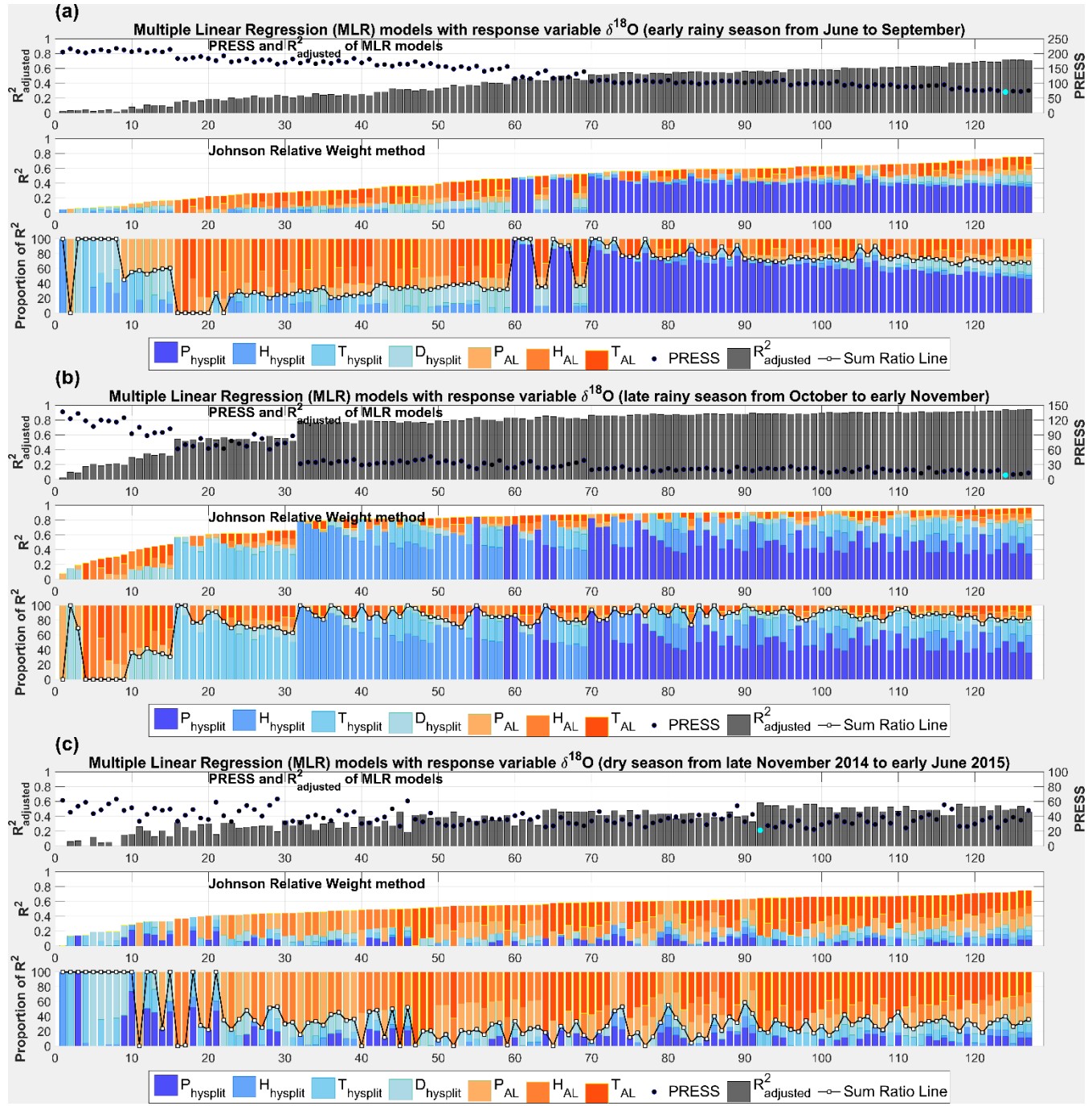

**Figure 13. MLR with response variable δ¹⁸O and relative importance analysis applied for all possible subsets (127 MLR models) for different seasons: a) early monsoon from June to September, b) late monsoon from October to mid-November, and c) the dry season from mid-November to mid-June.**