# Peer review of "What controls the stable isotope composition of precipitation in the Mekong Delta? A model-based statistical approach"

_Hydrology and Earth System Sciences, 2017_

## Referee Comment (RC1) · Anonymous Referee #1 · 20 Jun 2017

In recent years, a number of empirical, theoretical, and modeling studies have attempted to identify, characterize, and quantify the dominant controls of the stable isotopic composition of rainfall in tropics, particularly in the Asian monsoon domain. Duy et al manuscript, which at a first glance, seems like yet another manuscript along this line, indeed dives much deeper than the previous studies and attempts to provide more rigorous and quantitative assessments of various climatic factors that control stable isotope composition of rainfall in the Asian monsoon domain. Authors present a robust body of observational precipitation isotope data (weekly to bi-weekly samples over ∼1.5 years) collected from Vietnamese Mekong Delta region. This observational isotope data has been examined in the context of both local-and-regional-scale station-based climate data (temperature, precipitation amount, humidity), GNIP data, and fi-

nally climate data extracted from GDAS gridded dataset, the latter being used to drive the NOAA's HYSPLIT models. Authors conclude that the influence of the different factors on the isotopic condition is best quantified by multiple linear regressions (MLR) of all factor combinations and that explains up to 80% of the variation of $\delta$18O of precipitation. This study, like many previous studies, shows that local rainfall amount and temperature play a minor role in controlling the isotopic composition of the rainfall with upstream precipitation amount emerging as the dominant regional control— again a result consistent with previous studies, but the author's conclusion is backed by solid quantitative analysis. The manuscript is well-written, free of excessive jargon, logically structured with high-quality figures and graphics that are instructive and easy to understand. In sum, I did not find any major issues with this manuscript and I highly recommend its publication. I have provided here a few comments, which authors may find useful in further improving their manuscript. 1. Are results of this manuscript sensitive to the choice of gridded dataset (for example, R1/R2) vs GDAS, which was used to drive the HYSPLIT model? 2. Figure 5 shows backtracking trajectories (only those which produced rainfall). Perhaps I missed reading about it but can authors more clearly elaborate on the criteria they applied to establish when a certain air parcel was considered to produce rainfall? 3. Additionally, I think it will be useful to have another figure that shows major cluster tracks (instead of trajectories) and their relative weights). For example, what percentage of trajectories originate from the Indian Ocean vs continental sources during the rainy season? Furthermore, can these tracks be fingerprinted with their typical d18O values? I suppose this should not be too difficult given that authors have access to the d18O values of precipitation. 4. I think the authors need to be more specific (as opposed to providing generic comments) in suggesting how their conclusions need to be considered in paleoclimate studies. It would be helpful if they can cite some paleoclimate studies where proxy data may have been misinterpreted in light of the results obtained from this study.

[Figure]

164, 2017.

---

## Referee Comment (RC2) · Anonymous Referee #2 · 26 Jul 2017

In this paper, the authors used their new weekly precipitation isotope dataset in Vietnum's Mekong river delta region for 1.5 years, and they tried to reveal the controls of the temporal variation of the precipitation isotope ratio. To do so, they conducted some statistical analyses, and they concluded that the isotope ratio is controlled by mainly regional scale phenomena (mainly by the previous rainfall activity along the trajectory of air mass) especially during the early rainy season, and the contribution of the control varies by season.

In my opinion, even though they conducted multiple methods, nothing is quite new. The control of precipitation isotope had been discussed by many researchers as the authors mentioned, and the authors' findings were already pointed out by many, too. For example, the quantification of the controls was attempted by several model studies including

[Figure]

Yoshimura et al., 2003; Risi et al., 2008; Kurita et al., 2011; Ishizaki et al., 2012; etc. Some of these studies do not necessarily focus only Asian monsoon regions, but basically, they tried to reveal more general controls. In these studies, they used GCM or equivalent models to reveal the controls, whereas the present paper used statistical models. Furthermore, by the recent efforts, researchers already began to realize that it is indeed not appropriate to make a simple relationship between precipitation isotopes and climate parameters. The present paper's conclusion of necessity of consideration of multiple climate impacts and temporal (and spatial) dependency on the controls have been explicitly or implicitly stated many times. Therefore, nowadays, more advanced techniques of utilization of isotopic information have been utilized. One of them is data assimilation.

From the above aspect, I have to tell that this paper's methods (multiple regression and trajectory analysis) is no longer insufficient to fulfill the objectives of this study. What I mean is, there is no guarantee that this study's number of 70% regional control can be applied to any other year's temporal variation of precipitation isotopes. In this regard, 1.5-yr long data is not sufficient, too.

Major issue: 1. Drop unnecessary and unrelated analyses. Especially the parts with local meteoric line is not directly related to the conclusion of the study. It is too simple analysis. Even global meteoric line is just conceptual idea (slope of 8 and intercept of 10 is not certain). There maybe some physical reason to have smaller slope, especially by kinetic effect, but in this study, it is out discussed enough. It's better to drop the part.

2. One point data cannot represent Asian monsoon. Perhaps Mekong river delta data had some similarity with Bangkok, but with only 1.5-yr long data, the authors cannot reject possibility of "by chance". Furthermore, such similarity is nothing related to that Mekong data represent all Asian monsoon region. The title is quite misleading.

3. Organize the previous literature with focused temporal and spatial scales. The authors listed many previous studies, which partly investigated on precipitation isotope

controls, and (implicitly) stated that there is still huge discussion on the controls. However, it is misleading and not true. What is confusing is the controls can be different dependent on temporal and spatial scales. For example, daily variation of precipitation isotopes in some parts of the world is quite likely determined by synoptic-scale moisture circulation, in which previous rainfall activity along the trajectory matters a lot, rather than local precipitation or temperature, and nowadays there is consensus on this in the research community. However, even in the same place, the controls of monthly or interannual time series can be different. It is simply because those smaller scale impact can be offset each other in those scales, so that local signal only remains.

4. Limitation of statistical approach with such short-term data. The conclusion of the study is based on the statistical regression using all samples. The authors should validate their statistical model(s) with different independent samples. In this regard, the observation data is perhaps too short.

5. Most importantly, what is new in this study? As I wrote above, it is well known that precipitation isotope is not controlled by a single factor and the relationship can be different in time and space. The finding in this paper is nothing more than these.

Minor issues:

P2L17: what is "circulation effect"? Describe.

P2L23: what is difference between "distillation during vapor transport" and "upstream rainout". Aren't they essentially the same?

P2L22-P3L3: Different temporal scales are mixed.

P3L21: Before the authors' conclusion, there are many studies which state necessity of consideration of multiple parameters.

P3L27: For quantification of the controls, usually researchers try to develop a physical simulator. Any statistical model principally cannot explain the real control.

P4L20: There are many other definition of dry/wet season. What is the impact?

P5L5: "three methods" are not really regarded as different "method".

P6L4-L20: drop

P7L18: what is TRATIO?

P7L20: The uncertainty of trajectory analysis is not quantified. Perhaps it is minimized in the suggested framework, but how large is the "minimized" uncertainty and what is its potential consequence?

P8L4: PRESS is essentially the same as root mean square error (RMSE), which is more popular in the community.

P8L5: what is "leave-one-out cross validation"? and what does it mean by "equivalent to" it?

P8L16: what is physical meaning of using "mean values of their combinations"? Combination of 800hPa and 850hPa represent 825hPa level (somehow the precipitation was formed at that level at that time)? In this regard, what is meaning of 800/850/900hPa combination?

P10L4-L27: drop

P11L23-L24: I don't agree with this statement. More evidence is needed.

P14L9: Why was 124th model chosen as best?

P15L2: It is good idea. Why don't you do this trial?

---

## Author Comment (AC1) · 17 Aug 2017

**Response to 1ˢᵗ Referee's Comments on**

**What controls the stable isotope composition of precipitation in the Asian monsoon region?**

**by Le Duy Nguyen et al**

**General Comments:**

In recent years, a number of empirical, theoretical, and modeling studies have attempted to identify, characterize, and quantify the dominant controls of the stable isotopic composition of rainfall in tropics, particularly in the Asian monsoon domain. Duy et al manuscript, which at a first glance, seems like yet another manuscript along this line, indeed dives much deeper than the previous studies and attempts to provide more rigorous and quantitative assessments of various climatic factors that control stable isotope composition of rainfall in the Asian monsoon domain. Authors present a robust body of observational precipitation isotope data (weekly to bi-weekly samples over ~1.5 years) collected from Vietnamese Mekong Delta region. This observational isotope data has been examined in the context of both local-and-regional-scale station-based climate data (temperature, precipitation amount, humidity), GNIP data, and finally climate data extracted from GDAS gridded dataset, the latter being used to drive the NOAA's HYSPLIT models. Authors conclude that the influence of the different factors on the isotopic condition is best quantified by multiple linear regressions (MLR) of all factor combinations and that explains up to 80% of the variation of δ18O of precipitation. This study, like many previous studies, shows that local rainfall amount and temperature play a minor role in controlling the isotopic composition of the rainfall with upstream precipitation amount emerging as the dominant regional control again a result consistent with previous studies, but the author's conclusion is backed by solid quantitative analysis. The manuscript is well-written, free of excessive jargon, logically structured with high-quality figures and graphics that are instructive and easy to understand. In sum, I did not find any major issues with this manuscript and I highly recommend its publication. I have provided here a few comments, which authors may find useful in further improving their manuscript.

We thank the first anonymous referee for the positive and constructive comments, and the recommendation of publication of this paper. We are also grateful for the constructive comments, to which we reply below (in blue). Our answers will be included in a revised version of the manuscript.

**Specific Comments:**

1. Are results of this manuscript sensitive to the choice of gridded dataset (for example, R1/R2) vs GDAS, which was used to drive the HYSPLIT model?

Yes, we acknowledge that the results of this manuscript might be sensitive to the choice of the climate dataset driving the HYSPLIT model. Harris et al. (2005) studied the sensitivity of the trajectories to the meteorological input data (focusing on ERA-40 and NCEP/NCAR reanalysis data) and pointed out that the horizontal trajectory deviations summarized as a percentage of average distance traveled could be around 30-40% depending on the used data set. However, it is difficult to prove that in all situations a single meteorological data set was superior to another (Gebhart et al., 2005;Harris et al., 2005). Moreover, the backward-trajectory simulations by HYSPLIT are also influenced by other parameters to be defined for running HYSPLIT, such as starting time and height of the trajectories, trajectory duration, vertical motion options, and number of trajectories. Studying the sensitivity of HYSPLIT backward-trajectory simulations would be an interesting topic, but exceeds the scope of this study. More details about the uncertainty of trajectories, and a review of the types and uses for back

trajectories and the associated errors and probabilities within them has been provided by Stohl (1998), later by Fleming et al. (2012) and references therein.

With regard to the particular dataset mentioned by the reviewer (we assume that the R1/R2 dataset mentioned the 'NCEP/NCAR Reanalysis (1948 - present)' in http://ready.arl.noaa.gov/archives.php), we argue that the GDAS dataset used is more suitable for the purpose of our study. GDAS offers higher horizontal and vertical resolutions of the meteorological data (1-degree), compared to NCEP/NCAR Reanalysis (2.5 degree). Hence the HYSPLIT trajectories calculated on GDAS are supposed to be more detailed and reliable.

2. Figure 5 shows backtracking trajectories (only those which produced rainfall). Perhaps I missed reading about it but can authors more clearly elaborate on the criteria they applied to establish when a certain air parcel was considered to produce rainfall?

Because there is no daily precipitation data recorded at An Long, we used daily precipitation data at Cao Lanh instead. This is the closest national meteorological station to An Long, which is approximately 37 km SouthEast of An Long. Backtracking trajectories are plotted for the days when rainfall was recorded at Cao Lanh. This is based on assumption that days with precipitation at Cao Lanh and An Long are the same.

3. Additionally, I think it will be useful to have another figure that shows major cluster tracks (instead of trajectories) and their relative weights. For example, what percentage of trajectories originate from the Indian Ocean vs continental sources during the rainy season?

Thank you very much for this constructive suggestion. We will add the following figure to the manuscript.

[Figure]

**Figure 6: Spatial distribution of vapor trajectories (cluster means) for precipitation days at An Long for 3 barometric surfaces (800, 850, 900 hPa) between June 2014 and December 2015, and change in total spatial variance (TVS) for different cluster numbers. The TSV was used to identify the optimum number of clusters. Red texts indicate the cluster number (1-5) and the percent of all trajectories assigned to each of the five clusters. Magenta texts indicate the mean δ[18]O values for each cluster plus/minus the standard deviation of each cluster.**

The trajectory cluster analysis conducted by the HYSPLIT model groups similar trajectories. The cluster analysis merges trajectories that are near to each other and represent those clusters by their mean trajectory. Differences between trajectories within a cluster are minimized while differences between clusters are maximized. Computationally, trajectories are combined to decreasing number of clusters until the Total Spatial Variance starts to increase significantly. This occurs when disparate clusters are combined. This number of clusters is then selected as the optimal cluster number sorting similar trajectories. More information about the HYSPLIT cluster analysis can be found at: https://ready.arl.noaa.gov/documents/Tutorial/html/

Furthermore, can these tracks be fingerprinted with their typical d18O values? I suppose this should not be too difficult given that authors have access to the d18O values of precipitation.
The mean $\delta^{18}O$ values for the 5 clusters are plotted in the added figure (see above- magenta texts). The mean cluster values are similar for the three pressure levels. Also the mean values of the two clusters form the Indian Ocean, as well as the two clusters from the Pacific are similar. For a fingerprinting one also has to consider the variation of the values within the clusters, which partly overlap. This means that the $\delta^{18}O$ values of precipitation in the Mekong Delta cannot be used to uniquely identify the origin of the trajectory. However, they provide a coarse indication of the origin.

4. I think the authors need to be more specific (as opposed to providing generic comments) in suggesting how their conclusions need to be considered in paleoclimate studies. It would be helpful if they can cite some paleoclimate studies where proxy data may have been misinterpreted in light of the results obtained from this study.

We conclude that the isotopic variation of precipitation in the Asian monsoon region should not be regarded solely as being influenced by either local factors (e.g. local rainfall effect or temperature effect) or regional factors (e.g. circulation effect). Instead, it should be regarded a combination of both (Johnson and Ingram, 2004). However, to our best knowledge, there has been no study quantitatively investigates the interplay of local and regional factors in controlling isotopic composition in precipitation, which has been pointed out in this study. We will elaborate these findings in the conclusion in final version of the paper.

The suggestion of citing paleoclimate studies where our findings could have made a difference seems to be appealing, but we have to admit that paleoclimate is not our research focus and that we don't have an encompassing picture about all the past and ongoing research in this field. We thus don't feel qualified to criticize published studies in this field. We rather hope that the paleoclimate community will become aware of our results and methods, and that they might be considered in their future research.

**References**

Fleming, Z. L., Monks, P. S., and Manning, A. J.: Untangling the influence of air-mass history in interpreting observed atmospheric composition, Atmospheric Research, 104, 1-39, 2012.

Gebhart, K. A., Schichtel, B. A., and Barna, M. G.: Directional biases in back trajectories caused by model and input data, Journal of the Air & Waste Management Association, 55, 1649-1662, 2005.

Harris, J. M., Draxler, R. R., and Oltmans, S. J.: Trajectory model sensitivity to differences in input data and vertical transport method, Journal of Geophysical Research: Atmospheres, 110, 2005.

Stohl, A.: Computation, accuracy and applications of trajectories—a review and bibliography, Atmospheric Environment, 32, 947-966, 1998.

---

## Author Comment (AC2) · 22 Aug 2017

**Response to 2nd Referee's Comments on**
**What controls the stable isotope composition of precipitation in the Asian monsoon region?**
**by Le Duy Nguyen et al**

**General Comments:**
In this paper, the authors used their new weekly precipitation isotope dataset in Vietnam's Mekong river delta region for 1.5 years, and they tried to reveal the controls of the temporal variation of the precipitation isotope ratio. To do so, they conducted some statistical analyses, and they concluded that the isotope ratio is controlled by mainly regional scale phenomena (mainly by the previous rainfall activity along the trajectory of air mass) especially during the early rainy season, and the contribution of the control varies by season.
We thank the second anonymous referee for the comments, to which we reply below (in blue). Our answers will be included in a revised version of the manuscript.

In my opinion, even though they conducted multiple methods, nothing is quite new. The control of precipitation isotope had been discussed by many researchers as the authors mentioned, and the authors' findings were already pointed out by many, too. For example, the quantification of the controls was attempted by several model studies including Yoshimura et al., 2003; Risi et al., 2008; Kurita et al., 2011; Ishizaki et al., 2012; etc.
First of all, we would like to emphasize that the main objective of this study is to develop an approach to quantitatively estimate the relative contribution of regional and local factors controlling the isotopic variation of precipitation. The proposed approach is based on multiple linear regression (MLR) specifically considering the widespread issue of multicollinearity of the regression factors, in combination with a regression factor importance analysis.
We acknowledge that our methods (trajectory analysis, multiple regression and relative importance analysis) are simple and easy to apply and that each of it has already been used in previous studies. However, to our knowledge, the combination of these methods to investigate factors controlling isotopic composition in precipitation has never been applied before. In our opinion a study based on simple methodology is better than a study based on complex methods containing a larger number of uncertainty sources, if similar results are obtained (in line with the concept of parsimonious modelling). Any scientist can easily apply our method in order to investigate factors controlling isotopic composition in precipitation at any given study area around the world without the requirement of setting up and running a complex numerical circulation model. This is the novelty of this study. In the paper we already acknowledge that we don't come to new conclusion regarding the factors controlling the isotopic composition in rainfall in tropical areas, but we present a method that is universal and easy to apply (as mentioned above), and delivering a solid and reproducible quantitative analysis of the contribution of different factors.
The comment of the first anonymous referee: *"This study, like many previous studies, shows that local rainfall amount and temperature play a minor role in controlling the isotopic composition of the rainfall with upstream precipitation amount emerging as the dominant regional control again a result consistent with previous studies, but the author's conclusion is backed by solid quantitative analysis."* supports our point of view.

Moreover, this study quantitatively focuses on the interplay of various factors controlling isotopic composition in precipitation which has also never been studied before. The relative importance of these factors in controlling isotopic composition in precipitation has not been quantified as presented although first steps in this direction were taken by Ishizaki et al. (2012). However, Ishizaki et al. (2012) limited the analysis to two factors only (local precipitation amount and distillation of the moisture along its transport trajectories).

Some of these studies do not necessarily focus only Asian monsoon regions, but basically, they tried to reveal more general controls. In these studies, they used GCM or equivalent models to reveal the controls, whereas the present paper used statistical models.

Indeed we have taken climate reanalysis data derived from circulation models, extracted water transport trajectories by Lagrangian backtracing (HYSPLIT), and analyzed them with statistical models and relative importance analysis. So the presented study is not just a simple statistical data analysis. In fact, in can be seen as a substantial extension of the approach of Ishizaki et al. (2012).

Furthermore, by the recent efforts, researchers already began to realize that it is indeed not appropriate to make a simple relationship between precipitation isotopes and climate parameters. The present paper's conclusion of necessity of consideration of multiple climate impacts and temporal (and spatial) dependency on the controls have been explicitly or implicitly stated many times. Therefore, nowadays, more advanced techniques of utilization of isotopic information have been utilized. One of them is data assimilation.

As mentioned above, we acknowledge the fact that the results are not new, and that the focus of the paper is the development and testing of the combined method instead. We assume that you refer to the assimilation of data in atmospheric circulation models, which explicitly simulate the separation of water isotopes in the hydrological cycle. Of course, this would be one way to use the data and to derive information about the dominating factors for isotopic composition of rainfall. Indeed, these models could provide much more detailed information about the fractionation processes along the transport pathways of water in the atmosphere. However, the complexity of this approach is much higher compared to the one we propose, and requires in-depth knowledge about atmospheric modelling and data assimilation. In any case it would take much more effort to establish such a system if it is not already present. That means that such an approach is rather for the specialists in modelling of isotope-enabled general circulation models. For an application in a study in another field, as e.g. the mentioned paleo-climate studies, we believe that our proposed approach would be much more suitable. Besides this, even if circulation models are directly used, it is not straight forward to extract the impact of the different factors from the complex models and weight their relative importance. Some statistical procedure surely needs to be applied to come to similar conclusion as provided by our method. There are a lot of studies using isotope-enabled global climate models (GCMs) combined with some statistics to investigate the physical links between climate and water isotopes, e.g. (Vuille et al., 2005;LeGrande and Schmidt, 2009;Tindall et al., 2009;Ishizaki et al., 2012;Conroy et al., 2013). Some studies applied statistics such as principal component analysis (PCA) (Vuille et al., 2003;Curio and Scherer, 2016); machine learning technique random forests (Sánchez-Murillo et al., 2016); sensitivity experiments (Ishizaki et al., 2012) to investigate dominant factors of isotopic composition in precipitation. However, the relative importance of these parameters has not been quantitatively investigated yet (Ishizaki et al., 2012). Moreover, to our best knowledge, there is no study considering the interplay of both local and regional factors in controlling isotopic composition in precipitation, which is carefully taken into account in our study by the relative importance analysis dealing with the multicollinearity of controlling factors.

From the above aspect, I have to tell that this paper's methods (multiple regression and trajectory analysis) is no longer insufficient to fulfill the objectives of this study. What I mean is, there is no guarantee that this study's number of 70% regional control can be applied to any other year's temporal variation of precipitation isotopes. In this regard, 1.5-yr long data is not sufficient, too.

Of course, due to the limited length of the time series we cannot be 100% sure that the identified contribution of local and regional factors will be the same in other years. However, as shown in figure 7, the long term monthly isotopic values in Bangkok and the values of our two rainy seasons in the Mekong delta are quite similar. Considering also the climatic similarities between the two locations, this indicates that the recorded isotopic variation is likely to be representative

for a longer period and a wider area. This suggests in turn that the identified contribution of the factors could also be the same in other years. Also, the fact that our findings agree with the ones of Ishizaki et al. (2012) supports this assumption. Ideally one would perform a similar analysis for Bangkok for longer time series, but this is not possible due to the low resolution (monthly) of the publicly available isotope and rainfall data.

Figure 7 in the manuscript will be edited as follows to include also the short-term mean monthly isotopic signature of precipitation of Bangkok:

[Figure]

**Major issue:**
1. Drop unnecessary and unrelated analyses. Especially the parts with local meteoric line is not directly related to the conclusion of the study. It is too simple analysis. Even global meteoric line is just conceptual idea (slope of 8 and intercept of 10 is not certain). There maybe some physical reason to have smaller slope, especially by kinetic effect, but in this study, it is out discussed enough. It's better to drop the part.

You are right that the derivation of a local meteoric water line is a very simple analysis. We still think it provides valuable information for the following reasons:

- From our point of view the analysis of isotopic data in terms of meteoric water lines is a standard for such kind of data and should always be conducted, just as descriptive statistics of other data.
- Up to now, there is no LMWL for Vietnamese Mekong Delta (VMD) and Indochinese Peninsula, which can be used as a baseline for other studies using isotopic data to investigate hydrological processes in this area.
- The close fit of all considered regressions is one evidence indicating that secondary fractionation processes, e.g. sub-cloud evaporation, are insignificant in study area. This provides support for the discussion of sub-cloud evaporation in Sec. 4.3.1 in the manuscript.

2.       One point data cannot represent Asian monsoon. Perhaps Mekong river delta data had some similarity with Bangkok, but with only 1.5-yr long data, the authors cannot reject possibility of "by chance". Furthermore, such similarity is nothing related to that Mekong data represent all Asian monsoon region. The title is quite misleading.

We acknowledge that the title is too generic.  We will change it to "What controls the stable isotope composition of precipitation in the Mekong Delta?" and discuss the transferability to the greater region, i.e. SE-Asia. Actually, isotopic data of rainfall has never been collected for the

Mekong delta, and therefore the fact that the isotopic variation of the Mekong data is similar to that of Asian monsoon region has never been confirmed before.
We also went at length to illustrate that the variability of the isotopic data is similar to the long term data from Bangkok in order to provide evidence that the derived results might be representative for SE-Asia. This is already discussed in section 4.2, but we will add some critical discussion of the issue of representability in the discussion and conclusion of the revised manuscript, stating that there are indications that the obtained results could be representative for the southern part of SE-Asia.

3.     Organize the previous literature with focused temporal and spatial scales. The authors listed many previous studies, which partly investigated on precipitation isotope controls, and (implicitly) stated that there is still huge discussion on the controls. However, it is misleading and not true. What is confusing is the controls can be different dependent on temporal and spatial scales. For example, daily variation of precipitation isotopes in some parts of the world is quite likely determined by synoptic-scale moisture circulation, in which previous rainfall activity along the trajectory matters a lot, rather than local precipitation or temperature, and nowadays there is consensus on this in the research community. However, even in the same place, the controls of monthly or interannual time series can be different. It is simply because those smaller scale impact can be offset each other in those scales, so that local signal only remains.
We completely agree that scales matter. This is fundamental to hydrology. What we present is the result for daily variation (or bi-weekly, to be exact) in rainfall, in a monsoonal climate region with a strong seasonal variation. We will stress this more in the discussion and conclusion, and sort the cited literature according to the scales considered.

4.     Limitation of statistical approach with such short-term data. The conclusion of the study is based on the statistical regression using all samples. The authors should validate their statistical model(s) with different independent samples. In this regard, the observation data is perhaps too short.

As described in section 3.6, we use PRESS for selecting the best model. Within PRESS the model is fitted to all data except one, and the missing value is predicted with the fitted model, i.e. not all data is used for fitting the models at once. This procedure is repeated for every data point. Thus PRESS is equivalent to a so called leave-one-out cross validation (LOOCV), as described in section 3.6. LOOCV is the cross validation procedure appropriate for a limited data set, when a standard split sample validation cannot be applied. There are numerous papers available employing this method in different fields of environmental sciences. LOOCV is actually a split sample validation of the regression, where the data is split as often as data points are available. This means that our results are in fact validated.

5.     Most importantly, what is new in this study? As I wrote above, it is well known that precipitation isotope is not controlled by a single factor and the relationship can be different in time and space. The finding in this paper is nothing more than these.
As we have stated previously, we acknowledge the fact that our methods (trajectory analysis, multiple regression and relative importance analysis) are relatively simple and easy to apply, but we would like to stress again that the combination of these methods to investigate factors controlling isotopic composition in precipitation has never been applied before.
Moreover, our study focuses on the quantification of the impact of the various factors controlling isotopic composition in precipitation. This has not been performed in such an exhaustive way as presented here (as reviewer 1 actually points out particularly). Of course, the qualitative outcome of the study is not novel in itself, but the way we achieved these results constitutes a novel approach. Furthermore, this approach is easily reproducible and contains a rigorous analysis and quantification of the interplay of the different factors. Thus we argue that the manuscript indeed

goes beyond just stating that regional factors are more important than local factors in the daily rainfall isotopic composition for the study region. It rather supports this finding by a thorough and reproducible method that combines circulation modelling and statistical analysis.

**Minor issues:**
P2L17: what is "circulation effect"? Describe.
The term "circulation effect" (Tan, 2009;Tan, 2014) is used to describe the changes in isotopic composition in precipitation which originate from the changes in Indian/Pacific Ocean atmospheric circulation. We will add this explanation to the manuscript.

P2L23: what is difference between "distillation during vapor transport" and "upstream rainout". Aren't they essentially the same?
Yes, thank you. We will use only the term "distillation during vapor transport" in the manuscript.

P2L22-P3L3: Different temporal scales are mixed.
As mentioned above, we will sort the references according to scale.

P3L21: Before the authors' conclusion, there are many studies which state necessity of consideration of multiple parameters.
Yes, the paragraph is misleading. We will replace the whole paragraph with:
"Since it has been frequently stated and agreed to that local factors (e.g. local rainfall effect or temperature effect) and regional factors (e.g. circulation effect) should be considered simultaneously to explain the isotopic variation in rainfall (e.g. Johnson and Ingram, 2004), it can be hypothesized that using multiple factors in a single linear model is able to explain a larger share of the observed variance in isotopic composition."

P3L27: For quantification of the controls, usually researchers try to develop a physical simulator. Any statistical model principally cannot explain the real control.
Physical models are one way to address this problem. But statistical models are an alternative way to do this, and have in fact be applied many times in all sorts of environmental studies. Both approaches have their advantages and disadvantages, and they coexist, respectively supplement each other. And while statistical models are not able to represent the actual process causing a phenomenon, they are able to detect results of a process. And this is what we actually are aiming at. The statement on P3L27 expresses just this. Therefore we are arguing that the proposed approach is a) valid, and b) accepted by the majority of researchers, as long as the limitations are clearly taken into consideration. We will underline this point in more detail in the revised manuscript.

P4L20: There are many other definition of dry/wet season. What is the impact?
The definition used here is appropriate for a) the climatic condition, b) the problem to be solved and c) the data available, and is thus reasonable from our point of view. This is also supported by the reasonable results obtained for the two seasons. The impact of other definitions has not been studied in detail, but if anything would change, then some data points would be assigned to the other season. In general we do not expect any significant changes in the results, as long as the definition of the seasons is reasonable, i.e. that samples surely belonging to the dry or wet season are not assigned to the other season. Or expressed in other words: the definition of the seasons will most likely affect the samples from the transition period from one season to the other, i.e. samples that have the least explanatory value for the actual dry and wet seasons.

P5L5: "three methods" are not really regarded as different "method".
Thank you for this point. "three methods" will be changed to "three regression methods"

P6L4-L20: drop
As discussed in the 1st comment under 'major issues', we consider this part relevant and important for the manuscript.

P7L18: what is TRATIO?
We will modify the sentence from P7L17-L19 as follow:
"Secondly, we use the shortest possible integration time step (i.e. 1 h) and a small value for the parameter TRATIO (0.25), which is the fraction of a grid cell that a trajectory is permitted to transit in one advection time step. Smaller values of TRATIO help to minimize the trajectory computation error using the HYSPLIT model".

P7L20: The uncertainty of trajectory analysis is not quantified. Perhaps it is minimized in the suggested framework, but how large is the "minimized" uncertainty and what is its potential consequence?
This paragraph will be added to the manuscript to discuss about the uncertainty of trajectory analysis.
"While errors in trajectory calculation computed from analyzed wind fields seem to be typical on the order of 20% of the distance travelled (Stohl, 1998), the statistical analysis of a large number of trajectories arriving at a study site would increase the accuracy of the trajectory analysis (Cabello et al., 2008).
Harris et al. (2005) studied trajectory model sensitivity to the input meteorological data (focusing on ERA-40 and NCEP/NCAR reanalysis data) and vertical transport method. They pointed out five causes of trajectory uncertainty, expressed as percentage of deviation of the average travel distance: 1) minor differences in the computational methodology: 3–4%; 2) time interpolation: 9–25%; 3) vertical transport method: 18–34%; 4) meteorological input data: 30–40%; and 5) combined two-way differences in the vertical transport method and meteorological input data: 39–47%. However, it would be difficult to prove that in all situations a single meteorological data set or a single method of trajectory modelling was superior to another one (Gebhart et al., 2005;Harris et al., 2005). More details about the uncertainties in trajectory modelling were provided by Stohl (1998), later by Fleming et al. (2012) and references therein."

P8L4: PRESS is essentially the same as root mean square error (RMSE), which is more popular in the community.
We challenge this view. RMSE is calculated from the residuals of the model fitted to all data, while PRESS is based on the residuals resulting from the model fitted to all data except one, for which the residual is calculated. Repeating this for all data points and summing the calculated residuals results in PRESS. PRESS is therefore a cross validation method. See also our comment above, and for example the definition in WIKIPEDIA as reference (https://en.wikipedia.org/wiki/PRESS_statistic).

P8L5: what is "leave-one-out cross validation"? and what does it mean by "equivalent to" it?
See our reply to major comment 4 and the previous comment.

P8L16: what is physical meaning of using "mean values of their combinations"? Combination of 800hPa and 850hPa represent 825hPa level (somehow the precipitation was formed at that level at that time)? In this regard, what is meaning of 800/850/900hPa combination?
In P7L4-L7 we discuss that three levels at 1000, 1500, and 2000 m above ground are corresponding to barometric surfaces of approximately 900, 850, and 800 hPa. These barometric surfaces were chosen because the 850 hPa vorticity is highly indicative of the strength of the boundary layer moisture convergence and of rainfall in regions away from the equator (Wang et al., 2001). Hence rainfall is expected to mostly originate from these altitudes.

Consequently, the combination of 800 hPa and 850 hPa barometric surfaces accounts for the fact that rainfall is expected to mostly originate between 1500 and 2000 m above ground level. Similarly, the combination of the barometric surfaces of 800, 850 and 900 hPa represents that rainfall is expected to mostly originate between 1000 and 2000 m above ground level.

P10L4-L27: drop
As discussed in the 1st comment under 'major issues', we argue that this part is relevant for the manuscript.

P11L23-L24: I don't agree with this statement. More evidence is needed.
The Levene test (Levene, 1960) for equality of variances was used to compare the data of the different stations across the Indochinese Peninsula. We think that the similarity of the isotopic values and their seasonal variances between An Long and the long term time series of Bangkok (Fig. 8c) (of which the visible similarity is also confirmed with high significance by the statistical Levene test) provides sufficient evidence for our statement. In order to substantiate this finding we added the time series of Bangkok covering the same time span as our data collected in the Mekong Delta to the analysis (new figure 8c, shown below). This time series is even more similar to the one of An Long, resulting in a highly significant Levene test statistic of 0.98. This means that the isotopic variation of the An Long time series is almost identical to the one from Bangkok, and that the variation of the short term time series of Bangkok and An Long is also very similar to the long term time series. In turn, one can infer from this that the data collected in An Long are likely to be representative for the area (i.e. the southern part of SE-Asia).
However, we will modify the statement acknowledging the remaining uncertainty to: "In summary, the analyzed GNIP data suggests that the data and results from this study are likely to be representative for the Southern continental part of the Indochinese Peninsula."

The figure 8 in the manuscript will be replaced by the following figure, where the time series of Bangkok for the same period as our observation is added:

[Figure]

P14L9: Why was 124th model chosen as best?
Because the PRESS value of 124th model is smallest. The sentence provides this information. We also state this information in the methodology (P8L13).

P15L2: It is good idea. Why don't you do this trial?
We actually did this. The result are shown in Figure 12 and discussed in section 4.4 (from P15L6 to P16L8).

**References**

Cabello, M., Orza, J., Galiano, V., and Ruiz, G.: Influence of meteorological input data on backtrajectory cluster analysis? a seven-year study for southeastern Spain, Advances in Science and Research, 2, 65-70, 2008.

Conroy, J. L., Cobb, K. M., and Noone, D.: Comparison of precipitation isotope variability across the tropical Pacific in observations and SWING2 model simulations, Journal of Geophysical Research: Atmospheres, 118, 5867-5892, 2013.

Curio, J., and Scherer, D.: Seasonality and spatial variability of dynamic precipitation controls on the Tibetan Plateau, Earth System Dynamics, 7, 767, 2016.

Fleming, Z. L., Monks, P. S., and Manning, A. J.: Untangling the influence of air-mass history in interpreting observed atmospheric composition, Atmospheric Research, 104, 1-39, 2012.

Gebhart, K. A., Schichtel, B. A., and Barna, M. G.: Directional biases in back trajectories caused by model and input data, Journal of the Air & Waste Management Association, 55, 1649-1662, 2005.

Harris, J. M., Draxler, R. R., and Oltmans, S. J.: Trajectory model sensitivity to differences in input data and vertical transport method, Journal of Geophysical Research: Atmospheres, 110, 2005.

Ishizaki, Y., Yoshimura, K., Kanae, S., Kimoto, M., Kurita, N., and Oki, T.: Interannual variability of $H_2^{18}O$ in precipitation over the Asian monsoon region, Journal of Geophysical Research: Atmospheres, 117, 2012.

LeGrande, A. N., and Schmidt, G. A.: Sources of Holocene variability of oxygen isotopes in paleoclimate archives, Clim. Past, 5, 441-455, 10.5194/cp-5-441-2009, 2009.

Levene, H.: Robust tests for equality of variances, Contributions to probability and statistics, 1, 278-292, 1960.

Sánchez-Murillo, R., Birkel, C., Welsh, K., Esquivel-Hernández, G., Corrales-Salazar, J., Boll, J., Brooks, E., Roupsard, O., Sáenz-Rosales, O., and Katchan, I.: Key drivers controlling stable isotope variations in daily precipitation of Costa Rica: Caribbean Sea versus Eastern Pacific Ocean moisture sources, Quaternary Science Reviews, 131, 250-261, 2016.

Stohl, A.: Computation, accuracy and applications of trajectories—a review and bibliography, Atmospheric Environment, 32, 947-966, 1998.

Tan, M.: Circulation effect: climatic significance of the short term variability of the oxygen isotopes in stalagmites from monsoonal China—dialogue between paleoclimate records and modern climate research, Quaternary Sciences, 29, 851-862, 2009.

Tan, M.: Circulation effect: response of precipitation δ18O to the ENSO cycle in monsoon regions of China, Climate Dynamics, 42, 1067-1077, 2014.

Tindall, J., Valdes, P., and Sime, L. C.: Stable water isotopes in HadCM3: Isotopic signature of El Niño–Southern Oscillation and the tropical amount effect, Journal of Geophysical Research: Atmospheres, 114, 2009.

Vuille, M., Bradley, R., Werner, M., Healy, R., and Keimig, F.: Modeling δ18O in precipitation over the tropical Americas: 1. Interannual variability and climatic controls, Journal of Geophysical Research: Atmospheres, 108, 2003.

Vuille, M., Werner, M., Bradley, R., and Keimig, F.: Stable isotopes in precipitation in the Asian monsoon region, Journal of Geophysical Research: Atmospheres, 110, 2005.

Wang, B., Wu, R., and Lau, K.: Interannual variability of the Asian summer monsoon: contrasts between the Indian and the Western North Pacific-East Asian Monsoons*, Journal of climate, 14, 4073-4090, 2001.

---

## Author Response (AR1)

Nguyen Le Duy
Section 5.4 Hydrology
Tel.: +49 331 288 1562
Fax: +49 331 288 1570
Email: duy@gfz-potsdam.de
* * *
Helmholtz Centre Potsdam
GFZ German Research Centre For Geosciences
Telegrafenberg, 14473 Potsdam, Germany

Potsdam, 4th October, 2017

Dr. Lixin Wang
Editor
Hydrology and Earth System Sciences

**Re: HESS-2017-164**

Dear Dr. Wang,

enclosed please find a fully revised, original manuscript now titled "**What controls the stable isotope composition of precipitation in the Mekong Delta? A model-based statistical approach**", which is renamed from the previous title "What controls the stable isotope composition of precipitation in the Asian monsoon region?" (Reference #HESS-2017-164) by Nguyen Le Duy, Ingo Heidbüchel, Hanno Meyer, Bruno Merz, Heiko Apel. We are respectfully submitting our revised manuscript for your consideration in Hydrology and Earth System Sciences.

We have fully revised the paper to take into consideration the constructive comments from the two referees. Given the manuscript required major revision, we have not provided a line-by-line list of the changes since line numbers have been altered significantly. Instead, we summarize here the major revisions we have made:

- Rewriting of the abstract
- Rewriting of the introduction (focusing more specifically on highlighting the novelties and recent literature)
- Rewriting of the study area description (including a discussion about the definition of the dry/wet season)
- Rewriting of the methodology – Section 3.5 (adding a discussion about the uncertainties of trajectory analysis and applied measures to mitigate these uncertainties)
- Rewriting of the results and discussion (based on reviewers' feedback)
- Rewriting of the conclusions (focusing more specifically on highlighting the novelties).

This manuscript has neither been previously published in any language nor is it under consideration for publication by another journal. All authors have carefully read the revised manuscript and have agreed to its submission to Hydrology and Earth System Sciences. All results and innovations were developed by the authors using Matlab. Figures were generated using ArcGIS and Matlab. We also published the isotopic data in the open access data repository of GFZ. The data is already available to reviewers under:

http://pmd.gfz-potsdam.de/panmetaworks/review/9e1af507c8fce65a8d740033e5fea31c2e7c58ade81762c235c6f6bbab91166e/

Thank you for handling the manuscript during the review process, and to the reviewers for their valuable feedback and edits. We look forward to hearing from you.

Sincerely yours,
Nguyen Le Duy
Corresponding Author

**RESPONSE TO THE REFEREES' COMMENTS**

We sincerely thank both referees for their thorough reviews and most constructive comments on our manuscript (Reference #HESS-2017-164). We fully appreciate the reviewers' efforts in providing these informative reports on our research and their insights have led to an improved interpretation of our results. We have taken into full consideration all of these comments and have prepared responses to these as well as information on how the paper was revised following the referees' suggestions. Our responses to reviewers are provided below **in blue** following the individual comments requiring action from both reviewers, followed by a marked up version of the manuscript (all changes in the text are marked **in red**).

**Anonymous Referee #1**

**General Comments:**

In recent years, a number of empirical, theoretical, and modeling studies have attempted to identify, characterize, and quantify the dominant controls of the stable isotopic composition of rainfall in tropics, particularly in the Asian monsoon domain. Duy et al manuscript, which at a first glance, seems like yet another manuscript along this line, indeed dives much deeper than the previous studies and attempts to provide more rigorous and quantitative assessments of various climatic factors that control stable isotope composition of rainfall in the Asian monsoon domain. Authors present a robust body of observational precipitation isotope data (weekly to bi-weekly samples over ~1.5 years) collected from Vietnamese Mekong Delta region. This observational isotope data has been examined in the context of both local-and-regional-scale station-based climate data (temperature, precipitation amount, humidity), GNIP data, and finally climate data extracted from GDAS gridded dataset, the latter being used to drive the NOAA's HYSPLIT models. Authors conclude that the influence of the different factors on the isotopic condition is best quantified by multiple linear regressions (MLR) of all factor combinations and that explains up to 80% of the variation of δ18O of precipitation. This study, like many previous studies, shows that local rainfall amount and temperature play a minor role in controlling the isotopic composition of the rainfall with upstream precipitation amount emerging as the dominant regional control again a result consistent with previous studies, but the author's conclusion is backed by solid quantitative analysis. The manuscript is well-written, free of excessive jargon, logically structured with high-quality figures and graphics that are instructive and easy to understand. In sum, I did not find any major issues with this manuscript and I highly recommend its publication. I have provided here a few comments, which authors may find useful in further improving their manuscript.

We thank the first anonymous referee for the positive and constructive comments.

**Specific Comments:**

1. Are results of this manuscript sensitive to the choice of gridded dataset (for example, R1/R2) vs GDAS, which was used to drive the HYSPLIT model?

Yes, we acknowledge that the results of this manuscript might be sensitive to the choice of the climate dataset driving the HYSPLIT model. Moreover, the backward-trajectory simulations by HYSPLIT are also influenced by other parameters that have to be defined for running HYSPLIT, such as starting time and height of the trajectories, trajectory duration, vertical motion options, and number of trajectories. Studying the sensitivity of HYSPLIT backward-trajectory simulations would be an interesting topic, but exceeds the scope of this study.

In order to discuss the sensitivity with regard to the choice of the gridded dataset as well as the uncertainties of trajectory analysis, we included this paragraph to the revised manuscript:

"Single backward trajectory computations by the HYSPLIT model can have large uncertainties. The horizontal uncertainty of the trajectory calculations by HYSPLIT has been estimated to be 10–20 % of the travel distance (Draxler and Hess, 1998). While errors in trajectory calculation computed from analyzed wind fields seem to be typical on the order of 20% of the distance travelled (Stohl, 1998), the statistical analysis of a large number of trajectories arriving at a study site would increase the accuracy of the trajectory analysis (Cabello et al., 2008). Harris et al. (2005) studied trajectory model sensitivity to the input meteorological data (focusing on ERA-40 and NCEP/NCAR reanalysis data) and to the vertical transport method. They pointed out five causes of trajectory uncertainty, expressed as percentage of deviation of the average travel distance: 1) minor differences in the computational methodology: 3–4%; 2) time interpolation: 9–25%; 3) vertical transport method: 18–34%; 4) meteorological input data: 30–40%; and 5) combined two-way differences in the vertical transport method and meteorological input data: 39–47%. However, it would be difficult to prove that in all situations a single meteorological data set or a single method of trajectory modeling was superior to another one (Gebhart et al., 2005;Harris et al., 2005). More details about the uncertainties in trajectory modeling were provided by (Stohl, 1998), later by (Fleming et al., 2012) and references therein."

2. Figure 5 shows backtracking trajectories (only those which produced rainfall). Perhaps I missed reading about it but can authors more clearly elaborate on the criteria they applied to establish when a certain air parcel was considered to produce rainfall?

This paragraph was included to the revised manuscript (in section 4.1) to elaborate on the criteria applied to establish when a certain air parcel was considered to produce rainfall in Figure 5.

"Because there is no daily precipitation data recorded at An Long, we used daily precipitation data at Cao Lanh instead. This is the closest national meteorological station, located approximately 37 km Southeast of An Long. Backtracking trajectories in Fig. 5 are plotted for the days when rainfall was recorded at Cao Lanh. This is based on assumption that days with precipitation at Cao Lanh and An Long coincide."

3. Additionally, I think it will be useful to have another figure that shows major cluster tracks (instead of trajectories) and their relative weights. For example, what percentage of trajectories originate from the Indian Ocean vs continental sources during the rainy season?

Thank you very much for this constructive suggestion. We added Figure 6 to the manuscript.

This paragraph was also included to the revised manuscript (in section 3.5) to discuss the trajectory cluster analysis.

"The trajectory cluster analysis is conducted by the HYSPLIT model to group trajectories with similar pathways. The cluster analysis merges these trajectories that are near each other and represents those clusters by their mean trajectory. Differences between trajectories within a cluster are minimized while differences between clusters are maximized. Computationally, trajectories are combined to decrease the number of clusters until the total spatial variance (TSV) starts to increase significantly. This occurs when disparate clusters are combined. This number of clusters is then selected as the optimal cluster number for sorting and combining similar trajectories. More information about the HYSPLIT cluster analysis can be found at https://ready.arl.noaa.gov/documents/Tutorial/html/."

[Figure]

**Figure 6: Spatial distribution of vapor trajectories (cluster means) for precipitation days at An Long for 3 barometric surfaces (800, 850, 900 hPa) between June 2014 and December 2015, and change in total spatial variance (TVS) for different cluster numbers. The TSV was used to identify the optimum number of clusters. Red texts indicate the cluster number (1-5) and the percent of all trajectories assigned to each of the five clusters. Brown texts indicate the mean $\delta^{18}O$ values for each cluster plus/minus the standard deviation of each cluster.**

Furthermore, can these tracks be fingerprinted with their typical d18O values? I suppose this should not be too difficult given that authors have access to the d18O values of precipitation.

Thank you for this constructive suggestion. This paragraph was included to the revised manuscript (in section 4.1) to discuss how backward trajectories can be fingerprinted with their typical $d^{18}O$ values:

"The mean $\delta^{18}O$ values for the 5 clusters are plotted in Figure 6 (in brown). The mean cluster values are similar for the three pressure levels. Also, the mean values of the two clusters from the Indian Ocean, as well as the two clusters from the Pacific, are similar. For a fingerprinting one also has to consider the variation of the values within the clusters, which partly overlap. This means that the $\delta^{18}O$ values of precipitation in the Mekong Delta cannot be used to uniquely identify the origin of the trajectory. However, they provide a coarse indication of their origin."

4. I think the authors need to be more specific (as opposed to providing generic comments) in suggesting how their conclusions need to be considered in paleoclimate studies. It would be helpful if they can cite some paleoclimate studies where proxy data may have been misinterpreted in light of the results obtained from this study.

The suggestion of citing paleoclimate studies where our findings could have made a difference seems to be appealing, but we have to admit that paleoclimate is not our research focus and that we don't have an encompassing picture about all the past and ongoing research in this field. We thus don't feel qualified to criticize published studies in this field. We rather hope that the paleoclimate community will become aware of our results and model-based statistical approach, and that they might be considered in their future research.

**Anonymous Referee #2**

**General Comments:**

In this paper, the authors used their new weekly precipitation isotope dataset in Vietnam's Mekong river delta region for 1.5 years, and they tried to reveal the controls of the temporal variation of the precipitation isotope ratio. To do so, they conducted some statistical analyses, and they concluded that the isotope ratio is controlled by mainly regional scale phenomena (mainly by the previous rainfall activity along the trajectory of air mass) especially during the early rainy season, and the contribution of the control varies by season.

We thank the second anonymous referee for the constructive comments. Our answers are also included in the revised version of the manuscript.

In my opinion, even though they conducted multiple methods, nothing is quite new. The control of precipitation isotope had been discussed by many researchers as the authors mentioned, and the authors' findings were already pointed out by many, too. For example, the quantification of the controls was attempted by several model studies including Yoshimura et al., 2003; Risi et al., 2008; Kurita et al., 2011; Ishizaki et al., 2012; etc. Some of these studies do not necessarily focus only Asian monsoon regions, but basically, they tried to reveal more general controls. In these studies, they used GCM or equivalent models to reveal the controls, whereas the present paper used statistical models. Furthermore, by the recent efforts, researchers already began to realize that it is indeed not appropriate to make a simple relationship between precipitation isotopes and climate parameters. The present paper's conclusion of necessity of consideration of multiple climate impacts and temporal (and spatial) dependency on the controls have been explicitly or implicitly stated many times. Therefore, nowadays, more advanced techniques of utilization of isotopic information have been utilized. One of them is data assimilation.

We acknowledge the fact that the results are not new, and that the focus of the paper is the development and testing of the model-based statistical method instead. We also recognize that the title can be quite misleading (as mentioned in major issue #2), and thus may lead to a misunderstanding about the novelty of this study. We therefore modified the title to "What controls the stable isotope composition of precipitation in the Mekong Delta? A model-based statistical approach" and discussed the transferability to the greater region, i.e. SE-Asia. Actually, isotopic data of rainfall has never been collected for the Mekong delta, and therefore the fact that the isotopic variation of the Mekong data is similar to that of Asian monsoon region has never been confirmed before.

We revised the introduction and conclusion to specifically highlight the novelties of the study. Recent literature was also included accordingly. Because the revised introduction and conclusion are too long to present here, please find them in the submitted revised manuscript.

From the above aspect, I have to tell that this paper's methods (multiple regression and trajectory analysis) is no longer insufficient to fulfill the objectives of this study. What I mean is, there is no guarantee that this study's number of 70% regional control can be applied to any other year's temporal variation of precipitation isotopes. In this regard, 1.5-yr long data is not sufficient, too.

Of course, due to the limited length of the time series we cannot be 100% sure that the identified contribution of local and regional factors will be the same in other years. However, as shown in figure 7, the long term monthly isotopic values in Bangkok and the values of our two rainy seasons in the Mekong delta are quite similar. Considering also the climatic similarities between the two locations, this indicates that the recorded isotopic variation is likely to be representative for a longer period and a wider area. This suggests in turn that the identified contribution of the factors could also be the same in other years. Also, the fact that our findings agree with the ones of Ishizaki et al. (2012) supports this assumption.

Figure 7 (in the old-version of manuscript) was edited to include the short-term mean monthly

isotopic signature of precipitation of Bangkok, and renamed to Figure 8 (in the revised manuscript). The number of the other figures was edited accordingly.

[Figure]

**Figure 8:** Seasonal variation of the average monthly precipitation for An Long and Cao Lanh and δ¹⁸O values of precipitation for An Long (for the period of observation (red)) and Bangkok (both for the period of observation (blue) and the long-term mean (black)).

**Major issue:**

1. Drop unnecessary and unrelated analyses. Especially the parts with local meteoric line is not directly related to the conclusion of the study. It is too simple analysis. Even global meteoric line is just conceptual idea (slope of 8 and intercept of 10 is not certain). There maybe some physical reason to have smaller slope, especially by kinetic effect, but in this study, it is out discussed enough. It's better to drop the part.

You are right that the derivation of a local meteoric water line is a very simple analysis. We still think it provides valuable information for the following reasons:

- From our point of view the analysis of isotopic data by means of a meteoric water lines is a standard for such kind of data and should always be conducted, just as descriptive statistics of other data.
- Up to now, there is neither a LMWL for the Vietnamese Mekong Delta (VMD) nor for the Indochinese Peninsula, which could be used as a baseline for other studies using isotopic data to investigate hydrological processes in this area.
- The close fit of all considered regressions is one piece of evidence indicating that secondary fractionation processes, e.g. sub-cloud evaporation, are insignificant in the study area. This provides support for the discussion of sub-cloud evaporation in Sec. 4.3.1.

2. One point data cannot represent Asian monsoon. Perhaps Mekong river delta data had some similarity with Bangkok, but with only 1.5-yr long data, the authors cannot reject possibility of "by chance". Furthermore, such similarity is nothing related to that Mekong data represent all Asian monsoon region. The title is quite misleading.

We acknowledge that the title is too generic. We changed it to "What controls the stable isotope composition of precipitation in the Mekong Delta? A model-based statistical approach" and discussed the transferability to the greater region, i.e. SE-Asia. Actually, isotopic data of rainfall has never been collected for the Mekong delta, and therefore the fact that the isotopic variation of

the Mekong data is similar to that of Asian monsoon region has never been confirmed before.

We also went at length to illustrate that the variability of the isotopic data is similar to the long term data from Bangkok in order to provide evidence that the derived results might be representative for SE-Asia. This was already discussed in section 4.2, but we added some critical discussion of the issue of representability in the discussion and conclusion of the revised manuscript.

3.      Organize the previous literature with focused temporal and spatial scales. The authors listed many previous studies, which partly investigated on precipitation isotope controls, and (implicitly) stated that there is still huge discussion on the controls. However, it is misleading and not true. What is confusing is the controls can be different dependent on temporal and spatial scales. For example, daily variation of precipitation isotopes in some parts of the world is quite likely determined by synoptic-scale moisture circulation, in which previous rainfall activity along the trajectory matters a lot, rather than local precipitation or temperature, and nowadays there is consensus on this in the research community. However, even in the same place, the controls of monthly or interannual time series can be different. It is simply because those smaller scale impact can be offset each other in those scales, so that local signal only remains.

We completely agree that scales matter. This is fundamental to hydrology. What we present is the result for daily variation (or bi-weekly, to be exact) in rainfall, in a monsoonal climate region with a strong seasonal variation. We stressed this more in the discussion and conclusion, and sorted the cited literature according to the scales considered.

4.      Limitation of statistical approach with such short-term data. The conclusion of the study is based on the statistical regression using all samples. The authors should validate their statistical model(s) with different independent samples. In this regard, the observation data is perhaps too short.

As described in section 3.6, we use PRESS for selecting the best model. Within PRESS the model is fitted to all data except one, and the missing value is predicted with the fitted model, i.e. not all data is used for fitting the models at once. This procedure is repeated for every data point. Thus PRESS is equivalent to a so called leave-one-out cross validation (LOOCV), as described in section 3.6. LOOCV is the cross validation procedure appropriate for a limited data set, when a standard split sample validation cannot be applied. There are numerous papers available employing this method in different fields of environmental sciences. LOOCV is actually a split sample validation of the regression, where the data is split as often as data points are available. This means that our results are in fact validated.

5.      Most importantly, what is new in this study? As I wrote above, it is well known that precipitation isotope is not controlled by a single factor and the relationship can be different in time and space. The finding in this paper is nothing more than these.

We revised the introduction and conclusion to highlight more specific the novelties of this study. As we have stated previously, we acknowledge the fact that our methods (trajectory analysis, multiple linear regression and relative importance analysis) are relatively simple and easy to apply, but we would like to stress again that the combination of these methods to investigate factors controlling isotopic composition in precipitation has never been applied before.

Moreover, our study focuses on the quantification of the impact of the various factors controlling isotopic composition in precipitation. This has not been performed in such an exhaustive way as presented here (as reviewer 1 actually points out particularly). Of course, the qualitative outcome of the study is not novel in itself, but the way we achieved these results constitutes a novel approach. Furthermore, this approach is easily reproducible and contains a rigorous analysis and

quantification of the interplay of the different factors. Thus we argue that the manuscript indeed goes beyond just stating that regional factors are more important than local factors for the daily rainfall isotopic composition of the study region. It rather supports this finding by a thorough and reproducible method that combines trajectory modelling and statistical analysis.

In order to stress the novelty of this study, we also included this paragraph to the conclusion:

"The validity of the approach is confirmed by similar, but mainly qualitative results obtained in other studies. The comparable results provide a strong indication that the method is able to identify the dominant factors responsible for the isotopic composition of rainfall without a priori knowledge or assumptions. In contrast to previous studies, the presented approach and results provide, however, a quantitative assessment of the impact of different factors, and thus information about the dominant processes of isotopic fractionation. It can support the interpretation of processes responsible for observed patterns of isotopic composition. The rather simple approach can, of course, not provide detailed information about atmospheric dynamics, but it provides a relatively simple and easy to apply approach supplementing or preceding more complex studies of isotopic composition with circulation models. Due to the simplicity, any scientist can easily apply this method in order to investigate factors controlling isotopic composition in precipitation at any given study area around the world without the requirement of setting up and in-depth knowledge about running a complex numerical atmospheric circulation model. Furthermore, the approach is easily reproducible and contains a rigorous quantitative analysis of the interplay of different driving factors. Moreover, the analysis can easily be extended to other factors and processes of importance in order to capture particularly the d-excess better, e.g. the sea surface temperatures at the source regions."

**Minor issues:**

P2L17: what is "circulation effect"? Describe.

The term "circulation effect" (Tan, 2009;Tan, 2014) is used to describe the changes in isotopic composition in precipitation that appear because arriving moisture is coming from different areas of the ocean. The revised manuscript now includes this explanation.

P2L23: what is difference between "distillation during vapor transport" and "upstream rainout". Aren't they essentially the same?

Yes, thank you for pointing this out. We used only the term "distillation during vapor transport" in the revised manuscript.

P2L22-P3L3: Different temporal scales are mixed.

We sorted the references according to scale. The paragraph was revised as follows:

"Recently, many studies have presented evidence that large-scale monsoon circulation is the primary driver of variations in precipitation isotopes instead of local controls (e.g. local precipitation amount or temperature) in some parts of the Asian monsoon region. This evidence has been found at different temporal scales including daily isotopic variability (Yoshimura et al., 2003;Yoshimura et al., 2008), seasonal isotopic variability (Araguás-Araguás et al., 1998;Kurita et al., 2009;Dayem et al., 2010;Peng et al., 2010;Baker et al., 2015), and/or interannual isotopic variability (Vuille et al., 2005;LeGrande and Schmidt, 2009;Ishizaki et al., 2012;Tan, 2014;Kurita et al., 2015)."

P3L21: Before the authors' conclusion, there are many studies which state necessity of consideration of multiple parameters.

Yes, the paragraph is misleading. We replaced the whole paragraph in the introduction with:

"It has been frequently stated and agreed to that local and regional factors should be considered simultaneously to explain the isotopic variation in rainfall (e.g. Johnson and Ingram, 2004). Hence, it can be hypothesized that using multiple factors in a single linear model is able to explain a larger share of the observed variance in isotopic composition. We aim at developing and testing a model-based statistical approach for the quantification of the contribution of isotopic separation processes for explaining the isotopic variation of precipitation. Such a model-based statistical method could also be applied in paleoclimate studies, separating and quantifying the impacts of local and regional factors on the isotopic composition of local precipitation (Sturm et al., 2010), thus overcoming the shortcomings of single factor analysis."

P3L27: For quantification of the controls, usually researchers try to develop a physical simulator. Any statistical model principally cannot explain the real control.

Physical models are one way to address this problem. But statistical models are an alternative way and have in fact be applied many times in all sorts of environmental studies. Both approaches have their advantages and disadvantages, and they coexist, respectively supplement each other. And while statistical models are not able to represent the actual process causing a phenomenon, they are able to detect results of a process. And this is what we actually are aiming at. Therefore we are arguing that the proposed model-based statistical approach is valid and accepted by the majority of researchers, as long as the limitations are clearly taken into consideration. We underlined this point in more detail in the introduction (P3L3-P4L33) of revised manuscript.

P4L20: There are many other definition of dry/wet season. What is the impact?

We included this paragraph in the revised manuscript (Section 2. Study area) to discuss the impact of the definition of dry/wet season.

"The definition used here is particularly developed for the local climatic conditions, the problem to be solved, and the data available. Other definitions could cause some data points to be assigned to the other season. However, those data points will most likely be from the transition period from one season to the other, i.e. other definitions would affect samples that have the least explanatory value for the actual dry and wet seasons."

P5L5: "three methods" are not really regarded as different "method".

Thank you for this point. "three methods" was changed to "three regression methods"

P6L4-L20: drop

As discussed in the 1st comment under 'major issues', we consider this part relevant and important for the manuscript. Therefore we would like to keep it.

P7L18: what is TRATIO?

We modified the sentence from P7L17-L19 in the old version of the manuscript as follows:

"Secondly, we use the shortest possible integration time step (i.e. 1 h) and a small value for the parameter TRATIO (0.25), which is the fraction of a grid cell that a trajectory is permitted to transit in one advection time step. Smaller values of TRATIO help to minimize the trajectory computation error using the HYSPLIT model".

P7L20: The uncertainty of trajectory analysis is not quantified. Perhaps it is minimized in the suggested framework, but how large is the "minimized" uncertainty and what is its potential consequence?

In order to discuss the sensitivity with regard to the choice of the gridded dataset as well as the uncertainties of the trajectory analysis, we included this paragraph to the methodology (Section 3.5) in the revised manuscript:

"Single backward trajectory computations by the HYSPLIT model can have large uncertainties. The horizontal uncertainty of the trajectory calculations by HYSPLIT has been estimated to be 10–20 % of the travel distance (Draxler and Hess, 1998). While errors in trajectory calculation computed from analyzed wind fields seem to be typical on the order of 20% of the distance travelled (Stohl, 1998), the statistical analysis of a large number of trajectories arriving at a study site would increase the accuracy of the trajectory analysis (Cabello et al., 2008). Harris et al. (2005) studied trajectory model sensitivity to the input meteorological data (focusing on ERA-40 and NCEP/NCAR reanalysis data) and to the vertical transport method. They pointed out five causes of trajectory uncertainty, expressed as percentage of deviation of the average travel distance: 1) minor differences in the computational methodology: 3–4%; 2) time interpolation: 9–25%; 3) vertical transport method: 18–34%; 4) meteorological input data: 30–40%; and 5) combined two-way differences in the vertical transport method and meteorological input data: 39–47%. However, it would be difficult to prove that in all situations a single meteorological data set or a single method of trajectory modeling was superior to another one (Gebhart et al., 2005;Harris et al., 2005). More details about the uncertainties in trajectory modeling were provided by (Stohl, 1998), later by (Fleming et al., 2012) and references therein."

P8L4: PRESS is essentially the same as root mean square error (RMSE), which is more popular in the community.

RMSE is calculated from the residuals of the model fitted to all data, while PRESS is based on the residuals resulting from the model fitted to all data except one, for which the residual is calculated. Repeating this for all data points and summing the calculated residuals results in PRESS. PRESS is therefore a cross validation method. See also our comment above, and for example the definition in WIKIPEDIA as reference (https://en.wikipedia.org/wiki/PRESS_statistic).

P8L5: what is "leave-one-out cross validation"? and what does it mean by "equivalent to" it?

See our reply to major comment 4 and to the previous comment.

P8L16: what is physical meaning of using "mean values of their combinations"? Combination of 800hPa and 850hPa represent 825hPa level (somehow the precipitation was formed at that level at that time)? In this regard, what is meaning of 800/850/900hPa combination?

In P7L4-L7 we discuss that the three levels at 1000, 1500, and 2000 m above ground are corresponding to barometric surfaces of approximately 900, 850, and 800 hPa. These barometric surfaces were chosen because the 850 hPa vorticity is highly indicative of the strength of the boundary layer moisture convergence and of rainfall in regions away from the equator (Wang et al., 2001). Hence rainfall is expected to mostly originate from these altitudes. We included this paragraph in the revised manuscript (in section 3.5) to elaborate on the physical meaning of using "mean values of their combinations" as follows:

"Consequently, the combination of 800 hPa and 850 hPa barometric surfaces accounts for the fact that rainfall is expected to mostly originate between 1500 and 2000 m above ground level. Correspondingly, the combination of the barometric surfaces of 800, 850 and 900 hPa means that rainfall is expected to mostly originate between 1000 and 2000 m above ground level."

P10L4-L27: drop

As discussed in the 1st comment under 'major issues', we argue that this part is relevant for the manuscript. We would like to keep it.

P11L23-L24: I don't agree with this statement. More evidence is needed.

The Levene test (Levene, 1960) for equality of variances was used to compare the data of the different stations across the Indochinese Peninsula. We argue that the observed similarity of the isotopic values and their seasonal variances between An Long and the long term time series of Bangkok (Fig. 8c) (of which the visible similarity is also confirmed with high significance by the statistical Levene test) provides sufficient evidence for our statement. In order to substantiate this finding we added the time series of Bangkok covering the same time span as our data collected in the Mekong Delta to the analysis (new figure 8c, shown below). This time series is even more similar to the one of An Long, resulting in a highly significant Levene test statistic of 0.98. This means that the isotopic variation of the An Long time series is almost identical to the one from Bangkok, and that the variation of the short term time series of Bangkok and An Long is also very similar to the long term time series. In turn, one can infer from this that the data collected in An Long are likely to be representative for the area (i.e. the southern part of SE-Asia). This evidence was included in the revised manuscript (Section 4.2.2) as follows:

"In addition, the short-term time series of Bangkok and An Long (i.e. 2014-2015) show similar variances, resulting in a highly significant Levene test statistic of 0.98. The variation of the short-term time series of Bangkok and An Long is also very similar to the long-term time series, again shown by a highly significant Levene test statistic of 0.90 (Fig. 9c). This indicates that the isotopic variation of the An Long time series is almost identical to the one from Bangkok."

We also modified the statement acknowledging the remaining uncertainty to:

"In summary, the analyzed GNIP data suggests that the data and results from this study are likely to be representative of the Southern continental part of the Indochinese Peninsula."

Figure 8 (in the old version of the manuscript) was replaced by the following figure (Figure 9 in the revised manuscript), where the time series of Bangkok for the same period as our observation is added:

[Figure]

P14L9: Why was 124th model chosen as best?

Because the PRESS value of the 124th model is smallest. The sentence provides this information. We also stated this in the methodology section (P8L13) in the old version of the manuscript. In revised manuscript, this evidence is at P11L4.

P15L2: It is good idea. Why don't you do this trial?

We actually did this. The result are shown in Figure 12 and discussed in section 4.4 (from P15L6 to P16L8) in the old version of the manuscript. In revised manuscript, the result are shown in Figure 13 and discussed in section 4.4 (from P18L6 to P19L7).

**References**

[revised manuscript text omitted]

---

## Referee Report (RR1)

This study evaluates local vs. regional controls on the stable isotopic composition of precipitation in the Vietnamese Mekong Delta by assigning relative weights to multiple linear regression coefficients. As stated in the manuscript, distinguishing local and regional controls on precipitation isotope ratios is a critical concern for accurate interpretation of paleo-proxy records. This study applies a very thorough and novel statistical approach to disentangle these factors. However, it is not entirely clear to me how one would invert this procedure, given a record of precipitation isotope ratios, to reconstruct past climate.

Overall, I find the analysis thorough and compelling, though the methodological descriptions are a bit dense and perhaps lose some clarity in being too detailed. I would like to see the presentation condensed and reorganized in places, as well as a bit more discussion about the broader implications of this work for paleo-proxy interpretations. More specific comments are provided below.

**Introduction  - could better focus on the main story.**
1. I'd like to see a strong beginning, emphasizing the scientific question at hand. Why not make the second paragraph the lede?
2. The 4th paragraph suggests "other relevant processes were identified…" presumably for the Monsoon Region. Do all the ensuing publications specifically address the Monsoon Region?
3. The 5th paragraph suggests statistical models are "not able to represent the actual processes…" Some re-wording/re-phrasing here is required. All models are a representation. GCMs, for example, can only approximate many physical processes.
4. Limitations and assumptions of paleoclimate reconstructions discussed in the 6th paragraph are nicely described.
5. Also in the 6th paragraph: what is the difference in isotopic signatures of Indian and Pacific Ocean air?
6. Paragraph 7 and onward, some of the narrative flow is lost. What is the purpose of discussing advances and limitations of GCMs? There is a statement about developing GCM code being too daunting a task, but there is code and there are researchers actively developing it, so the argument doesn't quite make sense. Are GCMs and Lagrangian models two different ways of approaching paleoclimate reconstructions? How do these models fit with the methods used in this work? It almost seems as though Paragraphs 12 or 13 could directly follow 6: a monofactorial approach has many limitations…therefore this study suggests a multifactorial one. The multiple factors considered include both local and regional meteorological variables, with back trajectories used to characterize the regional ones. Yes, GCMs also allow one to consider both local and regional factors, but, as stated, their complexity can make interpretation difficult. Perhaps the more detailed GCM discussion could be moved to a proper discussion section. This would help focus and condense the Introduction, which would be desirable.
7. Page 5, where the importance of multiple factors in influencing precipitation are discussed, this would be a good place to introduce the need for a multiple linear

regression approach and tie this paper's statistical approach to the larger scientific questions at hand.

8. Page 5, Line 22: LMWLs should be defined for those unfamiliar with isotopic analyses. More broadly, it is not clear to me that the LMWLs play a significant role in this analysis other than to show that re-evaporation may be relevant during the dry season. It seems their presentation could be minimized. More on this below.

9. The Intro ends by emphasizing the drivers of isotopic variation. But isn't the underlying motivation using the isotopic records from the past to interpret hydroclimate? How do we go from one direction to the other?

**Study area**

10. An Long and its relationship to Cao Lanh should be described here. The best description of this is the first paragraph of Section 4.1. Specifically, the paper should describe why it is okay (or at least necessary) to interchange data from these sites.

**Methodology – could be shortened.**

11. The section begins with "An overview of the proposed methodology…" Yet this is in fact the methodology used. "Proposed" can be dropped.

12. Describing the LMWL is fairly standard practice, and the comparison of three distinct regression methods seems overkill, particularly since all three give equivalent results. I would suggest moving this sensitivity test to supporting information, which would help shorten the methods and the number of figures.

13. Similarly, the description of HYSPLIT is a bit more detailed than really necessary. I'd like to see Section 3.5 considerably shortened.

14. "Moving distance" is not clear. I believe what is intended is the distance the air parcel moved. It would be helpful to clarify that this is measured (in km?) along the parcel trajectory (as opposed to the Euclidean distance between start and finish).

15. I would suggest removing the clause "In order to derive figures representative for each trajectory…" from Line 13 on Page 9, as it is not clear.

16. Some of the remaining paragraphs on Page 9 related to HYSPLIT assumptions can be shifted to a Discussion section.

17. The first paragraph of Section 3.6 is quite clear and helpful in describing the paper's methodology.

18. Equations 2 and 3 should follow immediately after they are mentioned.

19. The number of ML regressions considered is quite impressive and reflects the thoroughness of the paper's approach.

20. I appreciate the fact that the paper openly acknowledges the correlations among predictor variables and address multicollinearity using relative weight analysis. This method will be somewhat new for many readers and should be given a bit more description. (This is one of the only sections where I would recommend expanding the text!)

21. I had assumed all weights described in the results are relative weights. Is the RPSS used as well? If so, this is not clear. Similar to my suggestion for LMWLs,

I would recommend emphasizing one method and simply stating that other methods did not provide qualitatively different results. This will help streamline the methodology tremendously and help give other researchers a roadmap for conducting a similar statistical analysis for their region(s) of interest.

**Results – could be reorganized.**

22. I might suggest a bit of reorganization (and condensing!) here: what if the section began by describing the local data, contextualized it within the larger region, then discussed the distant moisture sources to the region? This would give some additional motivation for evaluating local vs. regional controls on precipitation as the final, most important segment of this section.
23. The first paragraph really belongs in the Methods, as does description of TSV.
24. Line 30, Page 12: the d18O values are "noted" or "written" not "plotted." How about an isotopic bar chart to actually plot them? This would be much easier to "read" than the text.
25. As written, it is not clear how section 4.2.1 (LMWLs) answers the local vs. regional control question. See previous comments about shortening the presentation and discussion of LMWLs. The seasonal LMWLs do provide some evidence of secondary fractionation (re-evaporation), which is presumably a local process. But that's really the only message I took from their inclusion in this work (and it's not clear that this is the intended use of the LMWLs in the paper.)
26. It's not clear from the Methods that the GNIP data will be used to set this paper's measurements within a larger regional context. This could be stated earlier in the paper so that the reader knows to expect this and to understand how the GNIP data will be used.
27. Top of Page 14: the paper highlights differences between An Long and Bangkok, but the figure doesn't really show substantial differences. Moreover, wouldn't an unusually dry period tend to enrich An Long compared to Bangkok's climatology? I don't see this in the data. Lastly, it doesn't really make sense that one would use the sites to "represent or complement each other." Perhaps one could rephrase to say the overall similarity suggests an important role for regional or larger-scale controls on An Long precipitation isotope ratios.
28. The Levene test description can be moved to Methods.
29. Page 15 first sentence: we can't yet know that precipitation is "mainly controlled by large-scale circulation." What we infer is that it is influenced by other factors such as the large-scale circulation.
30. Page 16, Line 7: the correlations can only show a correlation, not that P-hysplit is the dominant control. Our physical understanding of isotopic responses to precipitation is what suggests precipitation is the control.
31. Section 4.4: I'm a bit confused how the MLR models are evaluated. Aren't all factors, including met variables at various heights and for various trajectory lengths considered all at once to select the best model? The section almost suggests the height and length are picked first, and then the best met variables are identified second, which wouldn't make sense. Some re-phrasing is needed.
32. Page 17 is really quite compelling and well written.

33. Up to 7 predictors for seasonal regressions with 42,18, and 14 data points is not ideal. Some discussion of this potential limitation would be useful in a proper discussion section.
34. Moreover, it would be useful to see the final best model (and which predictors are included!) for both the annual and seasonal analyses.
35. Some discussion of why dxs seems to reflect regional processes more than the individual isotope ratios would be useful. Again, this could go in a proper discussion section.

**Conclusion**
36. Page 20, Line 20: Perhaps "play a smaller role in influencing" rather than "modulate."
37. Page 21, Line 8: scratch "without a priori knowledge or assumptions." The method of course makes a priori assumptions when picking variables like P and T as predictors of the isotope ratios. Also, assumptions are made about the importance of both local and regional factors.
38. Page 21, Line 20: Where are the LMWLs of all stations compared? Perhaps this statement should just be eliminated as the LMWLs don't seem to add much to the analysis.
39. Last paragraph: again, how can we go from understanding controls on isotopes to using isotope ratios to reconstruct climate?

**Tables and Figures**
40. Table 3: d18O-d2H order should be swapped in first column, second row, to be consistent with other rows.
41. Figure 6, in addition to the isotopic bar chart suggested above, the brown text could be re-colored so it is more distinguishable from the red text.
42. Figure 10: Consider plotting the arithmetic vs. amount-weighted means as a difference for faster viewing and interpretation.
43. Figure 11 caption: the best model is "marked" or "annotated" with red text.
44. Figure 12 caption: the dots and bars in the top panel should be identified.

---

## Author Response (AR2)

Nguyen Le Duy
Section 5.4 Hydrology
Tel.: +49 331 288 1562
Fax: +49 331 288 1570
Email: duy@gfz-potsdam.de
* * *
Helmholtz Centre Potsdam
GFZ German Research Centre For Geosciences
Telegrafenberg, 14473 Potsdam, Germany

Potsdam, 10th January 2018

Dr. Lixin Wang
Editor
Hydrology and Earth System Sciences

**Re: HESS-2017-164**

Dear Dr. Wang,

enclosed please find a thoroughly revised, original manuscript now titled "**What controls the stable isotope composition of precipitation in the Mekong Delta? A model-based statistical approach**", which is renamed from the previous title "What controls the stable isotope composition of precipitation in the Asian monsoon region?" (Reference #HESS-2017-164) by Nguyen Le Duy, Ingo Heidbüchel, Hanno Meyer, Bruno Merz, Heiko Apel. We are respectfully submitting our revised manuscript again for your consideration in Hydrology and Earth System Sciences.

We have thoroughly revised the paper to take into consideration the constructive comments from the second referee. The following revisions have been made:
- the abstract was slightly extended
- the introduction was completely reworked (focusing more specifically on highlighting the novelties and recent literature)
- the study area description was extended (based on reviewers' feedback)
- the methodology section 3.5 was changed (deleting unnecessary parts, but adding a discussion about the uncertainties of trajectory analysis and applied measures to mitigate these uncertainties), but other parts were shortened (LMWL regressions)
- the results and discussion section was rewritten (based on reviewers' feedback)
- the conclusions were extended (highlighting the novelties and adding some thought on the implications for paleo-climate reconstruction).

This manuscript has neither been previously published in any language nor is it under consideration for publication by another journal. All authors have carefully read the revised manuscript and have agreed to its submission to Hydrology and Earth System Sciences. All results and innovations were developed by the authors using Matlab. Figures were generated using ArcGIS and Matlab. We also published the isotopic data in the open access data repository of GFZ. The data is already available to reviewers under:

http://pmd.gfz-potsdam.de/panmetaworks/review/9e1af507c8fce65a8d740033e5fea31c2e7c58ade81762c235c6f6bbab91166e/

Thank you for handling the manuscript during the review process, and to the reviewers for their valuable feedback and edits. We look forward to hearing from you.

Sincerely yours,
Nguyen Le Duy
Corresponding Author

**RESPONSE TO THE REFEREES' COMMENTS**

We sincerely thank the first referee for the acceptance and the second referee for his/her thorough reviews and most constructive comments on our manuscript (Reference #HESS-2017-164). We fully appreciate the reviewers' efforts in providing these informative reports on our research and their insights have led to an improved interpretation of our results. We have taken into full consideration all of these comments and have prepared responses to these as well as information on how the paper was revised following the referees' suggestions. Our responses are provided below **in blue** following the individual comments requiring action from the second reviewer, followed by a marked up version of the manuscript (all changes as appearing in the text are marked **in red**).

**Anonymous Referee #1**

We thank the first anonymous referee the suggestion to accept the manuscript for publication.

**Anonymous Referee #2**

This study evaluates local vs. regional controls on the stable isotopic composition of precipitation in the Vietnamese Mekong Delta by assigning relative weights to multiple linear regression coefficients. As stated in the manuscript, distinguishing local and regional controls on precipitation isotope ratios is a critical concern for accurate interpretation of paleo-proxy records. This study applies a very thorough and novel statistical approach to disentangle these factors. However, it is not entirely clear to me how one would invert this procedure, given a record of precipitation isotope ratios, to reconstruct past climate.

Overall, I find the analysis thorough and compelling, though the methodological descriptions are a bit dense and perhaps lose some clarity in being too detailed. I would like to see the presentation condensed and reorganized in places, as well as a bit more discussion about the broader implications of this work for paleo-proxy interpretations. More specific comments are provided below.

We thank the second referee for the positive and constructive comments. Our answers are also included in the revised version of the manuscript.

**Introduction - could better focus on the main story.**

1. I'd like to see a strong beginning, emphasizing the scientific question at hand. Why not make the second paragraph the lede?

Thank you for this recommendation. We deleted the first paragraph in the Introduction to show a strong beginning the scientific question.

2. The 4th paragraph suggests "other relevant processes were identified…" presumably for the Monsoon Region. Do all the ensuing publications specifically address the Monsoon Region?

We removed reference (Pausata et al., 2011) of which study sites are not explicitly related to the Monsoon Region. All other references were checked for relevance.

3. The 5th paragraph suggests statistical models are "not able to represent the actual processes…" Some re-wording/re-phrasing here is required. All models are a representation. GCMs, for example, can only approximate many physical processes.

The paragraph was reworded.

4. Limitations and assumptions of paleoclimate reconstructions discussed in the 6th paragraph are nicely described.

Thank you for your nice comment.

5. Also in the 6th paragraph: what is the difference in isotopic signatures of Indian and Pacific Ocean air?

The sentence was re-written as follows:

"The isotopic signatures of air masses originating from the Indian Ocean differing considerably from those of the Pacific Ocean, where the average δ18O of the latter is about 2.5‰ more negative (Araguás‐Araguás et al., 1998)".

6. Paragraph 7 and onward, some of the narrative flow is lost. What is the purpose of discussing advances and limitations of GCMs? There is a statement about developing GCM code being too daunting a task, but there is code and there are researchers actively developing it, so the argument doesn't quite make sense. Are GCMs and Lagrangian models two different ways of approaching paleoclimate reconstructions? How do these models fit with the methods used in this work? It almost seems as though Paragraphs 12 or 13 could directly follow 6: a monofactorial approach has many limitations…therefore this study suggests a multifactorial one. The multiple factors considered include both local and regional meteorological variables, with back trajectories used to characterize the regional ones. Yes, GCMs also allow one to consider both local and regional factors, but, as stated, their complexity can make interpretation difficult. Perhaps the more detailed GCM discussion could be moved to a proper discussion section. This would help focus and condense the Introduction, which would be desirable.

This part has been shortened considerably. The discussion of the advantages/disadvantages of the Eulerian and Lagrangian approaches has been dropped.

7. Page 5, where the importance of multiple factors in influencing precipitation are discussed, this would be a good place to introduce the need for a multiple linear regression approach and tie this paper's statistical approach to the larger scientific questions at hand.

We followed this suggestion and introduced the MLR here.

8. Page 5, Line 22: LMWLs should be defined for those unfamiliar with isotopic analyses. More broadly, it is not clear to me that the LMWLs play a significant role in this analysis other than to show that re-evaporation may be relevant during the dry season. It seems their presentation could be minimized. More on this below.

We dropped the notion of the LMWL in the introduction, as suggested, because it distracts from the actual focus of the study.

9. The Intro ends by emphasizing the drivers of isotopic variation. But isn't the underlying motivation using the isotopic records from the past to interpret hydroclimate? How do we go from one direction to the other?

We end the Introduction with the mentioning of the study area to test the proposed method for the identification of the primary processes controlling isotopic composition, which is the core aspect of the manuscript.

**Study area**

10. An Long and its relationship to Cao Lanh should be described here. The best description of this is the first paragraph of Section 4.1. Specifically, the paper should describe why it is okay (or at least necessary) to interchange data from these sites.

We added a paragraph describing the need for using data from Cao Lanh, as well as justifying the usage of the data:

"The local climate of An Long is described by data from Cao Lanh station. Cao Lanh is the closest

national meteorological station to An Long with continuous climate records, located approximately 37 km Southeast of An Long. It is assumed that the climatic conditions of An Long and Cao Lanh are similar. The proximity of the two locations and the similar geographical setting (flat topography, located at the Eastern bank of the Mekong river) justify this assumption."

**Methodology – could be shortened.**

11. The section begins with "An overview of the proposed methodology…" Yet this is in fact the methodology used. "Proposed" can be dropped.

"Proposed" was dropped.

12. Describing the LMWL is fairly standard practice, and the comparison of three distinct regression methods seems overkill, particularly since all three give equivalent results. I would suggest moving this sensitivity test to supporting information, which would help shorten the methods and the number of figures.

We present and describe only on regression method in the main manuscript, and moved the other two for reference to the supplementary material.

13. Similarly, the description of HYSPLIT is a bit more detailed than really necessary. I'd like to see Section 3.5 considerably shortened.

We changed this paragraph a bit. Specifically, we removed sentences which are not necessary from the section 3.5 and moved the paragraphs discussing "uncertainties" and "quality control measures to increase confidence of backward trajectories" to Discussion (in section 4.1)

14. "Moving distance" is not clear. I believe what is intended is the distance the air parcel moved. It would be helpful to clarify that this is measured (in km?) along the parcel trajectory (as opposed to the Euclidean distance between start and finish).

We clarified this, writing:

"The HYSPLIT outputs, i.e., precipitation, temperature, relative humidity along the backward trajectories, and the length of trajectories (the distance of moisture sources traveled), were used as regional factors potentially controlling the variation of the isotopic composition of precipitation at An Long. Accumulated precipitation, mean values of temperature and humidity of the hourly HYSPLIT output were calculated along the trajectory, as well as the length of the trajectory. All these factors were used as predictors in the MLR."

15. I would suggest removing the clause "In order to derive figures representative for each trajectory…" from Line 13 on Page 9, as it is not clear.

Thank you for this suggestion. We deleted "In order to derive figures representative for each trajectory" from the sentence.

16. Some of the remaining paragraphs on Page 9 related to HYSPLIT assumptions can be shifted to a Discussion section.

As mentioned above, the remaining paragraphs on Page 9 were move to the discussion.

17. The first paragraph of Section 3.6 is quite clear and helpful in describing the paper's methodology.

Thank you for your nice comment.

18. Equations 2 and 3 should follow immediately after they are mentioned.

Equations 2 and 3 were moved to follow immediately after they are mentioned.

19. The number of ML regressions considered is quite impressive and reflects the thoroughness of the paper's approach.

Thank you for your nice comment.

20. I appreciate the fact that the paper openly acknowledges the correlations among predictor variables and address multicollinearity using relative weight analysis. This method will be somewhat new for many readers and should be given a bit more description. (This is one of the only sections where I would recommend expanding the text!)

We re-wrote the section. Please find it in the revised manuscript.

21. I had assumed all weights described in the results are relative weights. Is the RPSS used as well? If so, this is not clear. Similar to my suggestion for LMWLs, I would recommend emphasizing one method and simply stating that other methods did not provide qualitatively different results. This will help streamline the methodology tremendously and help give other researchers a roadmap for conducting a similar statistical analysis for their region(s) of interest.

Thank you for this suggestion. We removed RPSS and used only Johnson's relative weights for the relative importance analysis in the study.

**Results – could be reorganized.**

22. I might suggest a bit of reorganization (and condensing!) here: what if the section began by describing the local data, contextualized it within the larger region, then discussed the distant moisture sources to the region? This would give some additional motivation for evaluating local vs. regional controls on precipitation as the final, most important segment of this section.

We followed the suggestion and re-structured the result part. However, we did not separate it into Results and Discussion, because this would unnecessarily lengthen the manuscript. Some results need to be repeated in order to understand the discussion, if separated from the results.

23. The first paragraph really belongs in the Methods, as does description of TSV.

This paragraph was moved to Methodology with some editing (see section 3.1).

24. Line 30, Page 12: the d18O values are "noted" or "written" not "plotted." How about an isotopic bar chart to actually plot them? This would be much easier to "read" than the text.

Thank you for the good ideal. The word "plotted" was replaced by "noted". We also added isotopic bar charts to the Figure 6 and re-color the text for a better view.

25. As written, it is not clear how section 4.2.1 (LMWLs) answers the local vs. regional control question. See previous comments about shortening the presentation and discussion of LMWLs. The seasonal LMWLs do provide some evidence of secondary fractionation (re-evaporation), which is presumably a local process. But that's really the only message I took from their inclusion in this work (and it's not clear that this is the intended use of the LMWLs in the paper.)

The presentation and discussion of the LMWL were shortened to the minimum, showing only one regression result and illustrating the evidence of some secondary fractionation process.

26. It's not clear from the Methods that the GNIP data will be used to set this paper's measurements within a larger regional context. This could be stated earlier in the paper so that the reader knows to expect this and to understand how the GNIP data will be used.

We underlined the use and purpose of the GNIP data in the method section.

27. Top of Page 14: the paper highlights differences between An Long and Bangkok, but the figure doesn't really show substantial differences. Moreover, wouldn't an unusually dry period tend to enrich An Long compared to Bangkok's climatology? I don't see this in the data. Lastly, it doesn't really make sense that one would use the sites to "represent or complement each other." Perhaps one could rephrase to say the overall similarity suggests

an important role for regional or larger-scale controls on An Long precipitation isotope ratios.

We rephrased this part to clarify the rational for the comparison with the variability of the isotopic composition with Bangkok: we want to show that the results obtained are likely representative not only for An Long, but also for a larger region.

28. The Levene test description can be moved to Methods.

The Levene test description was moved to section 3.1.

29. Page 15 first sentence: we can't yet know that precipitation is "mainly controlled by large-scale circulation." What we infer is that it is influenced by other factors such as the large-scale circulation.

Thank you for the comment. We replaced "mainly controlled by" by "influenced by other factors such as".

30. Page 16, Line 7: the correlations can only show a correlation, not that P-hysplit is the dominant control. Our physical understanding of isotopic responses to precipitation is what suggests precipitation is the control.

Thank you for this comment. We removed the sentence "Thus, P_hysplit is likely the dominant factor controlling the isotopic composition of precipitation. " from the manuscript.

31. Section 4.4: I'm a bit confused how the MLR models are evaluated. Aren't all factors, including met variables at various heights and for various trajectory lengths considered all at once to select the best model? The section almost suggests the height and length are picked first, and then the best met variables are identified second, which wouldn't make sense. Some re-phrasing is needed.

We clarified this: all factor combinations are used for all travel distances and pressure levels.

32. Page 17 is really quite compelling and well written.

Thank you for your nice comment.

33. Up to 7 predictors for seasonal regressions with 42,18, and 14 data points is not ideal. Some discussion of this potential limitation would be useful in a proper discussion section.

We now explicitly acknowledge the limitation/uncertainty introduced by the low seasonal sample size, but also give reasons for the validity of the findings.

34. Moreover, it would be useful to see the final best model (and which predictors are included!) for both the annual and seasonal analyses.

We added the Table 5 which show the final best models (both for annual and seasonal analyses) for $\delta^{18}O$, $\delta^2H$, and d-excess as the response variable in MLR.

35. Some discussion of why dxs seems to reflect regional processes more than the individual isotope ratios would be useful. Again, this could go in a proper discussion section.

We extended the discussion on the d-excess and provided reasons for the different factors controlling the d-excess in different seasons. We also included a statement about the uncertainty stemming from the limited sample size in the dry season regression.

**Conclusion**

36. Page 20, Line 20: Perhaps "play a smaller role in influencing" rather than "modulate."

Thank you. We replaced "modulate" by "play a smaller role in influencing" as your suggestion.

37. Page 21, Line 8: scratch "without a priori knowledge or assumptions." The method of

course makes a priori assumptions when picking variables like P and T as predictors of the isotope ratios. Also, assumptions are made about the importance of both local and regional factors.

Thank you. "without a priori knowledge or assumption" was removed.

38. Page 21, Line 20: Where are the LMWLs of all stations compared? Perhaps this statement should just be eliminated as the LMWLs don't seem to add much to the analysis.

We deleted this statement and refer to the test for similarity in variance instead.

39. Last paragraph: again, how can we go from understanding controls on isotopes to using isotope ratios to reconstruct climate?

We extended this part accordingly. It now reads:

"The results have direct implications for the interpretation of paleorecords of stable water isotopes in terms of past climate conditions for Southeast Asia. This study shows that the factors controlling the isotopic signature of precipitation are changing between and even within seasons, and that regional factors have substantial impacts on the local isotopic composition of rainfall. This needs to be considered in the reconstruction of past climates based on isotopic records: for the presented study area δ18O and δ2H values are likely to be representative of the local climatic conditions during the dry season. However, regional factors dominate during most of the rainy season receiving the bulk of the total annual rainfall. In this case, reconstructions of past climates based on paleo isotopic records would have to be carefully interpreted. The proposed approach might open a pathway for an improved reconstruction of paleoclimates based on isotopic records. It may e.g. be used for identifying suitable variables to improve the performance of proxy data assimilation in paleoclimate reconstruction by circulation models. Moreover, assuming that the general circulation is stable over the period considered in paleoclimate reconstruction, which is reasonable for e.g. the Holocene, the presented findings can be used to infer moisture source regions and the strength of the two monsoonal regimes influencing SE-Asia from paleo isotopic records."

**Tables and Figures**

40. Table 3: d18O-d2H order should be swapped in first column, second row, to be consistent with other rows.

Agree. We edited the table as your suggestion.

41. Figure 6, in addition to the isotopic bar chart suggested above, the brown text could be re-colored so it is more distinguishable from the red text.

Thank you. We added the isotopic bar chart and re-colored the text for a better view as your suggestion.

42. Figure 10: Consider plotting the arithmetic vs. amount-weighted means as a difference for faster viewing and interpretation.

Agree. We changed the figure as your suggestion.

43. Figure 11 caption: the best model is "marked" or "annotated" with red text.

Thank you. The word "highlight" was replaced by "marked".

44. Figure 12 caption: the dots and bars in the top panel should be identified.

Thank you. We re-colored the figure.

[revised manuscript text omitted]